# γδ T cells are effectors of immunotherapy in cancers with HLA class I defects

Natasja L. de Vries[1,2,13], Joris van de Haar[3,4,5,13], Vivien Veninga[3,4,13], Myriam Chalabi[3,6,7,13], Marieke E. Ijsselsteijn[1], Manon van der Ploeg[1], Jitske van den Bulk[1], Dina Ruano[1], Jose G. van den Berg[8], John B. Haanen[3,7], Laurien J. Zeverijn[3,4], Birgit S. Geurts[3,4], Gijs F. de Wit[3,4], Thomas W. Battaglia[3,4], Hans Gelderblom[9], Henk M. W. Verheul[10], Ton N. Schumacher[3,4,11], Lodewyk F. A. Wessels[4,5,12], Frits Koning[2,14], Noel F. C. C. de Miranda[1,14 ✉] & Emile E. Voest[3,4,14 ✉]

DNA mismatch repair-deficient (MMR-d) cancers present an abundance of neoantigens that is thought to explain their exceptional responsiveness to immune checkpoint blockade (ICB)[1,2]. Here, in contrast to other cancer types[3–5], we observed that 20 out of 21 (95%) MMR-d cancers with genomic inactivation of β2-microglobulin (encoded by *B2M*) retained responsiveness to ICB, suggesting the involvement of immune effector cells other than CD8[+] T cells in this context. We next identified a strong association between *B2M* inactivation and increased infiltration by γδ T cells in MMR-d cancers. These γδ T cells mainly comprised the Vδ1 and Vδ3 subsets, and expressed high levels of PD-1, other activation markers, including cytotoxic molecules, and a broad repertoire of killer-cell immunoglobulin-like receptors. In vitro, PD-1[+] γδ T cells that were isolated from MMR-d colon cancers exhibited enhanced reactivity to human leukocyte antigen (HLA)-class-I-negative MMR-d colon cancer cell lines and *B2M*-knockout patient-derived tumour organoids compared with antigen-presentation-proficient cells. By comparing paired tumour samples from patients with MMR-d colon cancer that were obtained before and after dual PD-1 and CTLA-4 blockade, we found that immune checkpoint blockade substantially increased the frequency of γδ T cells in B2M-deficient cancers. Taken together, these data indicate that γδ T cells contribute to the response to immune checkpoint blockade in patients with HLA-class-I-negative MMR-d colon cancers, and underline the potential of γδ T cells in cancer immunotherapy.

ICB targeting the PD-1–PD-L1 and/or CTLA-4 axes provides durable clinical benefits to patients who have cancers with MMR-d and high microsatellite instability[6–9]. The exceptional responses of cancers with MMR-d and high microsatellite instability to ICB is thought to be explained by their substantial burden of putative neoantigens, which originate from the extensive accumulation of mutations in their genomes[1,2]. This is consistent with the current view that PD-1 blockade mainly boosts endogenous antitumour immunity driven by CD8[+] T cells, which recognize HLA-class-I-bound neoepitopes on cancer cells[10–12]. However, MMR-d colon cancers frequently lose HLA-class-I-mediated antigen presentation due to silencing of HLA class I genes, inactivating mutations in β2-microglobulin (encoded by *B2M*) or other defects in the antigen processing machinery[13–16], which can render these tumours resistant to CD8[+] T-cell-mediated immunity[3–5,17]. Notably, early evidence has indicated that B2M-deficient, MMR-d cancers can obtain durable

responses to PD-1 blockade[18], suggesting that immune cell subsets other than CD8[+] T cells contribute to these responses.

HLA-class-I-unrestricted immune cell subsets, which have the ability to kill tumour cells, include natural killer (NK) cells and γδ T cells. γδ T cells share many characteristics with their αβ T cell counterpart, such as cytotoxic effector functions, but express a distinct TCR that is composed of a γ and a δ chain. Different subsets of γδ T cells are defined by their TCR δ chain use, of which those expressing Vδ1 and Vδ3 are primarily 'tissue-resident' at mucosal sites, whereas those expressing Vδ2 are mainly found in blood[19]. Both adaptive and innate mechanisms of activation—for example, through stimulation of their γδ TCR or innate receptors such as NKG2D, DNAM-1, NKp30 or NKp44—have been described for γδ T cells[20]. Killer-cell immunoglobulin-like receptors (KIRs) are expressed by γδ T cells and regulate their activity depending on HLA class I expression in target cells[21]. Furthermore, γδ T cells

[1]Department of Pathology, Leiden University Medical Center, Leiden, The Netherlands. [2]Department of Immunology, Leiden University Medical Center, Leiden, The Netherlands. [3]Department of Molecular Oncology and Immunology, Netherlands Cancer Institute, Amsterdam, The Netherlands. [4]Oncode Institute, Utrecht, The Netherlands. [5]Division of Molecular Carcinogenesis, Netherlands Cancer Institute, Amsterdam, The Netherlands. [6]Gastrointestinal Oncology, Netherlands Cancer Institute, Amsterdam, The Netherlands. [7]Medical Oncology, Netherlands Cancer Institute, Amsterdam, The Netherlands. [8]Department of Pathology, Netherlands Cancer Institute, Amsterdam, The Netherlands. [9]Department of Medical Oncology, Leiden University Medical Center, Leiden, The Netherlands. [10]Department of Medical Oncology, Erasmus MC, Rotterdam, The Netherlands. [11]Department of Hematology, Leiden University Medical Center, Leiden, The Netherlands. [12]Faculty of EEMCS, Delft University of Technology, Delft, The Netherlands. [13]These authors contributed equally: Natasja L. de Vries, Joris van de Haar, Vivien Veninga, Myriam Chalabi. [14]These authors jointly supervised this work: Frits Koning, Noel F. C. C. de Miranda, Emile E. Voest. ✉e-mail: n.f.de_miranda@lumc.nl; e.voest@nki.nl

were found to express high levels of PD-1 in MMR-d colorectal cancers (CRCs)[22], suggesting that these cells may be targeted by PD-1 blockade.

Here, we applied a combination of transcriptomic and imaging approaches for an in-depth analysis of ICB-naive and ICB-treated MMR-d colon cancers, as well as in vitro functional assays, and found evidence indicating that γδ T cells mediate responses to HLA-class-I-negative MMR-d tumours during treatment with ICB.

## ICB is effective in *B2M$^{MUT}$* MMR-d cancers

We evaluated responses to PD-1 blockade therapy in a cohort of 71 patients with MMR-d cancers from various anatomical sites treated in the Drug Rediscovery Protocol (DRUP)[23] in relation to their *B2M* status (Fig. 1a, Extended Data Fig. 1a–c and Supplementary Table 1). A clinical benefit (CB; defined as at least 4 months of disease control; the primary outcome of the DRUP) was observed in 20 out of 21 (95%) of patients with tumours with mutant or deleted *B2M* (*B2M$^{MUT}$*) tumours versus 31 out of 50 (62%) of patients with tumours with wild-type *B2M* (*B2M$^{WT}$*) (two-sided Fisher's exact test, $P = 0.0038$; logistic regression, $P = 0.022$ and $P = 0.027$, adjusted for tumour mutational burden (TMB), and TMB plus tumour type, respectively; Fig. 1b). Among patients with *B2M$^{MUT}$* tumours, 3 out of 21 (14%) individuals experienced a complete response (according to RECIST1.1 criteria), 12 (57%) experienced a partial response, 5 (24%) experienced a durable stable disease and 1 (4.8%) experienced progressive disease as the best overall response. All 44 *B2M* alterations across 21 patients were clonal (Methods), consistent with previous observations in MMR-d cancers[18]. A total of 13 out of 21 (62%) patients with *B2M$^{MUT}$* tumours had biallelic *B2M* alterations, 4 (19%) had potentially biallelic alterations and 4 (19%) had non-biallelic alterations (Fig. 1c and Methods). Non-biallelic alterations have also been associated with complete loss of B2M protein expression in MMR-d tumours[18]. Thus, *B2M* alterations are associated with a high clinical benefit rate of PD-1 blockade in patients with MMR-d cancers.

## Vδ1 and Vδ3 TCRs are overexpressed in *B2M$^{MUT}$* cancers

To gain insights into the immune cell subsets that are involved in immune responses to HLA-class-I-negative MMR-d cancers, we used data of The Cancer Genome Atlas (TCGA) and studied the transcriptomic changes associated with the genomic loss of *B2M* in three cohorts of individuals with MMR-d cancer in colon adenocarcinoma (COAD; $n = 50$ (*B2M$^{WT}$*), $n = 7$ (*B2M$^{MUT}$*)), stomach adenocarcinoma (STAD; $n = 48$ (*B2M$^{WT}$*) and $n = 12$ (*B2M$^{MUT}$*)), and endometrium carcinoma (UCEC; $n = 118$ (*B2M$^{WT}$*) and $n = 4$ (*B2M$^{MUT}$*)). We found that *B2M* was among the most significantly downregulated genes in *B2M$^{MUT}$* cancers (two-sided limma-voom-based regression, $P = 3.5 \times 10^{-4}$, Benjamini–Hochberg false-discovery rate (FDR)-adjusted $P = 0.12$, adjusted for tumour type; Fig. 1d). Genes encoding components of the HLA class I antigen presentation machinery other than *B2M* were highly upregulated in *B2M$^{MUT}$* tumours, which may reflect reduced evolutionary pressure on somatic inactivation of these genes in the *B2M$^{MUT}$* context[18] (Fig. 1d). Notably, we found *TRDV1* and *TRDV3*, which encode the variable regions of the δ1 and δ3 chains of the γδ T cell receptor (TCR), among the most significantly upregulated loci in *B2M$^{MUT}$* tumours (*TRDV1*, two-sided limma-voom-based regression, FDR-adjusted $P = 0.00090$, adjusted for tumour type; *TRDV3*, two-sided limma-voom-based regression, FDR-adjusted $P = 0.0015$, adjusted for tumour type; Fig. 1d), regardless of the allelic status of the *B2M* alteration (Extended Data Fig. 1d). Consistent with this, the expression levels of *TRDV1* and *TRDV3* were higher in *B2M$^{MUT}$* compared with in *B2M$^{WT}$* MMR-d cancers (two-sided Wilcoxon rank-sum test, $P = 6.5 \times 10^{-8}$ for all of the cohorts combined; two-sided linear regression, $P = 4.7 \times 10^{-6}$, adjusted for tumour type; Fig. 1d–f). Moreover, *B2M$^{MUT}$* tumours showed overexpression of multiple KIRs

(Fig. 1d), which clustered together with *TRDV1* and *TRDV3* on the basis of hierarchical clustering (Extended Data Fig. 1e). The expression level of different KIRs (Supplementary Table 2) was higher in *B2M$^{MUT}$* tumours compared with in *B2M$^{WT}$* MMR-d tumours (two-sided Wilcoxon rank-sum test, $P = 4.4 \times 10^{-6}$ for all cohorts combined; two-sided linear regression, $P = 4.7 \times 10^{-5}$, adjusted for tumour type; Fig. 1d–f). Together, these results suggest that ICB-naive *B2M$^{MUT}$* MMR-d cancers show increased levels of Vδ1 and Vδ3 T cells as well as increased numbers of these or other immune cells expressing KIRs—a potential mechanism of recognition of HLA class I loss.

We used marker gene sets (modified from ref. [24]; Methods and Supplementary Table 2) to estimate the abundance of a broad set of other immune cell types on the basis of the RNA expression data of the TCGA cohorts. Hierarchical clustering identified a high- and a low-infiltrated cluster in each of the three tumour types (Fig. 1e). Compared with the Vδ1 and Vδ3 T cell and KIR gene sets, the other marker gene sets showed no or only weak association between expression level and *B2M* status, indicating that our findings were not solely driven by a generally more inflamed state of *B2M$^{MUT}$* tumours (Fig. 1e,f and Extended Data Fig. 1f).

We next revisited the DRUP cohort and specifically applied the marker gene sets to RNA expression data. Despite the low patient numbers and high heterogeneity regarding tumour types and biopsy locations of this cohort, we confirmed increased *TRDV1* and *TRDV3* expression in *B2M$^{MUT}$* tumours pan-cancer (two-sided linear regression, $P = 0.017$, adjusted for tumour type and biopsy site; Fig. 1g, Extended Data Fig. 1g and Methods). KIR expression was significantly associated with *B2M* status only in CRC (Fig. 1g). Across mismatch repair-proficient (MMR-p) metastatic cancers in the Hartwig database[25], 36 out of 2,256 (1.6%) cancers had a clonal *B2M* alteration, which was frequently accompanied by loss of heterozygosity (LOH) (Extended Data Fig. 1h and Supplementary Table 3). Although rare, *B2M* alterations were also significantly associated with increased expression of *TRDV1/TRDV3* loci in this context (two-sided linear regression, $P = 2.2 \times 10^{-17}$, adjusted for tumour type; Extended Data Fig. 1i and Methods). Taken together, *B2M* defects are positively associated with clinical benefits of ICB treatment, as well as infiltration by Vδ1 and Vδ3 T cells and expression of KIRs.

## Vδ1 and Vδ3 T cells are activated in MMR-d CRC

To investigate which γδ T cell subsets are present in MMR-d colon cancers and to determine their functional characteristics, we performed single-cell RNA-sequencing (scRNA-seq) analysis of γδ T cells isolated from five MMR-d colon cancers (Extended Data Figs. 2 and 3 and Supplementary Table 4). Three distinct Vδ subsets were identified (Fig. 2a)— Vδ1 T cells were the most prevalent (43% of γδ T cells), followed by Vδ2 (19%) and Vδ3 T cells (11%) (Fig. 2b). *PDCD1* (encoding PD-1) was predominantly expressed by Vδ1 and Vδ3 T cells, whereas Vδ1 cells expressed high levels of genes that encode activation markers such as CD39 (*ENTPD1*) and CD38 (Fig. 2c and Extended Data Fig. 2b). Furthermore, proliferating γδ T cells (expressing *MKI67*) were especially observed in the Vδ1 and Vδ3 subsets (Fig. 2c). Other distinguishing features of the Vδ1 and Vδ3 T cell subsets included the expression of genes encoding activating receptors NKp46 (encoded by *NCR1*), NKG2C (encoded by *KLRC2*) and NKG2D (encoded by *KLRK1*) (Fig. 2c). Notably, the expression of several KIRs was also higher in the Vδ1 and Vδ3 subsets as compared to Vδ2 T cells (Fig. 2c). Almost all γδ T cells displayed expression of the genes encoding granzyme B (*GZMB*), perforin (*PRF1*) and granulysin (*GNLY*) (Fig. 2c). Together, these data support a role for γδ T cells in mediating natural cytotoxic antitumour responses in HLA-class-I-negative MMR-d colon cancers.

Next, we applied imaging mass cytometry (IMC) analysis to a cohort of 17 individuals with ICB-naive MMR-d colon cancers (Supplementary Table 4). High levels of γδ T cell infiltration were observed in cancers with B2M defects as compared to B2M-proficient cancers, albeit this

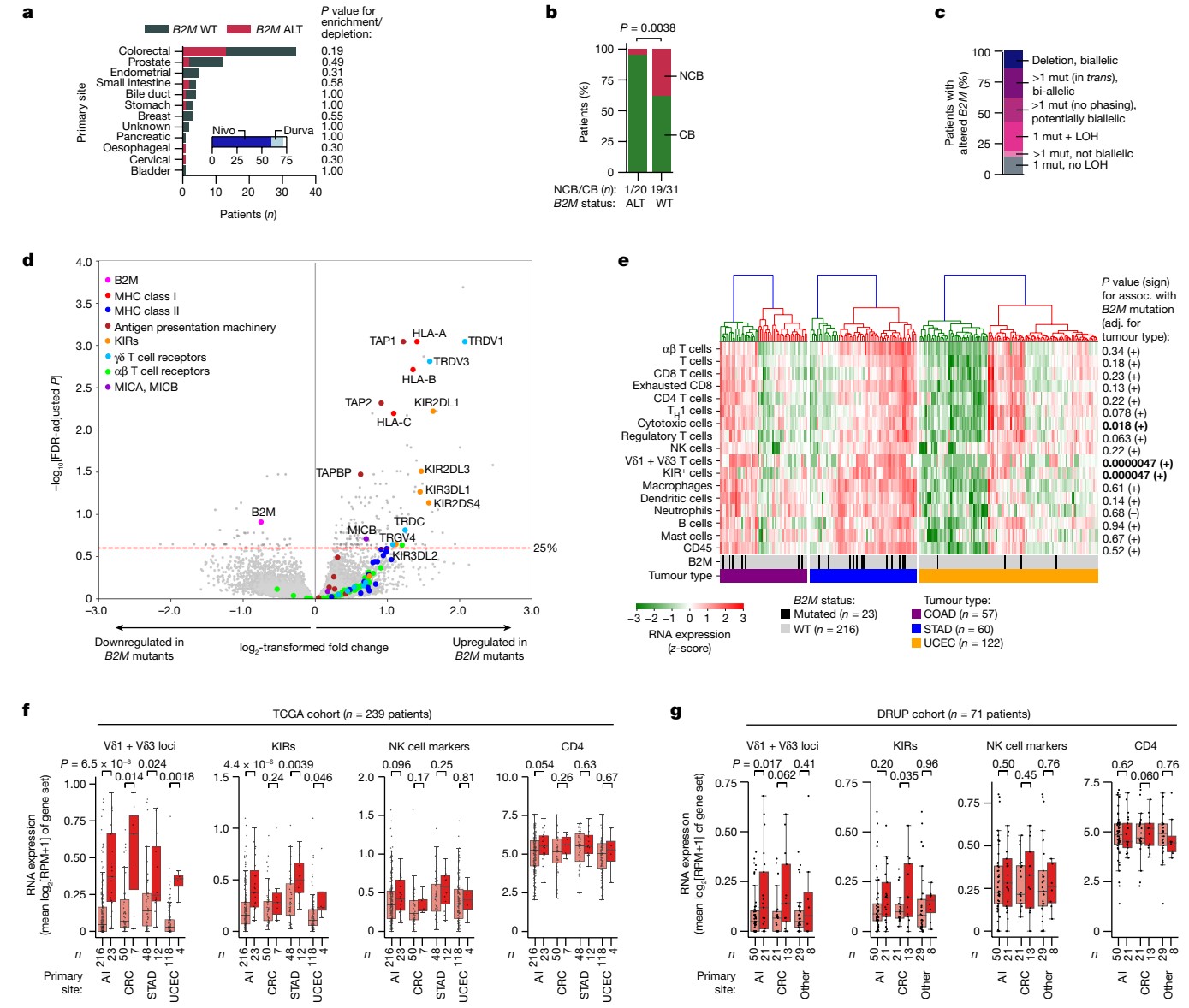

**Fig. 1 | In MMR-d cancers, *B2M* defects are positively associated with ICB responsiveness and infiltration by Vδ1 and Vδ3 T cells and KIR-expressing cells. a**, Tumour type distribution in the DRUP cohort (*n* = 71 patients). The colours denote patients' *B2M* status; grey, WT; red, altered (ALT). *P* values for the enrichment/depletion of *B2M*-altered tumours per primary site were calculated using two-sided Fisher's exact tests. The inset denotes the ICB treatment; dark blue, nivolumab (Nivo); light blue, durvalumab (Durva). **b**, *B2M* status (*x* axis) versus clinical benefit (green, CB; red, no clinical benefit (NCB)) of ICB treatment in the DRUP cohort. The *P* value was calculated using a two-sided Fisher's exact test. **c**, The allelic status of *B2M* alterations in the DRUP cohort. Mut, mutation. **d**, Differential gene expression between *B2M*MUT and *B2M*WT MMR-d cancers in the TCGA COAD (colon adenocarcinoma; *n* = 57 patients), STAD (stomach adenocarcinoma; *n* = 60 patients) and UCEC (uterus corpus endometrial carcinoma; *n* = 122 patients) cohorts. The results were adjusted (adj.) for tumour type and multiple-hypothesis testing (Methods). **e**, Immune marker gene set expression in MMR-d cancers of the COAD, STAD and UCEC cohorts of the TCGA. The bottom two bars indicate *B2M* status and cancer type. The association (assoc.) between gene set expression and *B2M* status was tested using ordinary least squares linear regression (adjusted for tumour type; Methods), of which two-sided *P* values and the association sign are shown on the right. Cancers were ranked on the basis of hierarchical clustering (top dendrograms). *P* values less than 0.05 are in bold. **f**, Immune marker gene set expression in *B2M*WT (pink) and *B2M*MUT (red) MMR-d cancers in the TCGA COAD, STAD and UCEC cohorts separately or combined (all). Boxes, whiskers and dots indicate the quartiles, 1.5× the interquartile range (IQR) and individual data points, respectively. *P* values were calculated using two-sided Wilcoxon rank-sum tests. **g**, Immune marker gene set expression in *B2M*WT (pink) and *B2M*MUT (red) as described in **f**, but for MMR-d cancers in the DRUP cohort. Results are shown for all cancers combined, only CRC or all non-CRC cancers (other). Two-sided *P* values were calculated using linear regression, adjusting for biopsy site and tumour type (Methods).

difference was not significant (Fig. 2d). The levels of other immune cells, including NK cells, CD4+ T cells and CD8+ T cells, were similar between B2M-deficient and B2M-proficient tumours (Fig. 2d). In B2M-deficient cancers, γδ T cells showed frequent intraepithelial localization and expression of CD103 (tissue-residency), CD39 (activation), granzyme

B (cytotoxicity) and Ki-67 (proliferation), as well as PD-1 (Fig. 2d–f and Extended Data Fig. 2c), consistent with the scRNA-seq data. Notably, γδ T cells in B2M-deficient cancers showed co-expression of CD103 and CD39 (Extended Data Fig. 2d), which has been reported to identify tumour-reactive CD8+ αβ T cells in a variety of cancers[26].

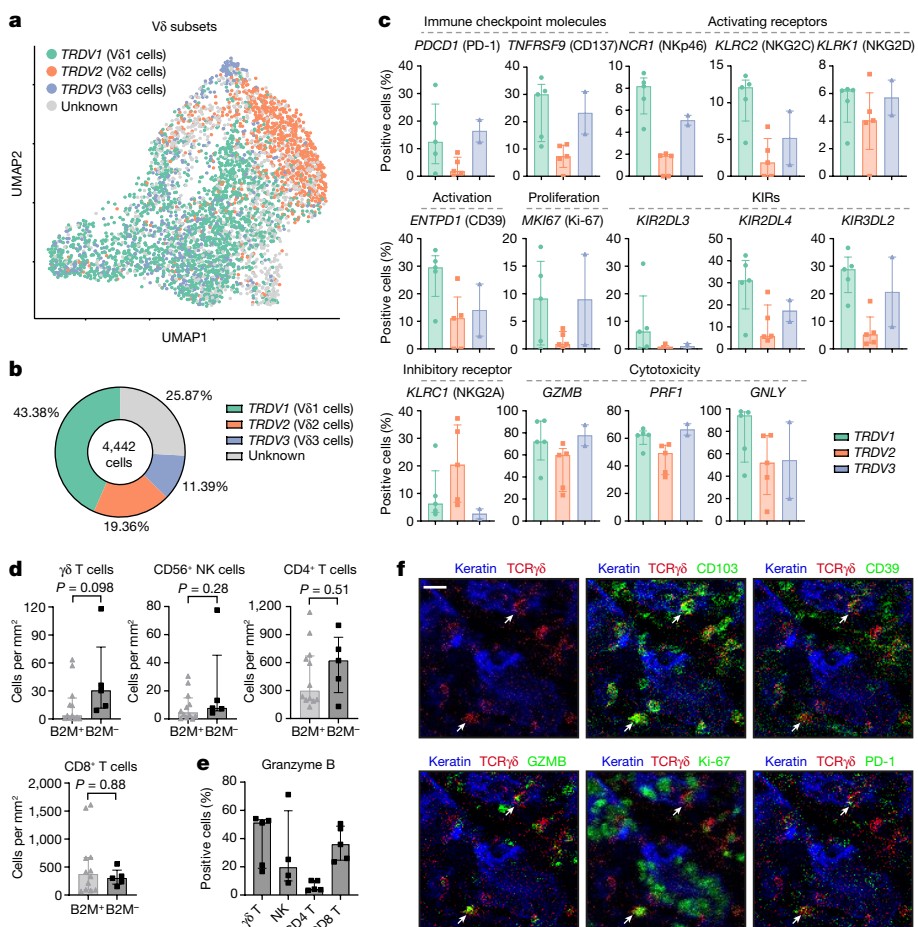

**Fig. 2 | Tumour-infiltrating Vδ1 and Vδ3 T cell subsets display hallmarks of cytotoxic activity in MMR-d colon cancers. a**, UMAP embedding showing the clustering of γδ T cells (n = 4,442) isolated from MMR-d colon cancers (n = 5) analysed using scRNA-seq. The colours represent the TCR Vδ chain usage. The functionally distinct γδ T cell clusters are shown in Extended Data Fig. 3. Dots represent single cells. **b**, The frequencies of TCR Vδ chain use of the γδ T cells (n = 4,442) analysed using scRNA-seq as a percentage of total γδ T cells. **c**, The frequencies of positive cells for selected genes across Vδ1 (n = 1,927), Vδ2 (n = 860) and Vδ3 (n = 506) cells as the percentage of total γδ T cells from each MMR-d colon tumour (n = 5) analysed using scRNA-seq. Vδ3 cells were present in two out of five colon cancers. Data are median ± IQR, with individual samples

(dots). **d**, The frequencies of γδ T cells, CD56⁺ NK cells, CD4⁺ T cells and CD8⁺ T cells in treatment-naive B2M⁺ (n = 12) and B2M⁻ (n = 5) MMR-d colon cancers. Data are median ± IQR, with individual samples (dots). P values were calculated using two-sided Wilcoxon rank-sum tests. **e**, The frequencies of granzyme-B-positive γδ T cells, CD56⁺ NK cells, CD4⁺ T cells and CD8⁺ T cells in treatment-naive B2M⁻ (n = 5) MMR-d colon cancers. CD56⁺ NK cells were present in four out of five B2M⁻ cancer samples. Data are median ± IQR, with individual samples (dots). **f**, Representative images of the detection of tissue-resident (CD103⁺), activated (CD39⁺), cytotoxic (granzyme B⁺), proliferating (Ki-67⁺) and PD-1⁺ γδ T cells (white arrows) by IMC analysis of a treatment-naive MMR-d colon cancer with B2M defects. Scale bar, 20 μm.

## PD-1⁺ Vδ1 and Vδ3 T cells kill HLA-class-I⁻ CRC cells

We next sought to determine whether tumour-infiltrating γδ T cells can recognize and kill CRC cells. We isolated and expanded PD-1⁻ and PD-1⁺ γδ T cells from five MMR-d colon cancers (Extended Data Fig. 4a–c and Supplementary Table 4). Consistent with the scRNA-seq data, expanded PD-1⁺ γδ T cell populations lacked Vδ2⁺ cells and comprised the Vδ1⁺ or Vδ3⁺ subsets, whereas the PD-1⁻ fractions contained Vδ2⁺ or a mixture of Vδ1⁺, Vδ2⁺ and Vδ3⁺ populations (Fig. 3a and Extended Data Fig. 4d). Detailed immunophenotyping of the expanded γδ T cells (Fig. 3a and Extended Data Fig. 5a) showed that all of the subsets expressed the activating receptor NKG2D, whereas the surface expression of natural cytotoxicity receptors (NCRs) and KIRs was most frequent on PD-1⁺ γδ T cells (Vδ1 or Vδ3⁺), consistent with the scRNA-seq results of unexpanded populations.

We measured the reactivity of the expanded γδ T cell populations to HLA-class-I-negative and HLA-class-I-positive cancer cell lines (Fig. 3b and Extended Data Fig. 4b). After co-culture with the different cancer cell lines, reactivity (assessed by expression of activation markers and

secretion of IFNγ) was largely restricted to PD-1⁺ γδ T cells (Vδ1 or Vδ3⁺), whereas activation of PD-1⁻ γδ T cells (Vδ2⁺) was generally not detected (Fig. 3c and Extended Data Fig. 4). PD-1⁺ γδ T cell (Vδ1 or Vδ3⁺) reactivity was variable and was observed against both HLA-class-I-negative and HLA-class-I-positive cell lines (Fig. 3c and Extended Data Fig. 4). To quantify and visualize the differences in the killing of CRC cell lines by PD-1⁺ and PD-1⁻ γδ T cells, we co-cultured the γδ T cell populations with three CRC cell lines (HCT-15, LoVo, HT-29) in the presence of a fluorescent cleaved-caspase-3/7 reporter to measure cancer cell apoptosis over time (Fig. 3d,e). We found pronounced cancer cell apoptosis after co-culture with PD-1⁺ γδ T cells (Vδ1 or Vδ3⁺) compared with PD-1⁻ cells; cancer cell death was more pronounced in HLA-class-I-negative HCT-15 cells (Fig. 3e and Supplementary Videos 1 and 2). Reintroduction of *B2M* in the *B2M*-deficient HCT-15 and LoVo cells diminished their killing by PD-1⁺ γδ T cells (Vδ1 or Vδ3⁺) cells (Extended Data Fig. 6), suggesting that *B2M* loss increases the sensitivity to γδ T cells.

Next, we established two parental patient-derived tumour organoid lines (PDTOs; Supplementary Table 5) of MMR-d CRC and generated isogenic *B2M*^KO lines using CRISPR. Genomic knockout of *B2M*

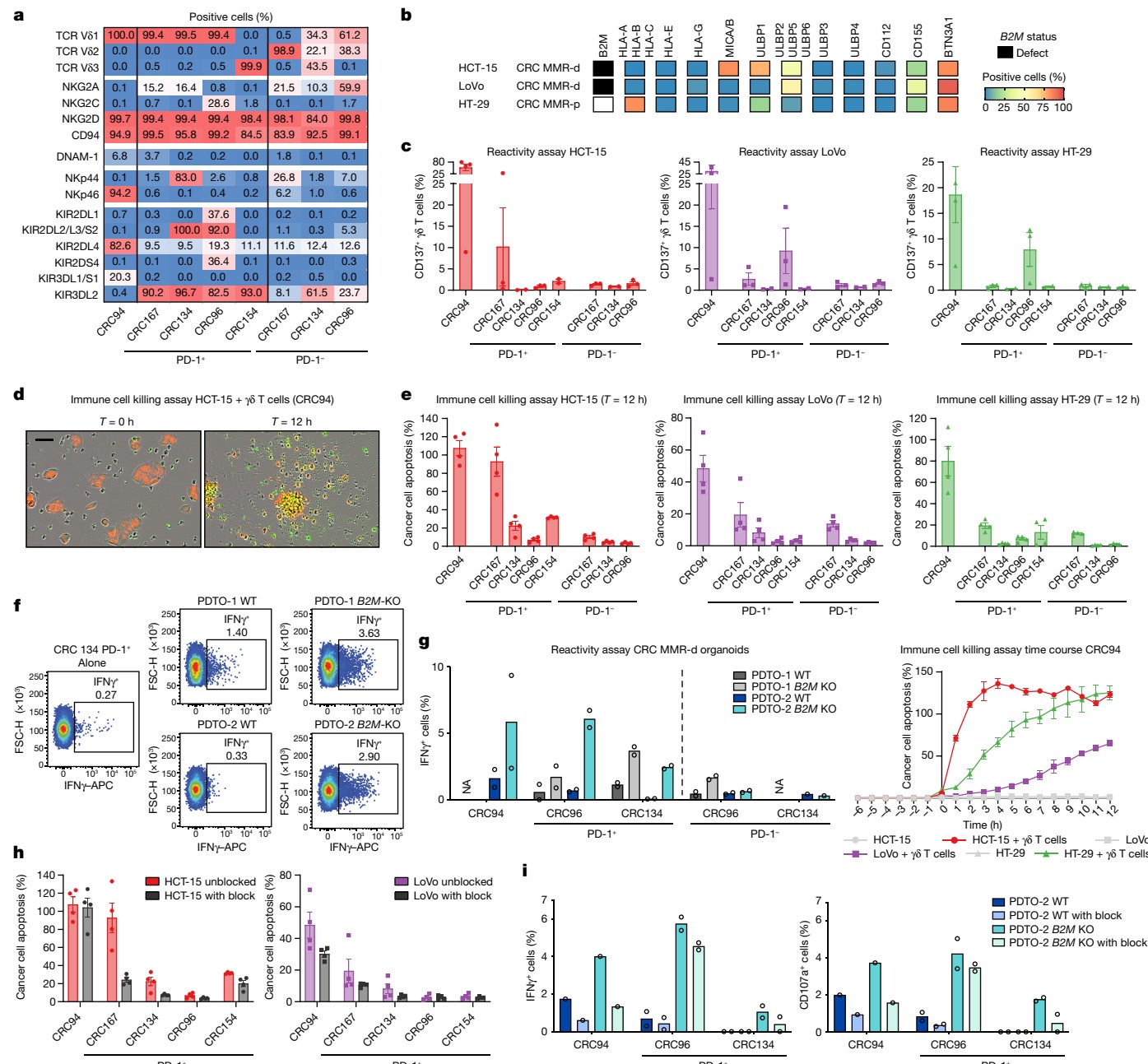

**Fig. 3 | γδ T cells from MMR-d colon cancers show preferential reactivity to HLA-class-I-negative cancer cell lines and organoids. a**, The percentage of positive cells for the indicated markers on expanded γδ T cells from MMR-d colon cancers (n = 5). **b**, Diagram showing the *B2M* status and surface expression of HLA class I, NKG2D ligands, DNAM-1 ligands and butyrophilin on CRC cell lines. MMR-p, MMR proficient. **c**, CD137 expression on γδ T cells after co-culture with CRC cell lines. Data are mean ± s.e.m. from at least two independent experiments. **d**, Representative images showing the killing of NucLightRed-transduced HCT-15 cells by γδ T cells in the presence of a green fluorescent caspase-3/7 reagent. Cancer cell apoptosis is visualized in yellow. Scale bar, 50 μm. **e**, Quantification of the killing of CRC cell lines after co-culture with γδ T cells as described in **d**. Data are mean ± s.e.m. of two wells with two images per well. A representative time course of cancer cell apoptosis is shown

at the bottom right. **f**, Representative flow cytometry plots showing IFNγ expression in γδ T cells unstimulated (alone) and after stimulation with two *B2M^WT* and *B2M^KO* CRC MMR-d organoids. **g**, IFNγ expression in γδ T cells after stimulation with two *B2M^WT* and *B2M^KO* CRC MMR-d organoids, shown as the difference compared with the unstimulated γδ T cell sample. Data are from two biological replicates, except for a single biological replicate of CRC134 PD-1⁻. NA, not available. **h**, The killing of CRC cell lines after 12 h co-culture with γδ T cells with or without NKG2D ligand blocking. Data are mean ± s.e.m. of two wells with two images per well. **i**, IFNγ (left) and CD107a (right) expression in γδ T cells after stimulation with *B2M^WT* PDTO-2 or *B2M^KO* PDTO-2, with or without NKG2D ligand blocking and subtracted background signal. Data are from two biological replicates, except for a single biological replicate of CRC94.

effectively abrogated cell surface expression of HLA class I (Extended Data Fig. 7). We exposed two *B2M^KO* lines and their parental *B2M^WT* lines to the expanded γδ T cell subsets, and quantified γδ T cell activation by determination of IFNγ expression. Similar to our cell line data, γδ T cells displayed increased reactivity to *B2M^KO* PDTOs in comparison

to the *B2M^WT* PDTOs (Fig. 3f,g). Furthermore, γδ T cell reactivity to *B2M^KO* tumour organoids was preferentially contained within the PD-1⁺ population of γδ T cells (Fig. 3g). Thus, a lack of HLA class I antigen presentation in MMR-d tumour cells can be effectively sensed by γδ T cells and stimulates their antitumour response.

Expression of NKG2D on γδ T cells decreased during co-culture with target cells (Extended Data Fig. 8a,b), suggesting the involvement of the NKG2D receptor in γδ T cell activity. The NKG2D ligands MICA/B and ULBPs were expressed by the cancer cell lines (Fig. 3b) and the MMR-d CRC PDTOs, irrespective of their *B2M* status (Extended Data Fig. 7). To examine which receptor–ligand interactions might regulate the activity of PD-1[+] γδ T cells, we performed blocking experiments focused on (1) NKG2D, (2) DNAM-1 and (3) γδ TCR signalling. Of these candidates, the only consistent inhibitory effect was observed for NKG2D ligand blocking on cancer cells, which decreased the activation and killing ability of most PD-1[+] γδ T cells (Fig. 3h and Extended Data Fig. 8c,d), confirming the mechanistic involvement of the NKG2D receptor in γδ T cell activation in this context. Moreover, blocking NKG2D ligands on MMR-d CRC PDTOs reduced the PDTO-directed tumour reactivity of γδ T cells from CRC94 and CRC134 (Fig. 3i). Together, these results show that γδ T cell reactivity to MMR-d tumours is partly dependent on NKG2D/NKG2D-ligand interactions.

## ICB boosts Vδ1 and Vδ3 T cells in *B2M*[MUT] CRC

We subsequently studied how ICB influences γδ T cell infiltration and activation in MMR-d colon cancers in the therapeutic context. For this purpose, we analysed pre- and post-treatment samples of the NICHE trial[9], in which patients with colon cancer were treated with neoadjuvant PD-1 plus CTLA-4 blockade. Consistent with our observations in the DRUP cohort, 4 out of 5 (80%) individuals with *B2M*[MUT] cancers in the NICHE trial showed a complete pathologic clinical response. Immunohistochemical analysis confirmed the loss of B2M protein expression on tumour cells in all mutated cases (Extended Data Fig. 9). Whereas expression of immune marker gene sets in the pretreatment samples was similar between 5 *B2M*[MUT] versus 13 *B2M*[WT] cancers, ICB induced a clear immunological divergence between these two groups (Fig. 4a). The *B2M*[MUT] subgroup was most significantly associated with higher post-treatment expression of *TRDV1* and *TRDV3* (two-sided Wilcoxon rank-sum test, *P* = 0.0067; Fig. 4a), followed by higher expression of the general immune cell marker CD45, NK-cell-related markers, KIRs and αβTCRs (two-sided Wilcoxon rank-sum test, *P* = 0.016, *P* = 0.016, *P* = 0.027 and *P* = 0.043, respectively; Fig. 4a and Extended Data Fig. 10a). The set of KIRs upregulated after ICB in *B2M*[MUT] cancers (Extended Data Fig. 10b) was consistent with the sets of KIRs upregulated in *B2M*[MUT] MMR-d cancers in TCGA (Fig. 1e), and those expressed by MMR-d tumour-infiltrating γδ T cells (Fig. 2c). Pre- and post-ICB gene expression levels related to CD4 and CD8 infiltration were not associated with *B2M* status (Fig. 4a and Extended Data Fig. 10a).

To quantify and investigate the differences in immune profiles after ICB treatment, we used IMC to analyse tissues derived from five *B2M*[MUT] HLA-class-I-negative and five *B2M*[WT] HLA-class-I-positive cancers before and after ICB treatment. In the ICB-naive setting, *B2M*[MUT] MMR-d colon cancers showed higher γδ T cell infiltration compared with *B2M*[WT] MMR-d colon cancers (two-sided Wilcoxon rank-sum test, *P* = 0.032; Fig. 4b and Extended Data Fig. 10c). Importantly, a large proportion of these γδ T cells showed an intraepithelial localization in *B2M*[MUT] MMR-d colon cancers compared with the *B2M*[WT] samples (two-sided Wilcoxon rank-sum test, *P* = 0.0079; Extended Data Fig. 10d). No significant differences were observed in the infiltration of other immune cells, such as NK cells, CD4[+] T cells and CD8[+] T cells, in ICB-naive *B2M*[MUT] versus *B2M*[WT] MMR-d colon cancers (Fig. 4b). ICB treatment resulted in major pathologic clinical responses, and residual cancer cells were absent in most post-ICB samples. All post-ICB tissues showed a profound infiltration of different types of immune cells (Extended Data Fig. 10e), of which γδ T cells were the only immune subset that was significantly higher in ICB-treated *B2M*[MUT] compared with *B2M*[WT] MMR-d colon cancers (two-sided Wilcoxon rank-sum test, *P* = 0.016; Fig. 4b and Extended Data Fig. 10c). In the sole *B2M*[MUT] case that still contained cancer cells after treatment with ICB, the majority of granzyme B[+]

immune cells infiltrating the tumour epithelium were γδ T cells (Fig. 4c). These γδ T cells displayed co-expression of CD103, CD39, Ki-67 and PD-1 (Extended Data Fig. 10f–h). Taken together, these results show that ICB treatment of MMR-d colon cancer increases the presence of activated, cytotoxic and proliferating γδ T cells at the tumour site, especially when these cancers are B2M-deficient, highlighting γδ T cells as effectors of ICB treatment within this context.

## Discussion

CD8[+] αβ T cells are major effectors of ICB[11,12,27] and rely on HLA class I antigen presentation of target cells. We confirm and shed light on the paradox that patients with HLA class I defects in MMR-d cancers retain the clinical benefit of ICB, suggesting that other immune effector cells are involved in compensating for the lack of conventional CD8[+] T cell immunity in this setting. We show that genomic inactivation of *B2M* in MMR-d colon cancers was associated with: (1) an elevated frequency of activated γδ T cells in ICB-naive tumours; (2) an increased presence of tumour-infiltrating γδ T cells after ICB treatment; (3) in vitro activation of tumour-infiltrating γδ T cells by CRC cell lines and PDTOs; and (4) killing of tumour cell lines by γδ T cells, in particular by Vδ1 and Vδ3 subsets expressing PD-1.

Different subsets of γδ T cells exhibit substantially diverse functions that, in the context of cancer, range from tumour-promoting to tumour-icidal effects[20,28,29]. Thus, it is of interest to determine what defines antitumour reactivity of γδ T cells. Here we isolated Vδ1/3-expressing PD-1[+] T cells as well as Vδ2-expressing PD-1[−] T cells from MMR-d tumour tissues. Our data suggest that especially tumour-infiltrating Vδ1 and Vδ3 T cells can recognize and kill HLA-class-I-negative MMR-d tumours, whereas Vγ9Vδ2 cells, the most studied and main subset of γδ T cells in the blood, appear to be less relevant within this context. This is consistent with other studies showing that the cytotoxic ability of Vδ1 cells generally outperforms their Vδ2 counterparts[30–34]. Notably, reports of the cytotoxicity of tumour-infiltrating Vδ3 cells have been lacking. Furthermore, the observation that PD-1[+] γδ T cells (Vδ1 and Vδ3 phenotype) demonstrated clearly higher levels of antitumour reactivity compared with their PD-1[−] counterparts (Vδ2 phenotype) suggests that, as for CD8[+] αβ T cells[35], PD-1 expression may be a marker of antitumour reactivity in γδ T cells.

The mechanisms of activation of γδ T cells are notoriously complex and diverse[20]. Specifically, for Vδ1[+] cells, NKG2D has been described to be involved in tumour recognition, which is dependent on tumour cell expression of NKG2D ligands MICA/B and ULBPs[36–38]. Here, MICA/B and ULBPs were highly expressed by the MMR-d CRC cell lines and tumour organoids, and blocking these ligands reduced γδ T cell activation and cytotoxicity. This suggests a role for the activating receptor NKG2D in γδ T cell immunity to MMR-d tumours. Future research should address the outstanding question of how γδ T cells accumulate in B2M-deficient tumours, and whether the lack of CD8[+] T cell activity might contribute to the establishment of an attractive niche for γδ T cells and other immune effector cells. Potential mechanisms for the recognition of HLA-class-I-negative phenotypes may include KIR-, NKG2A- and LILRB1-mediated interactions with target cancer cells. Notably, we found that the expression of KIRs was most pronounced on PD-1[+] γδ T cells (Vδ1 or Vδ3[+] subsets), which demonstrated anti-tumour activity. Whether the lack of KIR-mediated signalling promotes the survival of γδ T cells and their intratumoural proliferation remains to be studied.

Our findings have broad implications for cancer immunotherapy. First, our findings strengthen the rationale for combining PD-1 blockade with immunotherapeutic approaches to further enhance γδ T-cell-based antitumour immunity. Second, the presence or absence in tumours of specific γδ T cell subsets (such as Vδ1 or Vδ3) may help to define patients who are responsive or unresponsive to ICB, respectively, especially in the case of MMR-d cancers and

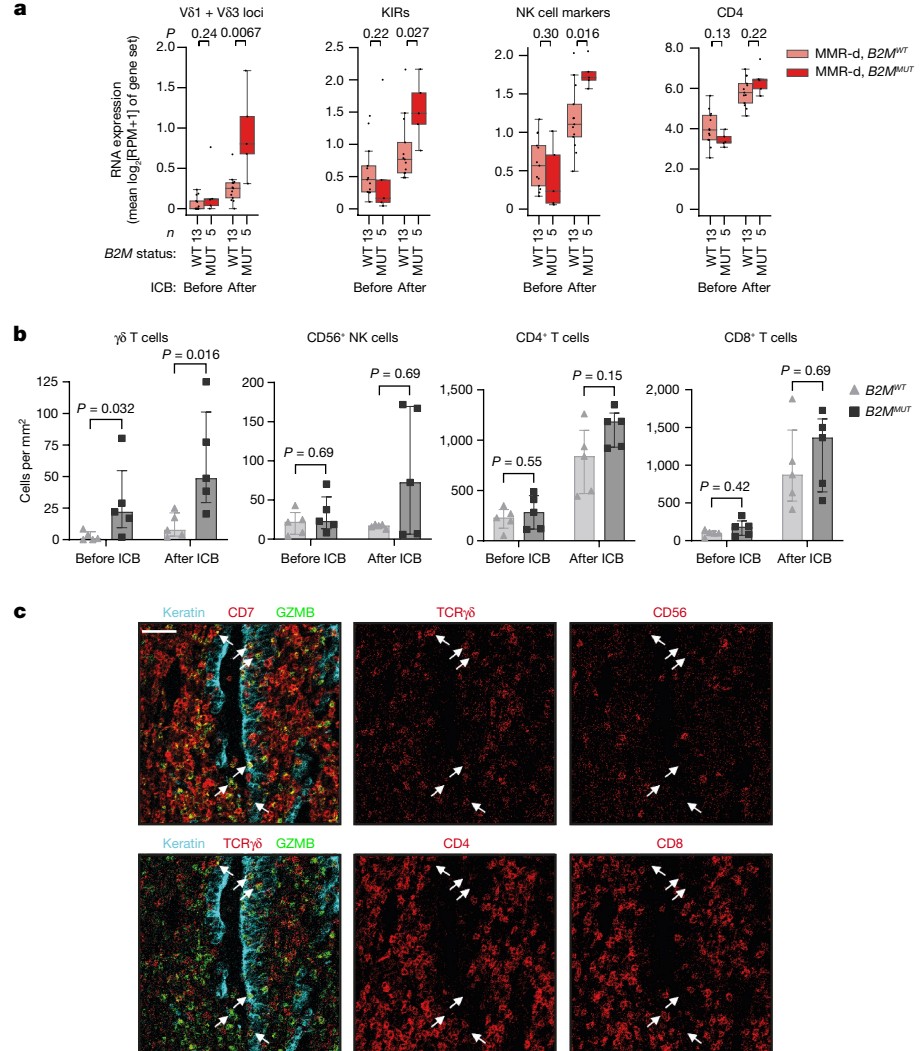

**Fig. 4 | ICB induces substantial infiltration of γδ T cells into MMR-d colon cancers with defects in antigen presentation. a**, The RNA expression of different immune marker gene sets in MMR-d $B2M^{WT}$ (pink) and MMR-d $B2M^{MUT}$ (red) cancers before (left) and after (right) neoadjuvant ICB in the NICHE study. The boxes, whiskers and dots indicate quartiles, 1.5 × IQR and individual data points, respectively. *P* values were calculated using two-sided Wilcoxon rank-sum tests comparing MMR-d $B2M^{WT}$ versus MMR-d $B2M^{MUT}$ cancers. **b**, The frequencies of γδ T cells, CD56[+] NK cells, CD4[+] T cells and CD8[+] T cells in $B2M^{WT}$ ($n = 5$) and $B2M^{MUT}$ ($n = 5$) MMR-d colon cancers before and after ICB treatment. Data are median ± IQR, with individual samples (dots). *P* values were calculated using two-sided Wilcoxon rank-sum tests. **c**, Representative images of granzyme-B-positive γδ T cells infiltrating the tumour epithelium (white arrows) by IMC analysis of a $B2M^{MUT}$ MMR-d colon cancer after ICB treatment. Scale bar, 50 μm.

other malignancies with frequent HLA class I defects, such as stomach adenocarcinoma[39] and Hodgkin's lymphoma[40]. Third, our results suggest that MMR-d cancers and other tumours with HLA class I defects may be particularly attractive targets for Vδ1 or Vδ3 T-cell-based cellular therapies.

Although we have provided detailed and multidimensional analyses, it is probable that γδ T cells are not the only factor driving ICB responses in HLA-class-I-negative MMR-d CRC tumours. In this context, other HLA-class-I-independent immune subsets, such as NK cells and neoantigen-specific CD4[+] T cells may also contribute. The latter were shown to have an important role in the response to ICB (as reported in mouse *B2M*-deficient MMR-d cancer models[41]), and may also support γδ T-cell-driven responses. Notably, no subset equivalent to Vδ1 or Vδ3 T cells has been identified in mice, which complicates their investigation in vivo models. In conclusion, our results provide strong evidence that γδ T cells are cytotoxic effector cells of ICB treatment in HLA-class-I-negative MMR-d colon cancers, with implications for further exploitation of γδ T cells in cancer immunotherapy.

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

# Methods

## TCGA data

RNA expression data (raw counts) of the colon adenocarcinoma (COAD), stomach adenocarcinoma (STAD) and Uterus Corpus Endometrium Carcinoma (UCEC) cohorts of The Cancer Genome Atlas (TCGA) Research Network were downloaded through the GDC data portal (https://portal.gdc.cancer.gov) on 10 April 2019. Of these cohorts, mutation, copy number, purity and ploidy data were downloaded from the GDC on 11 November 2021, as the controlled access ABSOLUTE-annotated[42] MAF file (mutations), SNP6 white-listed copy number segments file (copy numbers) and ABSOLUTE purity/ploidy file of the TCGA PanCanAtlas project[43]. Mismatch-repair-deficiency status was obtained from ref. [44] (TCGA subtype = GI.HM-indel or UCEC.MSI).

## DRUP data

A detailed description of the DRUP, including details on patient accrual, study design, oversight and end points was published previously[23]. In brief, the DRUP is a national, non-randomized multidrug and multitumour study in the Netherlands, in which patients receive off-label drugs registered for other treatment indications. These patients had advanced or metastatic solid tumours and had exhausted standard-treatment options, they were required to be at least 18 years of age, with acceptable organ function and performance status (Eastern Cooperative Oncology Group (ECOG) score ≤2), and have an objectively evaluable disease of which a fresh baseline tumour biopsy could safely be obtained. We analysed 71 patients with MMR-d cancers recruited and treated with PD-1 blockade in 22 Dutch hospitals participating in the DRUP[23] between 2016 and 2021. Patients included in this analysis had (1) a clinical follow-up ≥16 weeks after start of PD-1 blockade treatment; (2) WGS data passing standard quality controls (as defined previously, including a sequencing-based tumour purity of ≥20%)[25]; and (3) available RNA-seq data (Supplementary Table 1). MMR-d status was determined using routine diagnostics at the hospital of patient accrual and was confirmed by WGS, on the basis of an MSIseq (v.1.0.0)[45] score of >4, which represents a predefined threshold[25]. Consistent with the study protocol of the DRUP[23], the primary outcome measure for our analysis was clinical benefit, defined as disease control of ≥16 weeks, and the secondary outcome measure was best overall response, all assessed according to the RECIST 1.1 guidelines by the local treatment team at the site of accrual. As determined in the study protocol, these outcome measures were considered to be evaluable in patients who received at least two cycles of intravenous study medication, and for whom the response was radiologically or clinically evaluable (at the treating physician's discretion). For genomics and transcriptomics analyses, fresh frozen tumour biopsies were obtained at the baseline (that is, before PD-1 blockade). WGS analysis (median depths, ~100× and ~40× for tumour and normal, respectively) and bioinformatics analyses were performed as previously described[23,25], with an optimized pipeline based on open-source tools that is freely available at GitHub (https://github.com/hartwigmedical/pipeline5). The TMB per Mb was determined by counting the genome-wide number of mutations (SNVs, MNVs and indels) and dividing this number by the number of megabases sequenced. For RNA-seq analysis, we extracted total RNA using the QIAGEN QIAsymphony RNA kit (931636). Samples with approximately 100 ng total RNA were prepared using KAPA RNA Hyper + RiboErase HMR (8098131702) and the RNA libraries were paired-end sequenced on the Illumina NextSeq550 platform (2 × 75 bp) or the Illumina NovaSeq6000 platform (2 × 150 bp). Raw RNA reads (FASTA files) were aligned to the human reference genome (GRCh38) using STAR[46], (v.2.7.7a) using the default settings in two-pass mode.

## Hartwig data

We analysed 2,256 metastatic tumours included in the freely available Hartwig database[25] that (1) were MMR-p (WGS-based MSIseq[45] score ≤4);

(2) had available WGS data passing standard quality controls (as defined before, including a sequencing-based tumour purity ≥20%)[25]; (3) and had available RNA-seq data. We excluded 89 tumours from rare primary tumour locations, defined as locations with less that <20 patients in our selection. When individual patients had data available of biopsies obtained at different timepoints, we included data of only the first biopsy. Sequencing and bioinformatics were performed identically to the procedures used for the generation of the DRUP dataset (see above). Details of this cohort are provided in Supplementary Table 3.

## NICHE study data

Raw RNA reads (FASTA files) of our recently published NICHE study[9] (ClinicalTrials.gov: NCT03026140) were generated as described in the original publication and aligned to the human reference genome (GRCh38) with STAR[46] (v.2.7.7a) using the default settings in two-pass mode. For gene expression quantification, we used the gencode.v35. annotation.gtf annotation file. Somatic mutation data were obtained from DNA-seq of pretreatment tumour biopsies and matched germline DNA, as described in the original publication[9].

## *B2M* status

Consistent with the notion that both biallelic and monoallelic non-synonymous *B2M* mutations are strongly associated with tumour-specific loss of B2M protein expression[18], we considered all tumours with at least one somatic, non-synonymous *B2M* mutation to be a *B2M* mutant. As none of the *B2M* mutant tumours in TCGA had *B2M* copy number gains or losses, LOH of *B2M* could easily be assessed by a simple calculation estimating the mutation's copy number:

$$\text{Mut}_{CN} = \text{round}\left(2 \times \frac{\text{VAF}}{\text{purity}}\right)$$

where $\text{Mut}_{CN}$ represents the estimated mutation's copy number (rounded to an integer value), VAF represents the variant allele frequency of the mutation and purity equals the ABSOLUTE-based[42] tumour cell fraction of the sample.

A $\text{Mut}_{CN}$ equal to 2 was considered to be consistent with LOH, as the most parsimonious explanation of such a result is the scenario in which all tumour-derived reads spanning the region of the *B2M* mutation contain the mutation and none of the tumour-derived reads are WT.

In analyses of patients in the DRUP and Hartwig datasets, LOH of *B2M* mutations was determined as an integrated functionality of PURPLE (v.2.34)[47]. When multiple *B2M* mutations were present within a sample, we manually phased the mutations through inspection of the *B2M*-aligned reads using the Integrative Genomics Viewer (IGV)[48]. Here mutations were phased in case single reads were observed spanning the genomic locations of both mutations. We divided patients with multiple *B2M* mutations into three subgroups: (1) Biallelic, if (a) the multiple mutations were in *trans* AND the integer sum of the mutation copy numbers equalled (or exceeded) the integer copy number of the *B2M* gene (for mutations in *cis*, only one of these mutations was considered in the calculation); or (b) at least one of the mutations showed LOH. (2) Potentially biallelic, if the multiple mutations affected genomic locations too distant to be phased and the (integer) sum of the mutation copy numbers equalled (or exceeded) the integer copy number of the *B2M* gene (for mutations in *cis*, only one of these mutations was considered in the calculation) and none of the mutations showed LOH. (3) Not bi-allelic, if the integer sum of the mutation copy numbers was smaller than the integer copy number of the *B2M* gene (for mutations in *cis*, only one of these mutations was considered in the calculation) and none of the mutations showed LOH.

In these analyses, mutations were considered to be subclonal in the case in which the probability of subclonality was >0.5 (the situation in which a mutation is more likely subclonal than clonal), as determined using PURPLE (v.2.34)[47].

## Association of *B2M* status with outcome and tumour characteristics

To test whether somatic *B2M* alterations were associated with the clinical benefit rate of patients with MMR-d tumours treated with ICB in the DRUP, we used a Fisher's exact test (using the Python package Scipy[49] (v.1.3.1)) for unadjusted analyses and logistic regression (as implemented by the Python package Statsmodels (https://pypi.org/project/statsmodels/; v.0.10.1) for analyses adjusted for the continuous TMB per Mb and/or the primary site of the tumour. The association of *B2M* status with TMB was tested using Scipy's Wilcoxon rank-sum test. Associations of *B2M* status with the primary site of the tumour or the biopsy location were tested using Scipy's Fisher's exact test.

## Association of TMB with ICB treatment outcome

For the DRUP cohort, the association of clinical benefit with TMB was tested using Statsmodels' Wilcoxon rank-sum test.

## Differential gene expression analysis

Differential RNA expression of genes was tested in R using EdgeR[50] (v.3.28.1) and Limma[51] (including Voom[52]) (v.3.42.2). Raw read counts were filtered by removing low-expressed genes. Normalization factors were calculated using EdgeR to transform the raw counts to $\log_2$[counts per million reads (CPM)] and calculate residuals using Voom. Voom was then used to fit a smoothened curve to the √(residual standard deviation) by average gene expression, which was then plotted for visual inspection to confirm that the appropriate threshold was used for filtering of low-expressed genes (defined as the minimal amount of filtering necessary to overcome a dipping mean-variance trend at low counts). Next, Limma was used to calculate the differential expression of genes on the basis of a linear model fit, considering the smoothened curve for sample weights, and empirical Bayes smoothing of standard errors. FDR-adjusted *P* values were calculated using Benjamini–Hochberg correction of the obtained *P* values.

**TCGA.** Using TCGA data, we calculated the differential expression between tumours with and without mutations in *B2M*, adjusting for tumour type, using the following design formula: expression ∝ Primary_Site + B2M_status (+ intercept by default), for which Primary_Site was a three-levelled factor (COAD, STAD, or UCEC) and B2M_status was a two-levelled factor (mutated, or wildtype).

**NICHE study.** Using NICHE study data, we calculated differential expression between pre- and post-ICB treatment. To respect the paired nature of these data, we used the following design formula: expression ∝ Patient + ICB + intercept, where Patient is a factor for each individual patient and ICB is a two-levelled factor (ICB-treated, yes/no).

## Immune marker gene set expression analysis

To use RNA-seq data to obtain a relative estimate of the infiltration of specific immune cell types within tumours, we calculated the average $\log_2$[RPM + 1] expression of marker genes that are specifically expressed in the immune cell types of interest. To this end, we used the previously published marker gene sets[24], and extended this by (1) *CD4* as a CD4$^+$ T cell marker gene; (2) *TRDV1* and *TRDV3* as γδ1/3T cell marker genes; and (3) a killer-cell Ig-like receptor (KIR) gene set (comprising all genes of which the name starts with KIR and of which the name contains DL or DS). We excluded the gene set 'NK CD56$^{dim}$ cells' of ref. [24] (comprising *IL21R*, *KIR2DL3*, *KIR3DL1* and *KIR3DL2*) from our analyses, as three out of four genes within this set were KIRs and this set therefore showed high collinearity/redundancy to our full KIR gene set. As *XLC1* and *XLC2* are highly expressed by tumour-infiltrating γδ T cells, these genes were removed from the NK cell marker gene set and replaced by *KLRF1*, which encodes the well-established NK cell marker NKp80. The resulting gene set consisted of *NCR1* and *KLRF1*, encoding the well-established NK

cell markers NKp46 and NKp80, respectively. Finally, we reduced the 'cytotoxic cells' marker gene set of ref. [24] to those genes in the set encoding cytotoxic molecules (*GZMA*, *GZMB*, *GZMH*, *PRF1*, *GNLY*, *CTSW*). A list of the final collection of our marker gene sets is provided in Supplementary Table 2.

Association of immune cell marker gene set expression with *B2M* alteration status (alteration yes/no) was calculated as follows:
1. For TCGA-study-based analyses, we used (i) the Wilcoxon rank-sum test (for unadjusted analyses) and (ii) ordinary least squares linear regression (for analyses adjusted for tumour type; as implemented by the Python package 'Statsmodels), using a similar design formula as for the differential gene expression analysis.
2. For DRUP-cohort-based analyses, we used a linear mixed effects model (as implemented by the lmer function of the R package Lme4 (v.1.1.26)), adjusting for tumour type and biopsy site as random effects, using the following design formula: expression ∝ B2M_status + (1|tumour_type) + (1|biopsy_site) + intercept. In subgroup analysis of CRC, we omitted tumour type in this formula: expression ∝ B2M_status + (1|biopsy_site) + intercept.
3. For Hartwig data-based analyses, we used ordinary least squares linear regression (as implemented by the Python package Statsmodels) adjusting for tumour type, using the following design formula: expression ∝ Primary_Site + B2M_status + intercept.
4. For NICHE-study-based analyses, we used the Wilcoxon rank-sum test (as implemented by the Python package Scipy).

## Hierarchical clustering

Hierarchical clustering of expression profiles of individual genes or immune marker gene sets of TCGA cohorts was performed on *Z*-score-transformed $\log_2$[RPM + 1] expression values, using the Python package Scipy[49], with Euclidean distance as the distance metric and using the Ward variance minimization algorithm. Here, we used the default settings with one exception—for visualization purposes, the colour threshold was halved in the TCGA-based clustering of individual genes.

## Patient samples

The DRUP study and the generation of the Hartwig database were initiated and conducted on behalf of the Center for Personalized Cancer Treatment (CPCT; ClinicalTrials.gov: NCT02925234, NCT01855477). These studies were approved by the Medical Ethical Committee of the Netherlands Cancer Institute in Amsterdam and the University Medical Center Utrecht, respectively, and were conducted in accordance with good clinical practice guidelines and the Declaration of Helsinki's ethical principles for medical research. Written informed consent was obtained from all of the study participants. Moreover, primary colon cancer tissues from a total of 17 patients with colon cancer who underwent surgical resection of their tumour at the Leiden University Medical Center (LUMC, The Netherlands; Supplementary Table 4) were used for scRNA-seq, IMC and functional assays. No patient with a previous history of inflammatory bowel disease was included. This study was approved by the Medical Ethical Committee of the Leiden University Medical Center (protocol P15.282), and patients provided written informed consent. Finally, primary colon cancer tissues from ten patients with colon cancer included in the NICHE study (NCT03026140)[9] carried out at the Netherlands Cancer Institute (NKI, The Netherlands) were used for this study. All samples were anonymized and handled according to the ethical guidelines described in the Code for Proper Secondary Use of Human Tissue in the Netherlands of the Dutch Federation of Medical Scientific Societies.

## Processing of colon cancer tissues

Details on the processing of colon cancer tissues have been described previously[22]. In brief, macroscopic sectioning from the lumen to the most invasive area of the tumour was performed. Tissues were

collected in IMDM + GlutaMax medium (Gibco) complemented with 20% fetal calf serum (FCS) (Sigma-Aldrich), 1% penicillin–streptomycin (Gibco) and fungizone (Gibco), and 0.1% ciprofloxacin (provided by apothecary LUMC) and gentamicin (Invitrogen), and immediately cut into small fragments in a Petri dish. Enzymatic digestion was performed using 1 mg ml$^{-1}$ collagenase D (Roche Diagnostics) and 50 µg ml$^{-1}$ DNase I (Roche Diagnostics) in 5 ml of IMDM + GlutaMax medium for 30 min at 37 °C in gentleMACS C tubes (Miltenyi Biotec). During and after incubation, cell suspensions were dissociated mechanically on the gentleMACS Dissociator (Miltenyi Biotec). Cell suspensions were filtered through a 70 µm cell strainer (Corning), washed in IMDM + GlutaMax medium with 20% FCS, 1% penicillin–streptomycin and 0.1% fungizone, and the cell count and viability were determined using the Muse Count & Viability Kit (Merck) on the Muse Cell Analyser (Merck). On the basis of the number of viable cells, cells in IMDM + GlutaMax medium were cryopreserved in liquid nitrogen until time of analysis complemented 1:1 with 80% FCS and 20% dimethyl sulfoxide (Merck).

### Immunohistochemical detection of MMR, B2M and HLA class I proteins

For the tumour tissue samples from the LUMC, tumour MMR status was determined by immunohistochemical detection of PMS2 (anti-PMS2 antibodies; EP51, DAKO) and MSH6 (anti-MSH6 antibodies; EPR3945, Abcam) proteins[53]. MMR-deficiency was defined as the lack of expression of at least one of the MMR-proteins in the presence of an internal positive control. Tumour B2M status was determined by immunohistochemical detection of B2M (anti-B2M antibodies; EP2978Y, Abcam). Immunohistochemical detection of HLA class I expression on tumour cells was performed with HCA2 and HC10 monoclonal antibodies (Nordic-MUbio), and tumours were classified as HLA class I positive, weak or loss, as described previously[16]. For the tumour samples from the NICHE study, immunohistochemistry analysis of the formalin-fixed paraffin-embedded (FFPE) tissue was performed on the BenchMark Ultra autostainer (Ventana Medical Systems). In brief, paraffin sections were cut at 3 µm, heated at 75 °C for 28 min and deparaffinized in the instrument using EZ prep solution (Ventana Medical Systems). Heat-induced antigen retrieval was performed using Cell Conditioning 1 (CC1, Ventana Medical Systems) for 32 min at 95 °C (HC10) or 64 min at 95 °C (B2M and HCA2). HLA class I heavy chain expression was detected using clone HCA2 (1:5,000, 60 min at room temperature; Nordic-Mubio) and clone HC10 (1:20,000, 32 min at 37 °C; Nordic-Mubio). B2M was detected using clone D8P1H (1:1,500, 60 min at room temperature; Cell Signaling). Bound antibodies were detected using the OptiView DAB Detection Kit (Ventana Medical Systems). Slides were counterstained with haematoxylin and Bluing Reagent (Ventana Medical Systems). A PANNORAMIC 1000 scanner from 3DHISTECH was used to scan the slides at ×40 magnification.

### Sorting of γδ T cells from colon cancers and scRNA-seq

scRNA-seq was performed on sorted γδ T cells from colon cancers (MMR-d) of five patients from the LUMC in the presence of hashtag oligos (HTOs) for sample ID and antibody-derived tags (ADTs) for CD45RA and CD45RO protein expression by CITE-seq[54]. Cells were thawed, rested at 37 °C in IMDM (Lonza)/20% FCS for 1 h, and then incubated with human Fc receptor block (BioLegend) for 10 min at 4 °C. Cells were then stained with cell surface antibodies (anti-CD3-PE (1:50, SK7, BD Biosciences), anti-CD45-PerCP-Cy5.5 (1:160, 2D1, eBioscience), anti-CD7-APC (1:200, 124-1D1, eBioscience), anti-EPCAM-FITC (1:60, HEA-125, Miltenyi), anti-TCRγδ-BV421 (1:80, 11F2, BD Biosciences); and a 1:1,000 near-infrared viability dye (Life Technologies)), 1 µg of TotalSeq-C anti-CD45RA (HI100, BioLegend, 2 µl per sample) and 1 µg of TotalSeq-C anti-CD45RO (UCHL1, BioLegend, 2 µl per sample) antibodies, and 0.5 µg of a unique TotalSeq-C CD298/B2M hashtag antibody (LNH-94/2M2, BioLegend, 1 µl per sample) for 30 min at 4 °C. Cells were

washed three times in FACS buffer (PBS (Fresenius Kabi)/1% FCS) and kept under cold and dark conditions until cell sorting. Compensation was performed using CompBeads (BD Biosciences) and ArC reactive beads (Life Technologies). Single, live CD45$^+$EPCAM$^-$CD3$^+$TCRγδ$^+$ cells were sorted on the FACS Aria III 4L (BD Biosciences) system. After sorting, the samples were pooled.

scRNA-seq libraries were prepared using the Chromium Single Cell 5′ Reagent Kit v1 chemistry (10x Genomics) according to the manufacturer's instructions. The construction of 5′ gene expression libraries enabled the identification of γδ T cell subsets according to Vδ and Vγ usage. Libraries were sequenced on the HiSeq X Ten using paired-end 2 × 150 bp sequencing (Illumina). Reads were aligned to the human reference genome (GRCh38) and quantified using Cell Ranger (v.3.1.0). Downstream analysis was performed using Seurat (v.3.1.5) according to the author's instructions[55]. In brief, cells that had less than 200 detected genes and genes that were expressed in less than six cells were excluded. The resulting 5,669 cells were demultiplexed on the basis of HTO enrichment using the MULTIseqDemux algorithm[56]. Next, cells with a mitochondrial gene content of greater than 10% and cells with outlying numbers of expressed genes (>3,000) were filtered out from the analysis, resulting in a final dataset of 4,442 cells, derived from HTO1 (n = 332), HTO6 (n = 105), HTO7 (n = 1,100), HTO8 (n = 1,842) and HTO9 (n = 1,063). Data were normalized using the Log-Normalize function of Seurat with a scale factor of 10,000. Variable features were identified using the FindVariableFeatures function of Seurat returning 2,000 features. We next applied the RunFastMNN function of SeuratWrappers split by sample ID to adjust for potential batch-derived effects across the samples[57]. Uniform manifold approximation and projection (UMAP)[58] was used to visualize the cells in a two-dimensional space, followed by the FindNeighbors and FindClusters functions of Seurat. Data were scaled, and heterogeneity associated with mitochondrial contamination was regressed out. Cell clusters were identified by performing differentially expressed gene analysis using the FindAllMarkers function, with min.pct and logfc.threshold at 0.25. The number of $TRDV1^+$ (Vδ1, n = 1,927), $TRDV2^+$ (Vδ2, n = 860) or $TRDV3^+$ (Vδ3, n = 506) cells was determined as the percentage of all cells with an expression level of >1, with <1 for the other TCR Vδ chains. CRC96, 134 and 167 had less than ten $TRDV3^+$ cells, and were not included in the Vδ3 analysis. Transcripts of Vδ4 (TRDV4), Vδ5 (TRDV5) and Vδ8 (TRDV8) cells were not detected. The percentage of cells positive for a certain gene was determined as all cells with an expression level of >1.

### IMC staining and analysis

IMC analysis was performed on ICB-naive colon cancer tissues (MMR-d) of 17 patients from the LUMC; 5 of these colon cancer tissues had B2M defects and the remainder were B2M-positive (Supplementary Table 4). Moreover, IMC was performed on ICB-naive and ICB-treated colon cancer tissues (MMR-d) of ten patients from the NICHE study; five of these colon cancer tissues were $B2M^{WT}$ and five were $B2M^{MUT}$. Antibody conjugation and immunodetection were performed according to previously published methodology[59]. FFPE tissue (thickness, 4 µm) was incubated with 41 antibodies in four steps. First, sections were incubated overnight at room temperature with anti-CD4 and anti-TCRδ antibodies, which were subsequently detected using metal-conjugated secondary antibodies (1 µg ml$^{-1}$, donkey anti-rabbit IgG and goat anti-mouse IgG, respectively; Abcam). Second, the sections were incubated with 20 antibodies (Supplementary Table 6) for five hours at room temperature. Third, the sections were incubated overnight at 4 °C with the remaining 19 antibodies (Supplementary Table 6). Fourth, the sections were incubated with 0.125 µM Cell-ID intercalator-Ir (Fluidigm) to detect the DNA, and stored dry until measurement. For each sample, six 1,000 µm × 1,000 µm regions (two to three for pretreatment NICHE biopsies due to the small tissue size) were selected on the basis of consecutive haematoxylin and eosin

stains and ablated using the Hyperion Imaging system (Fluidigm). Data were acquired using the CyTOF Software (v.7.0) and exported using MCD Viewer (v.1.0.5). Data were normalized using semi-automated background removal in ilastik[60] (v.1.3.3), to control for variations in the signal-to-noise ratio between FFPE sections as described previously[61]. Next, the phenotype data were normalized at the pixel level. Cell segmentation masks were created for all cells in ilastik and CellProfiler[62] (v.2.2.0). In ImaCytE[63] (v.1.1.4), cell segmentation masks and normalized images were combined to generate single-cell FCS files containing the relative frequency of positive pixels for each marker per cell. Cells forming visual neighbourhoods in a *t*-distributed stochastic neighbour embedding[64] in Cytosplore[65] (v.2.3.0) were grouped and exported as separate FCS files. The resulting subsets were imported back into ImaCyte and visualized on the segmentation masks. Expression of immunomodulatory markers was determined as all cells with a relative frequency of at least 0.2 positive pixels per cell. Differences in cells per mm$^2$ were calculated using Mann–Whitney tests in Graph-Pad Prism (v.9.0.1). Image acquisition and analysis were performed blinded to group allocation.

## Sorting of γδ T cells from colon cancers and cell culturing

γδ T cells from colon cancers (MMR-d) of five patients from the LUMC were sorted for cell culture. Cells were thawed and rested at 37 °C in IMDM (Lonza)/10% nHS for 1 h. Next, cells were incubated with human Fc receptor block (BioLegend) and stained with cell surface antibodies (anti-CD3-Am Cyan (1:20, SK7, BD Biosciences), anti-TCRγδ-BV421 (1:80, 11F2, BD Biosciences) and anti-PD-1-PE (1:30, MIH4, eBioscience)) for 45 min at 4 °C together with different additional antibodies for immunophenotyping (anti-CD103-FITC (1:10, Ber-ACT8, BD Biosciences), anti-CD38-PE-Cy7 (1:200, HIT2, eBioscience); anti-CD39-APC (1:60, A1, BioLegend), anti-CD45RA-PE-Dazzle594 (1:20, HI100, Sony), anti-CD45RO-PerCP-Cy5.5 (1:20, UCHL1, Sony), anti-TCRαβ-PE-Cy7 (1:40, IP26, BioLegend), anti-TCRVδ1-FITC (1:50, TS8.2, Invitrogen) or anti-TCRVδ2-PerCP-Cy5.5 (1:200' B6, BioLegend). A 1:1,000 live/dead fixable near-infrared viability dye (Life Technologies) was included in each staining. Cells were washed three times in FACS buffer (PBS/1% FCS) and kept under cold and dark conditions until cell sorting. Compensation was performed using CompBeads (BD Biosciences) and ArC reactive beads (Life Technologies). Single, live CD3$^+$TCRγδ$^+$PD-1$^+$ and PD-1$^-$ cells were sorted on the FACS Aria III 4L (BD Biosciences) system. For CRC94, all γδ T cells were sorted owing to the low number of PD-1$^+$ cells. γδ T cells were sorted in medium containing feeder cells (1 × 10$^6$ per ml), PHA (1 μg ml$^{-1}$; Thermo Fisher Scientific), IL-2 (1,000 IU ml$^{-1}$; Novartis), IL-15 (10 ng ml$^{-1}$; R&D Systems), gentamicin (50 μg ml$^{-1}$) and fungizone (0.5 μg ml$^{-1}$). Sorted γδ T cells were expanded in the presence of 1,000 IU ml$^{-1}$ IL-2 and 10 ng ml$^{-1}$ IL-15 for 3–4 weeks. The purity and phenotype of γδ T cells were assessed using flow cytometry. We obtained a >170,000-fold increase in 3–4 weeks of expansion of γδ T cells (Extended Data Fig. 4c).

## Immunophenotyping of expanded γδ T cells by flow cytometry

Expanded γδ T cells from colon tumours were analysed by flow cytometry for the expression of TCR Vδ chains, NKG2 receptors, NCRs, KIRs, tissue-residency/activation markers, cytotoxic molecules, immune checkpoint molecules, cytokine receptors and Fc receptors. In brief, cells were incubated with human Fc receptor block (BioLegend) and stained with cell surface antibodies (Supplementary Table 7) for 45 min at 4 °C, followed by three wash steps in FACS buffer (PBS/1% FCS). Granzyme B and perforin were detected intracellularly using fixation buffer and intracellular staining permeabilization wash buffer (BioLegend). Compensation was performed using CompBeads (BD Biosciences) and ArC reactive beads (Life Technologies). Cells were acquired on the FACS LSR Fortessa 4L (BD Biosciences) system running FACSDiva software (v.9.0; BD Biosciences). Data were analysed using FlowJo (v.10.6.1; Tree Star).

## Cancer cell line models and culture

Human colorectal adenocarcinoma cell lines HCT-15 (MMR-d), LoVo (MMR-d), HT-29 (MMR-p), SW403 (MMR-p) and SK-CO-1 (MMR-p), as well as HLA-class-I-deficient human leukaemia cell line K-562 and Burkitt lymphoma cell line Daudi were used as targets for reactivity and immune cell killing assays. All of the cell lines were obtained from the ATCC. The cell lines were authenticated by STR profiling and tested for mycoplasma. HCT-15, LoVo, HT-29, K-562 and Daudi cells were maintained in RPMI (Gibco)/10% FCS. SW403 and SK-CO-1 were maintained in DMEM/F12 (Gibco)/10% FCS. All adherent cell lines were trypsinized before passaging. The *B2M*-knockin HCT-15 and LoVo cell lines were generated using the *B2M* plasmid (pLV[Exp]-EF1A>hB2M[NM_004048.4] (ns):T2A:Puro), produced in lentivirus according to standard methodology. Cells were selected using puromycin and then FACS-sorted on the basis of HLA-A/B/C expression using 1:100 anti-HLA-A/B/C-FITC (W6/32, eBioscience).

## Organoid models and culture

Tumour organoids were derived from MMR-d CRC tumours of two patients through resection from the colon (tumour organoid 1) or peritoneal biopsy (tumour organoid 2) (Supplementary Table 5). Establishment of the respective organoid lines from tumour material was performed as previously reported[66,67]. In brief, tumour tissue was mechanically dissociated and digested with 1.5 mg ml$^{-1}$ of collagenase II (Sigma-Aldrich), 10 μg ml$^{-1}$ of hyaluronidase type IV (Sigma-Aldrich) and 10 μM Y-27632 (Sigma-Aldrich). Cells were embedded in Cultrex RGF BME type 2 (3533-005-02, R&D systems) and placed into a 37 °C incubator for 20 min. Human CRC organoid medium is composed of Ad-DF+++ (Advanced DMEM/F12 (GIBCO) supplemented with 2 mM Ultraglutamine I (Lonza), 10 mM HEPES (GIBCO), 100 U ml$^{-1}$ of each penicillin and streptomycin (GIBCO), 10% noggin-conditioned medium, 20% R-spondin1-conditioned medium, 1× B27 supplement without vitamin A (GIBCO), 1.25 mM *N*-acetylcysteine (Sigma-Aldrich), 10 mM nicotinamide (Sigma-Aldrich), 50 ng ml$^{-1}$ human recombinant EGF (Peprotech), 500 nM A83-01 (Tocris), 3 μM SB202190 (Cayman Chemicals) and 10 nM prostaglandin E2 (Cayman Chemicals). Organoids were passaged depending on growth every 1–2 weeks by incubating in TrypLE Express (Gibco) for 5–10 min followed by embedding in BME. Organoids were authenticated by SNP array or STR analysis and were regularly tested for Mycoplasma using Mycoplasma PCR43 and the MycoAlert Mycoplasma Detection Kit (LT07-318). In the first two weeks of organoid culture, 1× Primocin (Invivogen) was added to prevent microbial contamination. Procedures performed with patient samples were approved by the Medical Ethical Committee of the Netherlands Cancer Institute–Antoni van Leeuwenhoek hospital (NL48824.031.14) and written informed consent was obtained from all of the patients. Mismatch repair status was assessed using a standard protocol for the Ventana automated immunostainer for MLH1 clone M1 (Roche), MSH2 clone G219-1129 (Roche), MSH6 clone EP49 (Abcam) and PMS2 clone EP51 (Agilant Technologies). The *B2M$^{KO}$* tumour organoid lines were generated using sgRNA targeting *B2M* (GGCCGAGATGTCTCGCTCCG), cloned into LentiCRISPR v2 plasmid. The virus was produced using a standard method.

## Screening of cancer cell lines and tumour organoids by flow cytometry

The cancer cell lines used in the reactivity and killing assays were screened for the expression of B2M, HLA class I molecules, NKG2D ligands, DNAM-1 ligands and butyrophilin using flow cytometry. In brief, cells were incubated with human Fc receptor block (BioLegend) and stained with the cell surface antibodies in different experiments (anti-CD112-PE (1:10, R2.525, BD Biosciences), anti-CD155-PE (1:10, 300907, R&D Systems), anti-CD277/BTN3A1-PE (1:50, BT3.1, Miltenyi), anti-B2M-PE (1:100, 2M2, BioLegend), anti-HLA-A/B/C-FITC (1:100, W6/32, eBioscience), anti-HLA-A/B/C-AF647 (1:160, W6/32, BioLegend),

anti-HLA-E-BV421 (1:20, 3D12, BioLegend), anti-HLA-G-APC (1:20, 87G, BioLegend), anti-MICA/B-PE (1:300, 6D4, BioLegend), anti-ULBP1-PE (1:10, 170818, R&D Systems), anti-ULBP2/5/6-PE (1:20, 165903, R&D Systems), anti-ULBP3-PE (1:20, 166510, R&D Systems) or anti-ULBP4-PE (1:20, 709116, R&D Systems)) for 45 min at 4 °C. A 1:1,000 live/dead fixable near-infrared viability dye (Life Technologies) was included in each staining. Cells were washed three times in FACS buffer (PBS/1% FCS). Compensation was performed using CompBeads (BD Biosciences) and ArC reactive beads (Life Technologies). Cells were acquired on the FACS Canto II 3L or FACS LSR Fortessa 4L (BD Biosciences) system running FACSDiva software (v.9.0; BD Biosciences). Isotype or FMO controls were included to determine the percentage of positive cancer cells. Data were analysed using FlowJo v.10.6.1 (Tree Star).

For organoid surface staining, tumour organoids were dissociated into single cells using TrypLE Express (Gibco), washed twice in cold FACS buffer (PBS, 5 mM EDTA, 1% bovine serum antigen), and stained with anti-HLA-A/B/C-PE (1:20, W6/32, BD Biosciences), anti-B2M-FITC (1:100, 2M2, BioLegend), anti-PD-L1-APC (1:200, MIH1, eBioscience) and 1:2,000 near-infrared (NIR) viability dye (Life Technologies), or isotype controls (1:1,000 FITC; 1:20, PE; or 1:200, APC) mouse IgG1 kappa (BD Biosciences). For NKG2D ligand expression analysis, cells were stained with anti-MICA/MICB (1:300), anti-ULBP1 (1:10), anti-ULBP2/5/6 (1:20), anti-ULBP3 (1:20), anti-ULBP4 (1:20) and 1:2,000 near-infrared (NIR) viability dye (Life Technologies). Tumour cells were incubated for 30 min at 4 °C in the dark and washed twice with FACS buffer. All of the samples were recorded with the BD LSR Fortessa Cell Analyzer SORP flow cytometer using FACSDiVa (v.8.0.2; BD Biosciences). Data were analysed using FlowJo (v.10.6.1; BD) and presented using GraphPad Prism (v.9.0.0; GraphPad).

### Reactivity assay of γδ T cells

The reactivity of γδ T cells to the different cancer cell lines was assessed by a co-culture reactivity assay. γδ T cells were thawed and cultured in IMDM + GlutaMax (Gibco)/8% nHS medium with penicillin (100 IU ml$^{-1}$) and streptomycin (100 µg ml$^{-1}$) in the presence of low-dose IL-2 (25 IU ml$^{-1}$) and IL-15 (5 ng ml$^{-1}$) overnight at 37 °C. Cancer cell lines were counted, adjusted to a concentration of $0.5 \times 10^5$ cells per ml in IMDM + GlutaMax/10% FCS medium with penicillin (100 IU ml$^{-1}$) and streptomycin (100 µg ml$^{-1}$), and seeded (100 µl per well) in coated 96-well flat-bottom microplates (Greiner CellStar) (for 5,000 cells per well) overnight at 37 °C. The next day, γδ T cells were collected, counted and adjusted to a concentration of $1.2 \times 10^6$ cells per ml in IMDM + GlutaMax/10% FCS medium. The γδ T cells were added in 50 µl (for 60,000 cells per well) and co-cultured (12:1 effector:target ratio) at 37 °C for 18 h in biological triplicates. The medium (without cancer cells) was used as a negative control and PMA (20 ng ml$^{-1}$)/ionomycin (1 µg ml$^{-1}$) was used as a positive control. After co-culture, the supernatant was collected to detect IFNγ secretion by enzyme-linked immunosorbent assay (Mabtech) according to the manufacturer's instructions. Moreover, cells were collected, incubated with human Fc receptor block (BioLegend) and stained with cell surface antibodies (anti-CD137-APC (1:100, 4B4-1, BD Biosciences), anti-CD226/DNAM-1-BV510 (1:150, DX11, BD Biosciences), anti-CD3-AF700 (1:400, UCHT1, BD Biosciences), anti-CD39-APC (1:80, A1, BioLegend), anti-CD40L-PE (1:10, TRAP1, BD Biosciences), or anti-PD-1-PE (1:30, MIH4, eBioscience), anti-TCRγδ-BV650 (1:40, 11F2, BD Biosciences), anti-NKG2D-PE-Cy7 (1:300, 1D11, BD Biosciences) and anti-OX40-FITC (1:20, ACT35, BioLegend)) for 45 min at 4 °C. A 1:1,000 live/dead fixable near-infrared viability dye (Life Technologies) was included in each staining. Cells were washed three times in FACS buffer (PBS/1% FCS). Compensation was performed using CompBeads (BD Biosciences) and ArC reactive beads (Life Technologies). Cells were acquired on the FACS LSR Fortessa X-20 4L (BD Biosciences) system running FACSDiva software (v.9.0; BD Biosciences). Data were analysed using FlowJo (v.10.6.1; Tree Star). All data are representative of at least two independent experiments.

### Immune cell killing assay γδ T cells

Killing of the different cancer cell lines by γδ T cells was visualized and quantified by a co-culture immune cell killing assay using the IncuCyte S3 Live-Cell Analysis System (Essen Bioscience). HCT-15, LoVo and HT-29 cells were transduced with IncuCyte NucLight Red Lentivirus Reagent (EF-1α, Puro; Essen BioScience) providing a nuclear-restricted expression of a red (mKate2) fluorescent protein. In brief, HCT-15, LoVo and HT-29 cells were seeded, transduced according to the manufacturer's instructions and stable cell populations were generated using puromycin selection. The *B2M*-knockin cell lines were created under puromycin selection; therefore, stable NucLight Red-expressing cell populations were generated by sorting for mKate2 (the red fluorescent protein) in the PE Texas Red filter set instead. Cancer cell lines were counted, adjusted to a concentration of $1 \times 10^5$ cells per ml in IMDM + GlutaMax/10% FCS medium with penicillin (100 IU ml$^{-1}$) and streptomycin (100 µg ml$^{-1}$), and seeded (100 µl per well) in 96-well flat-bottom clear microplates (Greiner CellStar) (for 10,000 cells per well). The target cell plate was placed in the IncuCyte system at 37 °C to monitor for cell confluency for 3 days. On day 2, γδ T cells were thawed and cultured in IMDM + GlutaMax/8% nHS medium with penicillin (100 IU ml$^{-1}$) and streptomycin (100 µg ml$^{-1}$) in the presence of low-dose IL-2 (25 IU ml$^{-1}$) and IL-15 (5 ng ml$^{-1}$) overnight at 37 °C. The next day, γδ T cells were collected, counted and adjusted to a concentration of $7.2 \times 10^5$ cells per ml in IMDM + GlutaMax/10% FCS medium. After aspiration of the medium of the target cell plate, 100 µl of new medium containing 3.75 µM IncuCyte Caspase-3/7 Green Apoptosis Reagent (Essen BioScience) (1.5× final assay concentration of 2.5 µM) was added together with 50 µl of γδ T cells (for 36,000 cells per well). They were co-cultured (4:1 effector:target ratio) in the IncuCyte system at 37 °C in biological duplicates. Cancer cells alone and cancer cells alone with caspase-3/7 were used as negative controls. Images (2 images per well) were captured every hour at ×20 magnification with the phase, green and red channels for up to 4 days.

Analysis was performed using the IncuCyte software (v.2020B) for each cancer cell line separately. The following analysis definitions were applied: a minimum phase area of 200 µm$^2$, RCU of 2.0, and a GCU of 2.0 (for HCT-15 cells) and 4.0 (for LoVo and HT-29 cells). Cancer cell apoptosis was then quantified in the IncuCyte software by counting the total number of green + red objects per image normalized (by division) to the total number of red objects per image after 12 h co-culture and displayed as a percentage (mean ± s.e.m.) of two wells with two images per well. For the comparison of the killing of *B2M*-knockin HCT-15 and LoVo cell lines versus the WT cell lines, Caspase-3/7 Red Apoptosis Reagent (Essen BioScience) was used. The transfection of the target reporter was not as successful in combination with the *B2M*-knockin. Thus, apoptosis was quantified by dividing the red area by the phase area and displayed as a percentage (mean ± s.e.m.) of two wells with two images per well. The following analysis definitions were applied: a minimum phase area of 100 µm$^2$ and a RCU of 0.5 (for HCT-15 cells) and 0.75 (for LoVo cells).

### Tumour organoid recognition assay

For evaluation of tumour reactivity towards *B2M*$^{WT}$ and *B2M*$^{KO}$ organoids and NKG2D ligand blocking conditions, tumour organoids and γδ T cells were prepared as described previously[9,66,67]. Two days before the experiment, organoids were isolated from BME by incubation in 2 mg ml$^{-1}$ type II dispase (Sigma-Aldrich) for 15 min before addition of 5 mM EDTA and washed with PBS before being resuspended in CRC organoid medium with 10 µM Y-27632 (Sigma-Aldrich). The organoids were stimulated with 200 ng ml$^{-1}$ IFNγ (Peprotech) 24 h before the experiment. For the recognition assay and intracellular staining, tumour organoids were dissociated into single cells and plated in anti-CD28-coated (CD28.2, eBioscience) 96-well U-bottom plates with γδ T cells at a 1:1 target:effector ratio in the presence of 20 µg ml$^{-1}$

anti-PD-1 (Merus). As a positive control, γδ T cells were stimulated with 50 ng ml$^{-1}$ of phorbol-12-myristate-13-acetate (Sigma-Aldrich) and 1 µg ml$^{-1}$ of ionomycin (Sigma-Aldrich). After 1 h of incubation at 37 °C, GolgiSTOP (BD Biosciences, 1:1,500) and GolgiPlug (BD Biosciences, 1:1,000) were added. After 4 h of incubation at 37 °C, γδ T cells were washed twice in cold FACS buffer (PBS, 5 mM EDTA, 1% bovine serum antigen) and stained with anti-CD3-PerCP-Cy5.5 (1:20, BD Biosciences), anti-TCRγδ-PE (1:20, BD Bioscience), anti-CD4-FITC (1:20, BD Bioscience) (not added in experiments with NKG2D ligand blocking), anti-CD8-BV421 (1:200, BD Biosciences) and 1:2,000 near-infrared (NIR) viability dye (Life Technologies) for 30 min at 4 °C. Cells were washed, fixed and stained with 1:40 anti-IFNγ-APC (BD Biosciences) for 30 min at 4 °C, using the Cytofix/Cytoperm Kit (BD Biosciences). After two wash steps, cells were resuspended in FACS buffer and recorded with the BD LSR Fortessa Cell Analyzer SORP flow cytometer using FACSDiVa software (v.8.0.2; BD Biosciences). Data were analysed using FlowJo (v.10.6.1, BD) and presented using GraphPad Prism (v.9.0.0, GraphPad).

## Blocking experiments with cancer cell lines and tumour organoids

The reactivity of and killing by the γδ T cells was examined in the presence of different blocking antibodies to investigate which receptor–ligand interactions were involved. For DNAM-1 blocking, γδ T cells were incubated with 3 µg ml$^{-1}$ purified anti-DNAM-1 (DX11, BD Biosciences) for 1 h at 37 °C. For γδ TCR blocking, γδ T cells were incubated with 3 µg ml$^{-1}$ purified anti-TCRγδ (5A6.E9, Invitrogen) for 1 h at 37 °C; the clone that we used was tested to be the best for use in γδ TCR blocking assays[68]. NKG2D ligands were blocked on the cancer cell lines and single cells of tumour organoids by incubating the target cells with 12 µg ml$^{-1}$ anti-MICA/B (6D4, BioLegend), 1 µg ml$^{-1}$ anti-ULBP1 (170818, R&D Systems), 3 µg ml$^{-1}$ anti-ULBP2/5/6 (165903, R&D Systems) and 6 µg ml$^{-1}$ anti-ULBP3 (166510, R&D Systems) for 1 h at 37 °C before plating with γδ T cells. After incubation with the blocking antibodies, the γδ T cells were added to cancer cell lines HCT-15, LoVo and HT-29 as described above with a minimum of two biological replicates per blocking condition. For organoid experiments, anti-CD107a-FITC (1:50, H4A3, BioLegend) was added during incubation.

As a control for Fc-mediated antibody effector functions, γδ T cells alone were incubated with the blocking antibodies in the presence of 2.5 µM IncuCyte Caspase-3/7 Green Apoptosis Reagent (Essen BioScience) in the IncuCyte system at 37 °C, and the number of apoptotic γδ T cells was quantified over time.

## Data analysis and visualization

Bulk DNA-seq and RNA-seq data were analysed using Python (v.3) and R (v.3.6.1) in Jupyter Notebook (v.6.0.1). Numpy (v.1.17.2) and Pandas (v.0.25.1) were used for array and data frame operations, respectively. Data visualization was performed using Matplotlib (v.3.2.1) and Seaborn (v.0.9.0). scRNA-seq data were analysed using Cell Ranger (v.3.1.0), R (v.4.1.0) and Seurat (v.3.1.5). IMC data were analysed using ilastik (v.1.3.3), CellProfiler (v.2.2.0), ImaCytE (v.1.1.4) and Cytosplore (v.2.3.0). Flow cytometry data were analysed using FlowJo (v.10.6.1). IncuCyte data were analysed using IncuCyte (v.2020B). Data visualization was performed using GraphPad Prism (v.9.0.0 and v.9.0.1).

## Reporting summary

Further information on research design is available in the Nature Portfolio Reporting Summary linked to this article.

## Data availability

The TCGA data used here are publicly available at the National Cancer Institute GDC Data Portal (https://portal.gdc.cancer.gov; cohorts COAD, STAD and UCEC). Of the DRUP study participants included in this preliminary analysis across all (complete and incomplete) cohorts of the study, we included all clinical data, genomics data on *B2M* status and RNA expression data of marker gene sets in Supplementary Table 1. The raw sequencing data of the DRUP and Hartwig cohorts can be accessed through Hartwig Medical Foundation on approval of a research access request (https://www.hartwigmedicalfoundation.nl/en/data/data-acces-request). As determined in the original publication, NICHE study RNA-seq and DNA-seq data have been deposited into the European Genome–Phenome Archive under accession number EGAS00001004160 and are available on reasonable request for academic use and within the limitations of the provided informed consent. The scRNA-seq data have been deposited at the GEO (GSE216534) and are publicly available. All other data are available from the corresponding author on reasonable request. The GRCh38 primary assembly of the human reference genome was downloaded from Gencode (https://ftp.ebi.ac.uk/pub/databases/gencode/Gencode_human/release_42/GRCh38.primary_assembly.genome.fa.gz) with Gencode's matching v29 annotation file (https://ftp.ebi.ac.uk/pub/databases/gencode/Gencode_human/release_29/gencode.v29.annotation.gtf.gz) for gene expression quantification.

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

**Acknowledgements** We thank K. C. M. J. Peeters, M. G. Kallenberg-Lantrua, D. Berends-van der Meer and F. A. Holman for their help in collecting and providing samples from patients with

colon cancer; the staff at the Flow Cytometry Core Facility of the Leiden University Medical Center for their help with cell sorting; the staff at the Leiden Genome Technology Center for their help with scRNA-seq; M. Ganesh for help with cell culturing; D. Thommen for discussions; I. S. Rodriguez for the establishment of a *B2M*-knockout organoid line; L. Hoes for initial clinical findings; the staff at the Flow Cytometry Core Facility at the Netherlands Cancer Institute for their support; X. Kong for providing the lentiCRISPR plasmid for *B2M* knockout; the staff at Merus for providing anti-PD-1 antibodies for organoid experiments; and the staff at The Cancer Genome Atlas (TCGA) for providing data used in this manuscript. N.F.C.C.d.M. is funded by the European Research Council (ERC) under the European Union's Horizon 2020 Research and Innovation Programme (grant agreement no. 852832). E.E.V. received funding from the Oncode Institute and Open Targets (Identification of targets modulating lymphocyte-mediated tumour cell killing (project ID: OTAR2061); project leaders, M. Garnett and E.E.V.) and the Josephine Nefkens Foundation. F.K. was supported by the collaboration project TIMID (LSHM18057-SGF) financed by the PPP allowance made available by Top Sector Life Sciences & Health to Samenwerkende Gezondheidsfondsen (SGF) to stimulate public–private partnerships and co-financing by health foundations that are part of the SGF.

**Author contributions** N.L.d.V., J.v.d.H. and V.V. conceived the study and performed experiments. J.v.d.H. performed genomic and bulk transcriptomic analyses of ICB-naive as well as ICB-treated MMR-d cancers. N.L.d.V. performed scRNA-seq and cell culturing experiments. V.V. performed organoid experiments. N.L.d.V., M.v.d.P. and J.v.d.B. performed cell line reactivity and immune cell killing experiments. N.L.d.V., V.V. and M.v.d.P. performed blocking experiments. M.C. provided tissue sections of patients in the NICHE study, which was designed and coordinated by M.C. under the joint supervision of T.N.S., E.E.V. and J.B.H.; M.E.I. performed IMC experiments. M.E.I., N.F.C.C.d.M., N.L.d.V. and D.R. analysed the IMC data. N.L.d.V. and D.R. analysed the scRNA-seq data. J.G.v.d.B. evaluated histological and immunohistochemical analyses. L.J.Z., B.S.G., G.F.d.W., T.W.B., H.G. and H.M.W.V. designed, coordinated and analysed data from the DRUP study. General scientific coordination was undertaken by J.v.d.H.; L.F.A.W., F.K., N.F.C.C.d.M. and E.E.V. supervised the study; T.N.S. had an advisory role. The manuscript was written by N.L.d.V., J.v.d.H. and V.V. in collaboration with all of the authors. All of the authors commented on and approved the manuscript.

**Competing interests** M.C. has performed an advisory role or offered expert testimony for BMS, MSD and NUMAB; has received honoraria from BMS and Roche; and has received financing of scientific research from Roche, BMS and MSD. J.B.H. has received research funding from BMS; and has performed an advisory role for BMS.

**Additional information**
**Correspondence and requests for materials** should be addressed to Noel F. C. C. de Miranda or Emile E. Voest.

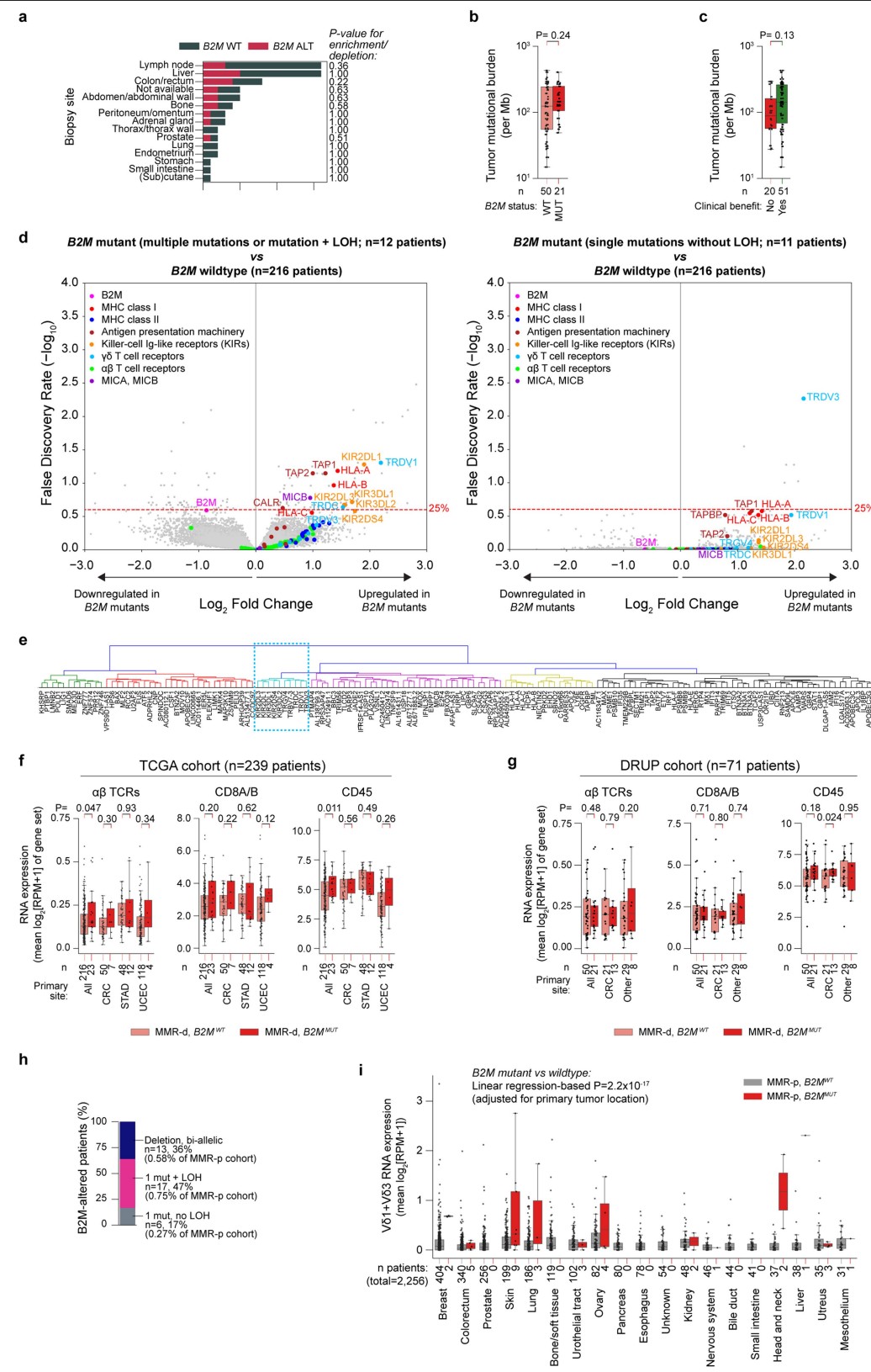

**Extended Data Fig. 1** | See next page for caption.

**Extended Data Fig. 1 | Association of *B2M* status to clinical, genomic, and transcriptomic characteristics of MMR-d and MMR-p tumours. a**. Biopsy site distribution in the DRUP cohort (n = 71 patients). Colours denote patients' *B2M* status (WT: wildtype,; ALT: altered). Fisher's exact test-based two-sided P-values for enrichment/depletion of *B2M* altered tumours per biopsy site are shown. **b**. Tumour mutational burden *vs B2M* status. Wilcoxon rank sum test-based two-sided P-value is shown. Boxes, whiskers, and dots indicate quartiles, 1.5 interquartile ranges, and individual data points, respectively. **c**. As **b**, but for clinical benefit to ICB (x-axis). **d**. Volcano plots indicating differential gene expression between B2M-based subgroups (see title) of MMR-d cancers in TCGA COAD (colon adenocarcinoma; n = 57 patients), STAD (stomach adenocarcinoma; n = 60 patients) and UCEC (uterus corpus endometrial carcinoma; n = 122 patients) cohorts. Results were adjusted for tumour type and multiple hypothesis testing (Methods). **e**. Hierarchical clusters of expression of genes significantly (FDR <25%; see Fig. 1D) upregulated in MMR-d *B2M*$^{MUT}$ *vs* MMR-d *B2M*$^{WT}$ cancers in the TCGA COAD/STAD/UCEC cohorts. The blue dashed rectangle denotes the Vδ1/3 T cell cluster. **f**. Immune marker gene set expression in *B2M*$^{WT}$ (pink), and *B2M*$^{MUT}$ (red) MMR-d cancers in the TCGA COAD/STAD/UCEC cohorts separately or combined (All). Boxes, whiskers, and dots indicate quartiles, 1.5 interquartile ranges, and individual data points, respectively. Wilcoxon rank sum test-based two-sided P-values are shown. **g**. As **f**, but for MMR-d cancers in the DRUP cohort, for all cancers combined (All), only colorectal cancer (CRC), or all non-CRC cancers (Other). Two-sided P-values were calculated with linear regression adjusting for biopsy site and tumour type (Methods). **h**. Allelic alteration status of *B2M* in the Hartwig cohort of MMR-p cancers. **i**. RNA expression of Vδ1+Vδ3 loci in MMR-p *B2M*$^{WT}$ (grey), and MMR-p *B2M*$^{MUT}$ (red) cancers in the Hartwig cohort, stratified per primary tumour location. Boxes, whiskers, and dots indicate quartiles, 1.5 interquartile ranges, and individual data points, respectively. The linear regression-based, two-sided, primary tumour location-adjusted P-value for association of *B2M* status with Vδ1+Vδ3 loci expression is shown.

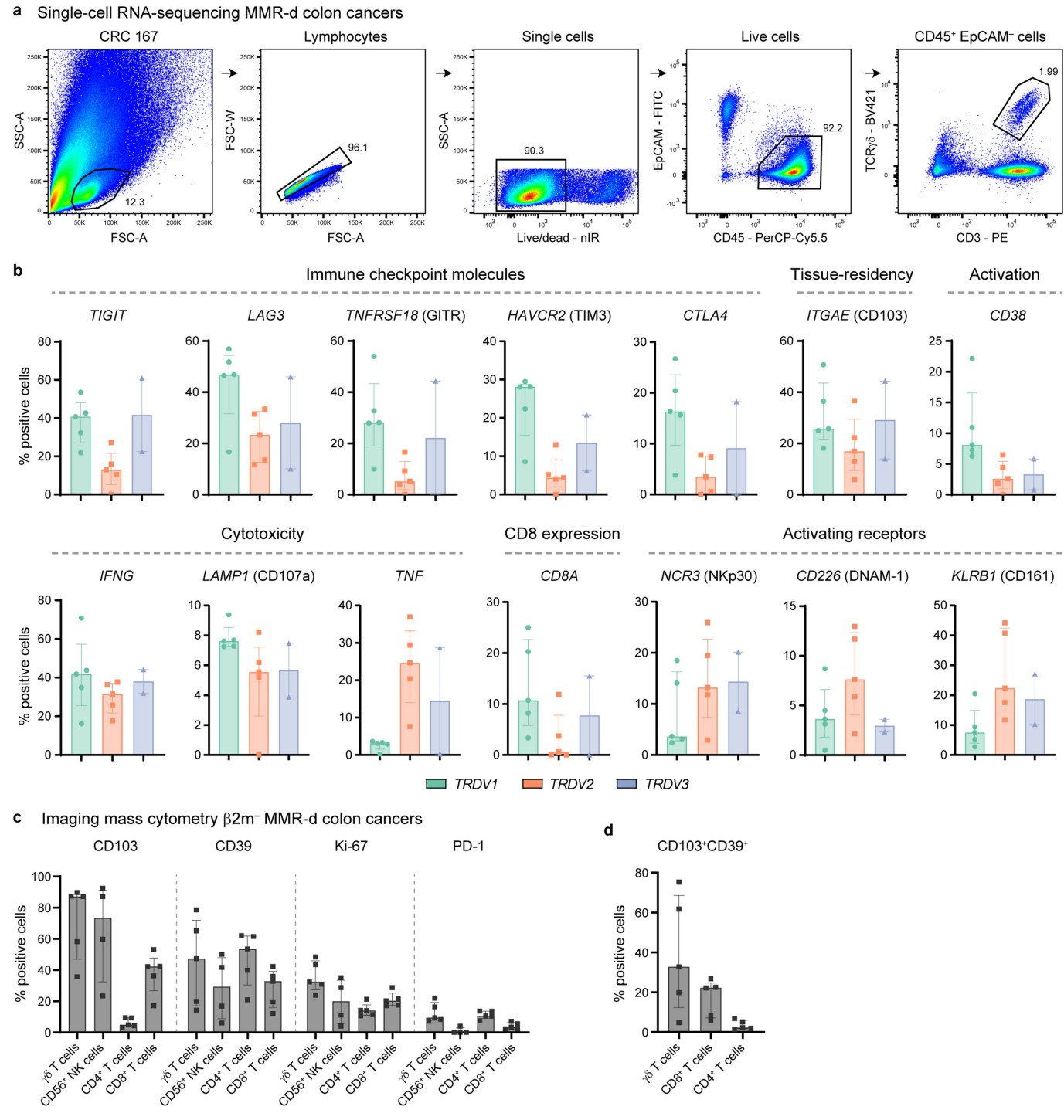

**a** Single-cell RNA-sequencing MMR-d colon cancers

**b** Immune checkpoint molecules | Tissue-residency | Activation

*TIGIT* *LAG3* *TNFRSF18* (GITR) *HAVCR2* (TIM3) *CTLA4* *ITGAE* (CD103) *CD38*

Cytotoxicity | CD8 expression | Activating receptors

*IFNG* *LAMP1* (CD107a) *TNF* *CD8A* *NCR3* (NKp30) *CD226* (DNAM-1) *KLRB1* (CD161)

*TRDV1*  *TRDV2*  *TRDV3*

**c** Imaging mass cytometry β2m⁻ MMR-d colon cancers

CD103  CD39  Ki-67  PD-1

**d** CD103⁺CD39⁺

**Extended Data Fig. 2 | Characterization of γδ T cells and other immune cell populations infiltrating MMR-d colon cancers. a.** FACS gating strategy for single, live CD45⁺ EpCAM⁻ CD3⁺ TCRγδ⁺ cells from a representative MMR-d colon cancer sample showing sequential gates with percentages. **b.** Frequencies of positive cells for selected genes across Vδ1 (n = 1927), Vδ2 (n = 860), and Vδ3 (n = 506) cells as percentage of total γδ T cells from each MMR-d colon tumour (n = 5) analysed by single-cell RNA-sequencing. Vδ3 cells were present in two out of five colon cancers. Bars and dots indicate median ± IQR and individual

samples, respectively. **c.** Frequencies of marker-positive γδ T cells, CD56⁺ NK cells, CD4⁺ T cells, and CD8⁺ T cells in treatment-naive β2m⁻ (n = 5) MMR-d colon cancers. CD56⁺ NK cells were present in four out of five β2m⁻ cancer samples. Bars and dots indicate median ± IQR and individual samples, respectively. **d.** Frequencies of CD103⁺CD39⁺ γδ T cells, CD8⁺ T cells, and CD4⁺ T cells in treatment-naive β2m⁻ (n = 5) MMR-d colon cancers. Bars indicate median ± IQR and individual samples, respectively.

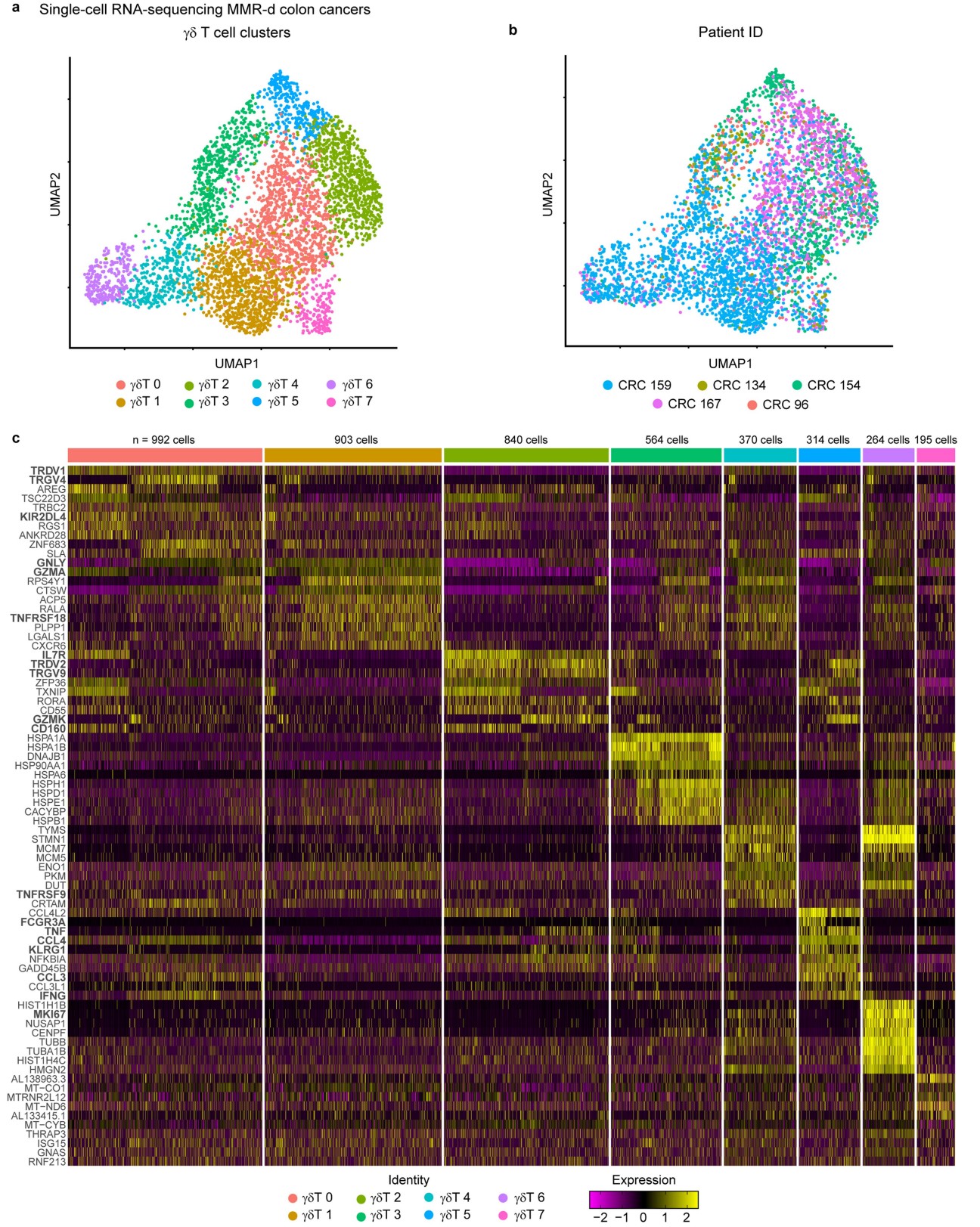

**Extended Data Fig. 3 | Distinct clusters of γδ T cells in MMR-d colon cancers by single-cell RNA-sequencing. a.** UMAP embedding showing γδ T cells (n = 4442) isolated from MMR-d colon cancers (n = 5) analysed by single-cell RNA-sequencing. Colours represent the functionally different γδ T cell clusters identified by graph-based clustering and non-linear dimensional reduction. Dots represent single cells. **b.** UMAP embedding of (**a**) coloured by patient ID. Dots represent single cells. **c.** Heatmap showing the normalized single-cell gene expression value (z-score, purple-to-yellow scale) for the top 10 differentially expressed genes in each identified γδ T cell cluster.

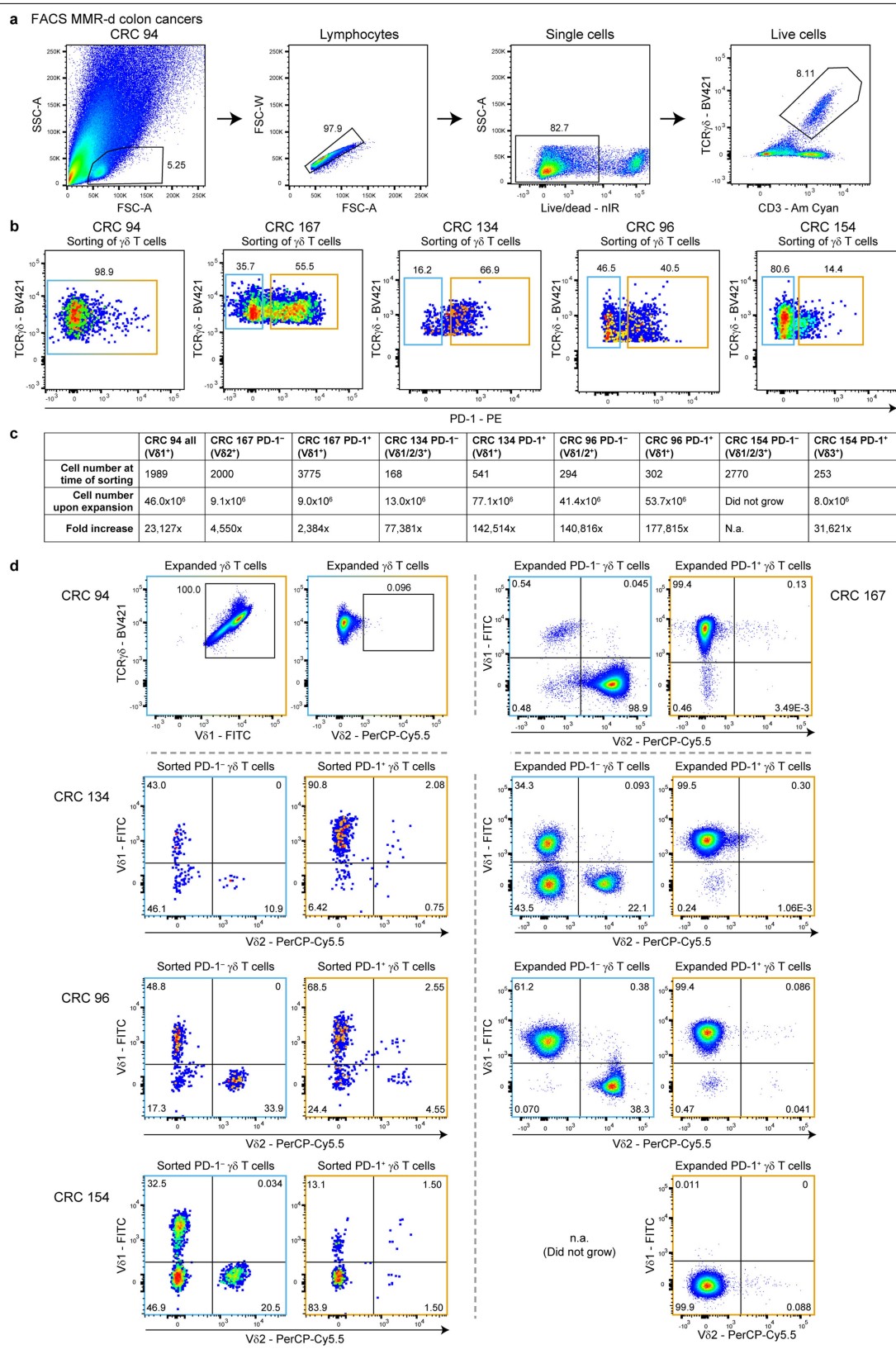

**Extended Data Fig. 4 | Sorting and expansion of γδ T cells from MMR-d colon cancers. a**. FACS gating strategy for single, live CD3⁺ TCRγδ⁺ cells from a representative MMR-d colon cancer sample showing sequential gates with percentages. **b**. Sorting of all γδ T cells from CRC94 (due to the low number of PD-1⁺ cells), and of PD-1⁻ (blue squares) and PD-1⁺ (orange squares) γδ T cells from CRC167, CRC134, CRC96, and CRC154. Dots represent single cells. **c**. Table showing the number of γδ T cells isolated from MMR-d colon cancers (n = 5) at the time of sorting *vs* 3-4 weeks after expansion, and the fold increase thereof. **d**. TCR Vδ chain usage after expansion of γδ T cells from CRC94 and CRC167 (first row), and at the time of sorting (left panel) as well as after expansion (right panel) of γδ T cells from CRC134, CRC96, and CRC154. PD-1⁻ γδ T cells from CRC154 did not expand in culture. Dots represent single cells.

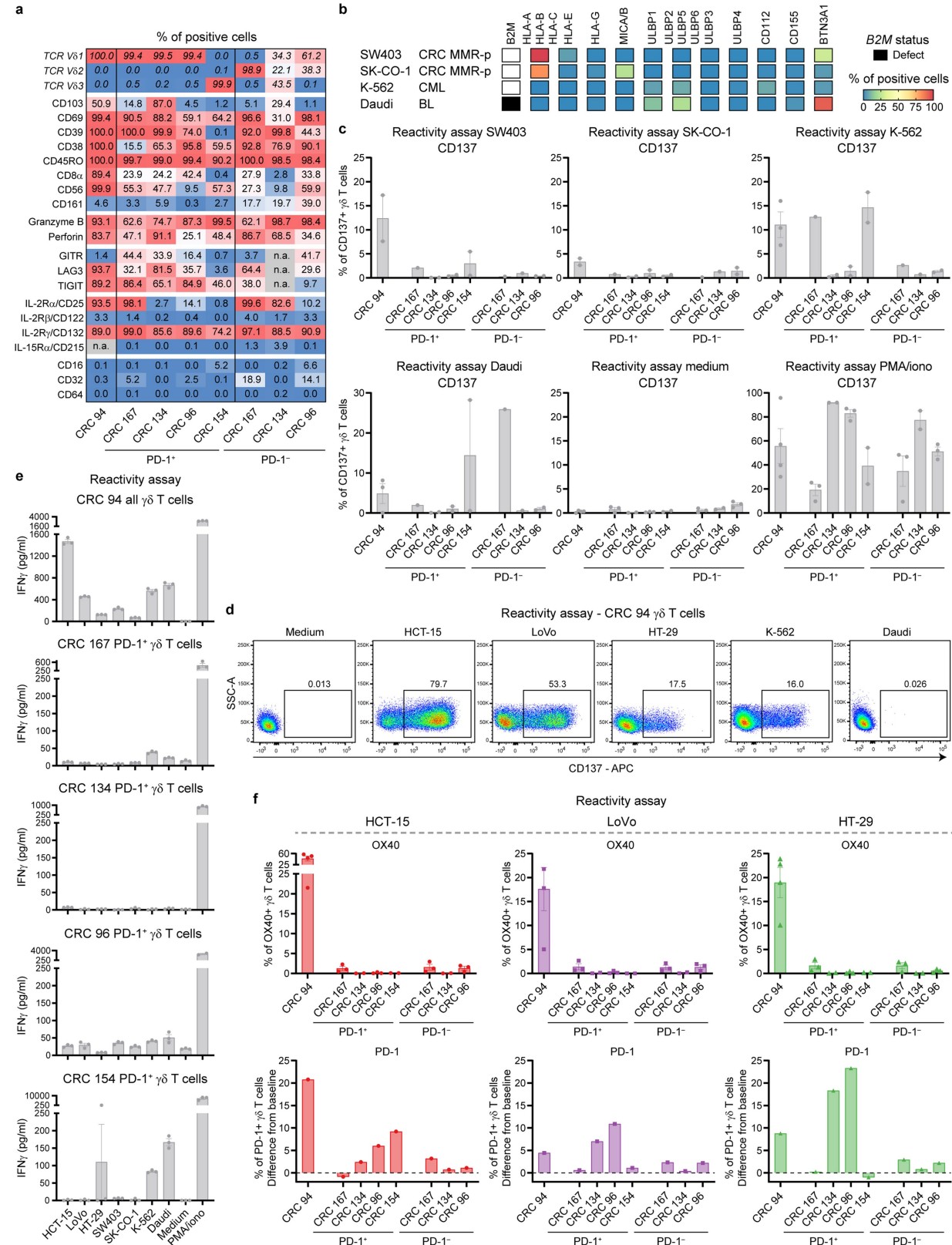

**Extended Data Fig. 5** | See next page for caption.

**Extended Data Fig. 5 | Phenotype and reactivity of γδ T cells towards cancer cell lines. a**. The percentage of positive cells for the indicated markers on expanded γδ T cells from MMR-d colon cancers (n = 5). **b**. Diagram showing the *B2M* status and surface expression of HLA class I, NKG2D ligands, DNAM-1 ligands, and butyrophilin on cancer cell lines. **c**. Bar plots showing CD137 expression on γδ T cells upon co-culture with cancer cell lines. Medium was used as negative control and PMA/ionomycin as positive control. Bars indicate mean ± SEM. Data from at least two independent experiments except for CRC167, depending on availability of γδ T cells. **d**. Representative flow cytometry plots showing CD137 expression on γδ T cells from a MMR-d colon cancer upon co-culture with cancer cell lines as compared to medium only. Gates indicate percentage of positive γδ T cells. **e**. Bar plots showing the presence of IFNγ in the supernatant upon co-culture of γδ T cells from MMR-d colon cancers (n = 5) with cancer cell lines. Medium as negative control and PMA/ionomycin as positive control are included. Bars indicate mean ± SEM of triplicates. **f**. Bar plots showing the expression of OX40 (first row) and PD-1 (second row) on γδ T cells upon co-culture of γδ T cells with CRC cell lines. PD-1 expression is shown as difference from baseline (medium) condition. Bars indicate mean ± SEM. Data from at least two independent experiments.

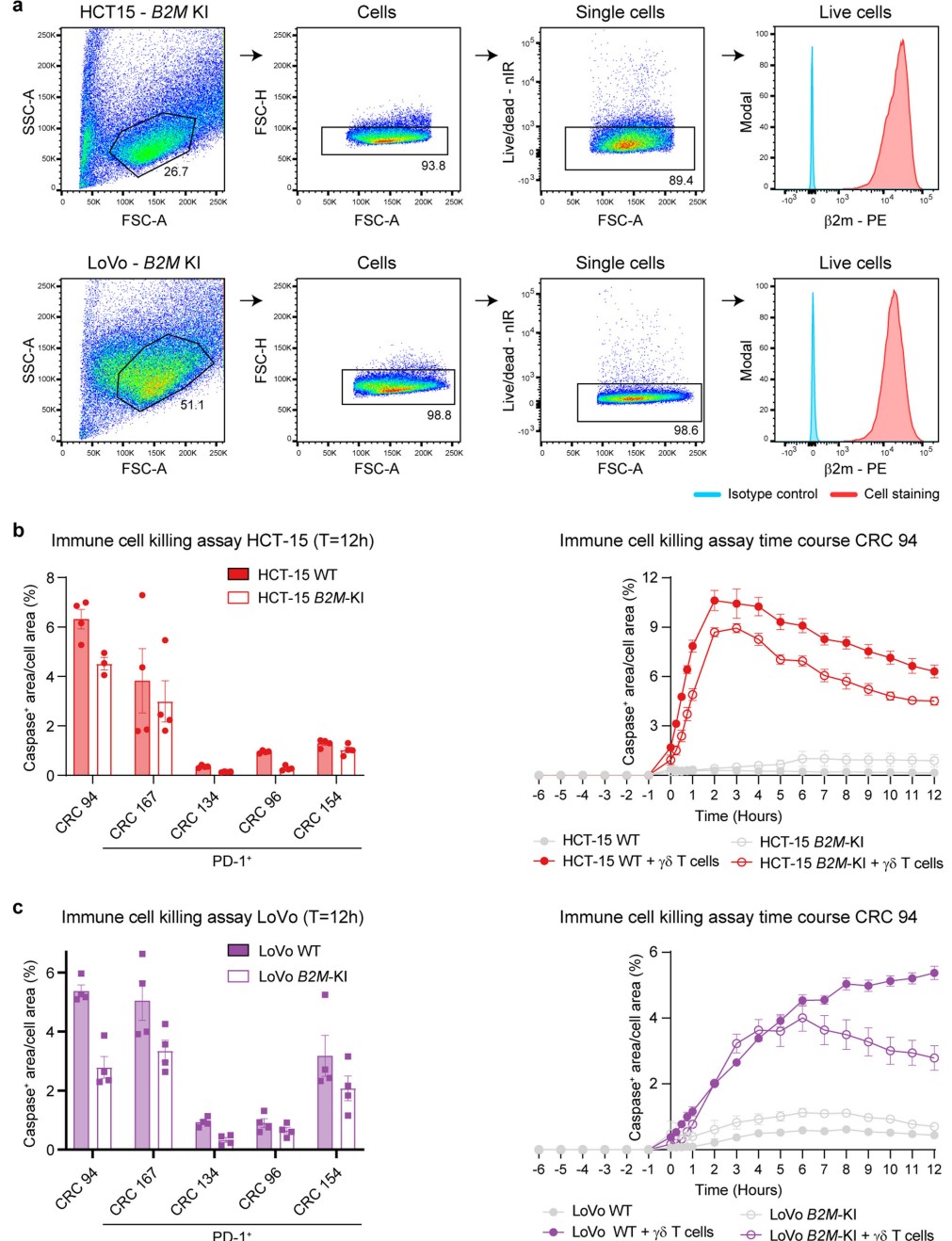

**Extended Data Fig. 6 | Reactivity of γδ T cells towards *B2M*-knockin vs -wildtype cancer cell lines. a.** Flow cytometry gating strategy to validate β2m expression on HCT-15 and LoVo *B2M*-knockin (*B2M*-KI) cell lines. Isotype controls were included as negative control. **b.** Bar plots showing the quantification of killing of HCT-15 *B2M*-KI *vs* wildtype (WT) cells upon co-culture with γδ T cells from MMR-d colon cancers (n = 5) in the presence of a red fluorescent caspase-3/7 reagent. Bars indicate mean ± SEM of two wells with two images/ well. Right panel shows representative time course of apoptosis. **c.** As **b**, but for LoVo *B2M*-KI *vs* WT cells.

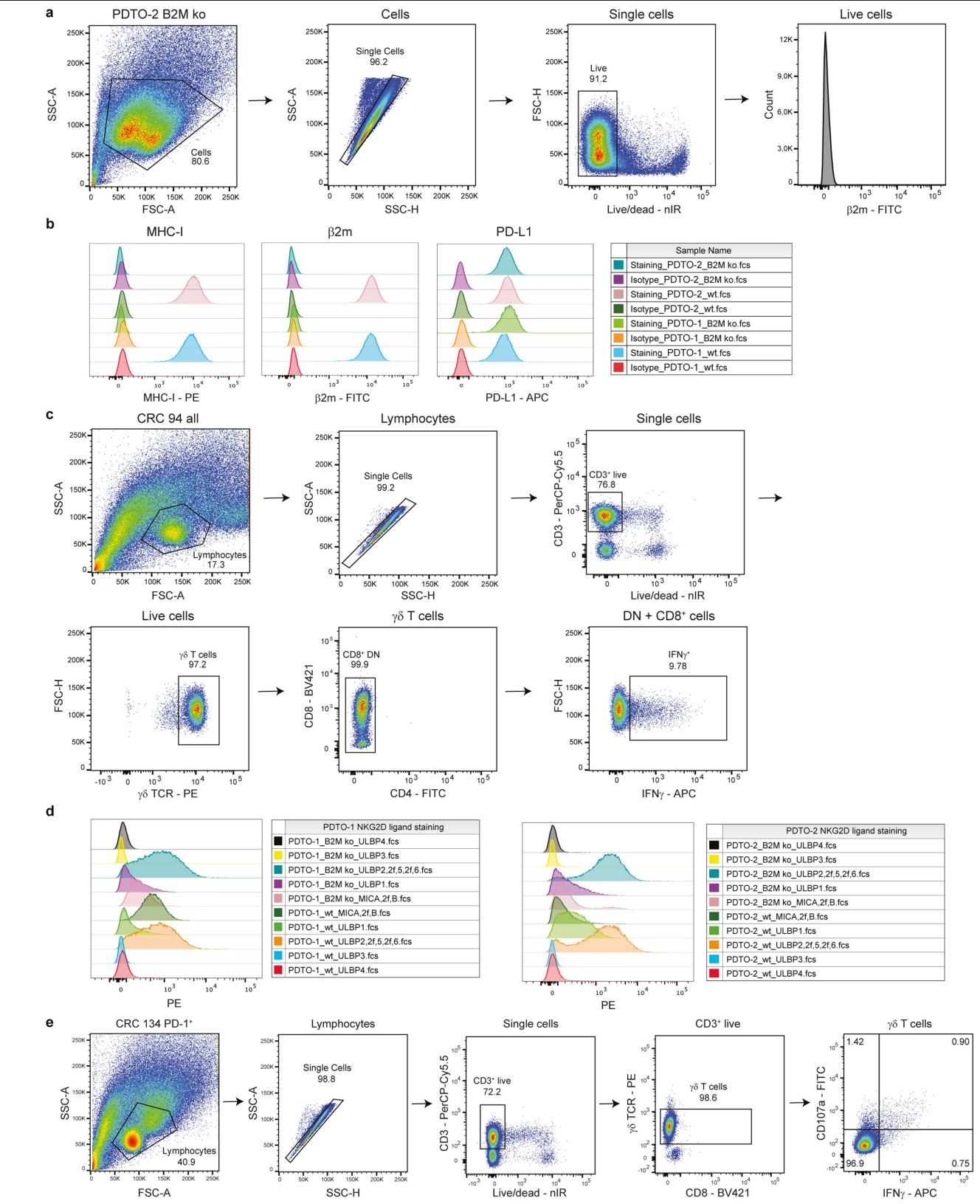

**Extended Data Fig. 7** | See next page for caption.

**Extended Data Fig. 7 | Tumour organoid characterization and reactivity assay readout. a**. Flow cytometry gating strategy on PDTO cells for analysis of surface staining. Selected cells were gated on single, live cells before quantification of staining signal. **b**. Histogram representation and count for surface staining of MHC-I, PD-L1, and β2m expression on two PDTO lines $B2M^{WT}$ and $B2M^{KO}$ after IFNγ pre-stimulation. Staining with isotype antibodies for each fluorochrome (PE, APC and FITC) were included as negative control. **c**. Flow cytometry gating strategy on γδ T cell samples for analysis of intracellular staining to test antitumour reactivity upon PDTO stimulation. Lymphocyte population was further gated on single cells, live and CD3$^+$ cells, γδTCR$^+$ cells and CD8$^+$ as well as CD8$^-$CD4$^-$ cells. Reactivity of the sample was based on IFNγ$^+$ cells of the selected population. **d**. Histogram representation and count for surface staining of NKG2D ligands MICA/B, ULBP1, ULBP2/5/6, ULBP3, and ULBP4 on two PDTO lines $B2M^{WT}$ and $B2M^{KO}$ after IFNγ pre-stimulation. **e**. Flow cytometry gating strategy on γδ T cell samples for analysis of intracellular staining after stimulation with PDTOs in the presence of NKG2D ligand blocking. Lymphocyte population was further gated on single cells, live and CD3$^+$ cells, followed by γδTCR$^+$ and CD8$^+$ as well as CD8$^-$ cells. Reactivity of final population was based on IFNγ$^+$ or CD107a$^+$ cells.

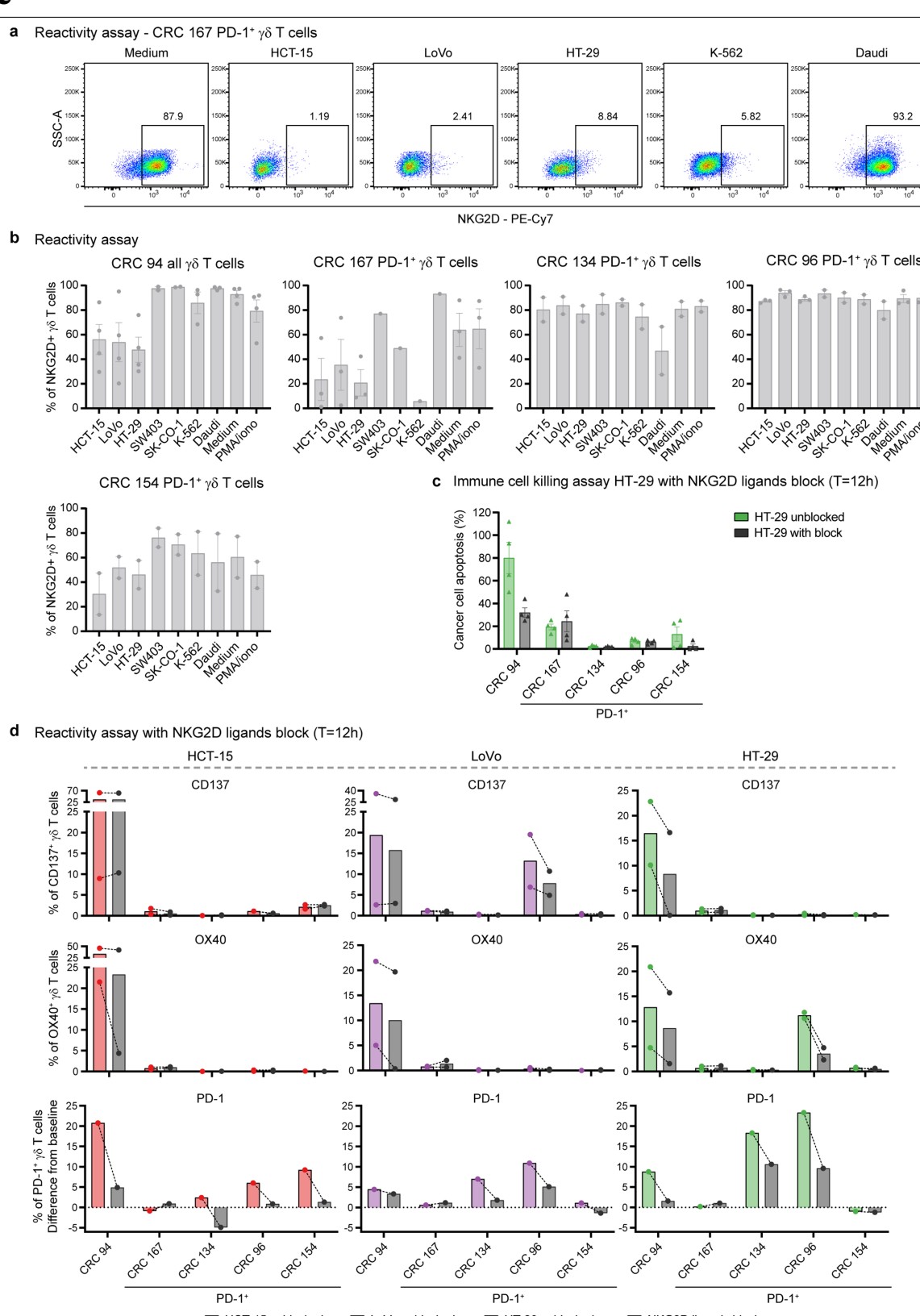

**Extended Data Fig. 8** | See next page for caption.

**Extended Data Fig. 8 | Reactivity of γδ T cells towards cancer cell lines in the presence of NKG2D ligand blocking. a**. Representative flow cytometry plots showing NKG2D expression on γδ T cells from a MMR-d colon cancer upon co-culture with cancer cell lines as compared to medium only. Gates indicate percentage of positive γδ T cells. **b**. Bar plots showing NKG2D expression on γδ T cells from MMR-d colon cancers (n = 5) upon co-culture with cancer cell lines. Medium as negative control and PMA/ionomycin as positive control are included. Bars indicate mean ± SEM. Data from at least two independent experiments except for CRC167 (SW403, SK-CO-1, K-562), depending on availability of γδ T cells. **c**. Bar plots showing the killing of HT-29 cells upon co-culture with γδ T cells with or without NKG2D ligand blocking. Bars indicate mean ± SEM of two wells with two images/well. **d**. Bar plots showing the expression of CD137 (first row), OX40 (second row), and PD-1 (third row) on γδ T cells upon co-culture of γδ T cells with CRC cell lines with or without NKG2D ligand blocking. PD-1 expression is shown as difference from baseline (medium) condition. Bars and lines indicate mean and similar experiments, respectively. Data from two independent experiments for CD137 and OX40.

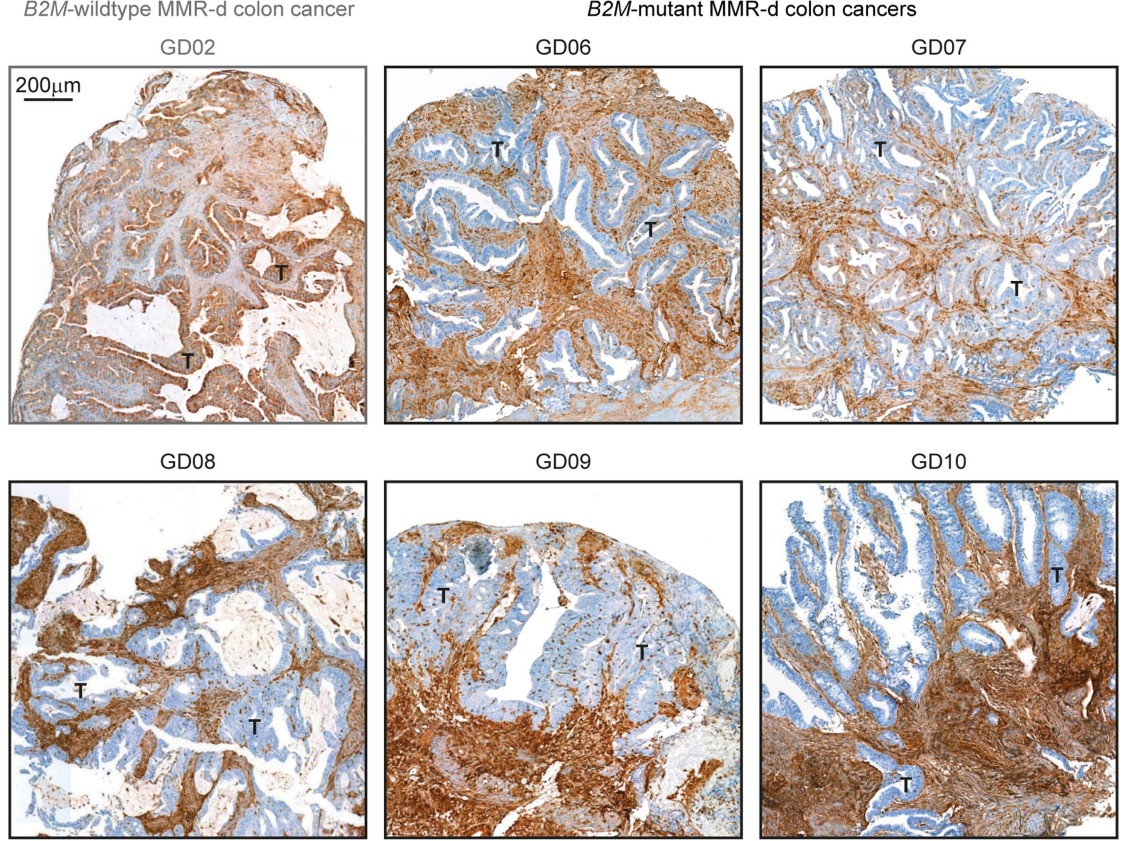

**Extended Data Fig. 9 | Loss of β2m protein expression on tumour cells in B2M-mutant MMR-d colon cancers.** Immunohistochemical analysis of β2m protein expression in FFPE tissue from all five *B2M*<sup>*MUT*</sup> MMR-d colon cancers of the NICHE cohort. A *B2M*<sup>*WT*</sup> case (GD02) staining positive for β2m is included as control. Details on the staining procedure can be found in Methods.

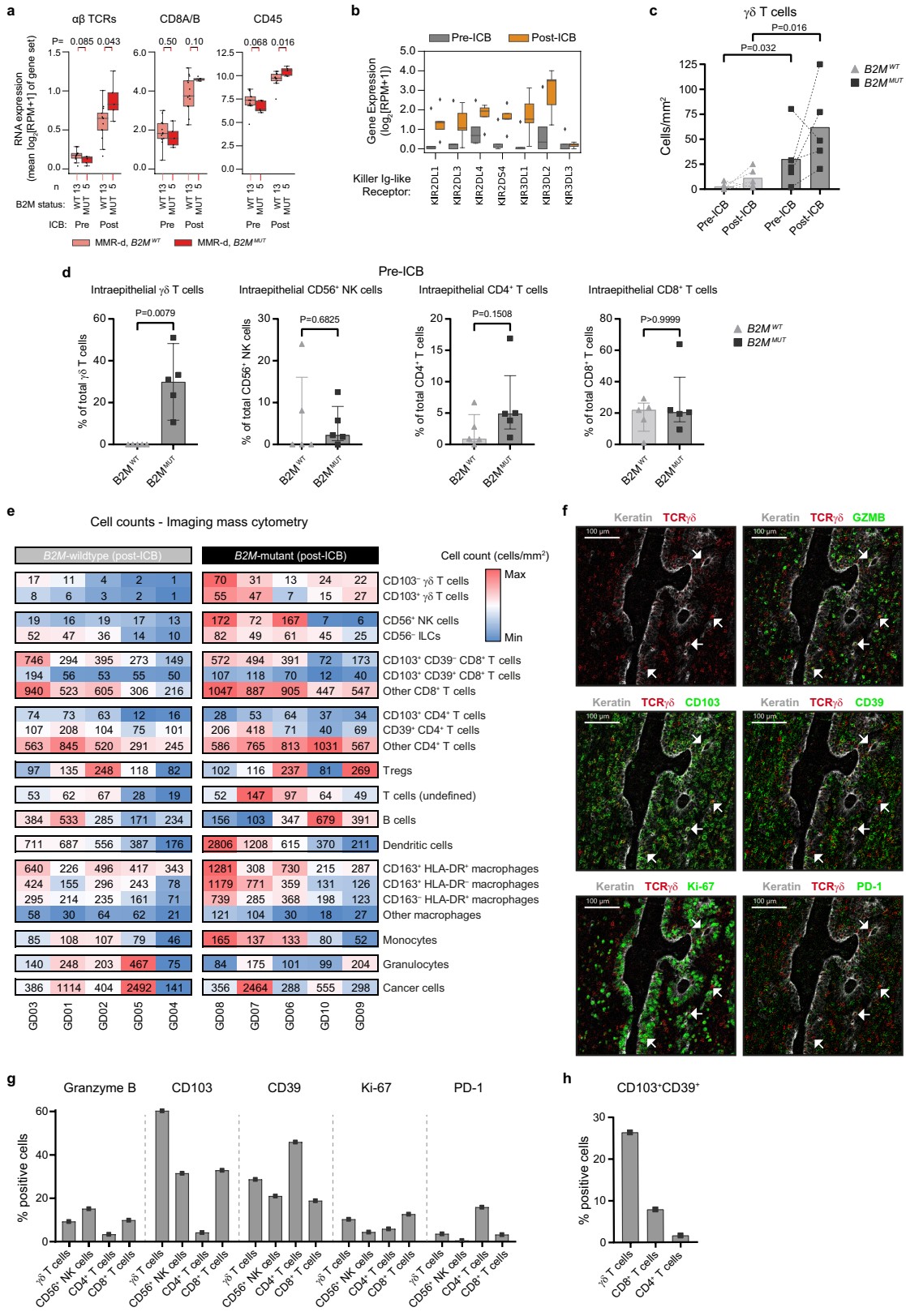

**Extended Data Fig. 10** | See next page for caption.

**Extended Data Fig. 10 | Distribution of immune cell populations in *B2M*-mutant and *B2M*-wildtype MMR-d colon cancers pre- and post-ICB treatment in the NICHE trial. a**. Immune marker gene set RNA expression in MMR-d *B2M*^WT^ (pink), and MMR-d *B2M*^MUT^ (red) cancers, before (left) and after (right) neoadjuvant ICB. Boxes, whiskers, and dots indicate quartiles, 1.5 interquartile ranges, and individual data points, respectively. Wilcoxon rank sum test-based two-sided P-values are shown. **b**. Pre- (grey) and post-ICB (orange) RNA expression of KIRs in *B2M*^MUT^ MMR-d cancers in the NICHE study (n = 5). Boxes, whiskers, and dots indicate quartiles, 1.5 interquartile ranges, and outliers, respectively. Two-sided P-values were calculated by Wilcoxon rank sum test. **c**. Pre- and post-ICB frequencies of γδ T cells by imaging mass cytometry in *B2M*^WT^ (n = 5) and *B2M*^MUT^ (n = 5) MMR-d colon cancers (corresponding to Fig. 4b). Lines indicate paired samples, dots represent individual samples. Wilcoxon rank sum test-based two-sided P-values are shown. **d**. Imaging mass cytometry-based frequencies of intraepithelial γδ T cells, CD56^+^ NK cells, CD4^+^ T cells, and CD8^+^ T cells in ICB-naive *B2M*^WT^ (n = 5) and *B2M*^MUT^ (n = 5) MMR-d colon cancers. Bars and dots indicate median ± IQR and individual samples, respectively. Wilcoxon rank sum test-based two-sided P-values are shown. **e**. Cell counts (cells/mm²) of immune populations from the imaging mass cytometry of *B2M*^WT^ (n = 5) and *B2M*^MUT^ (n = 5) MMR-d colon cancers upon ICB treatment. Colour bar is scaled per immune population. **f**. Representative images of the detection of cytotoxic (granzyme B^+^), tissue-resident (CD103^+^), activated (CD39^+^), proliferating (Ki-67^+^), and PD-1^+^ γδ T cells by imaging mass cytometry in a *B2M*^MUT^ MMR-d colon cancer upon ICB treatment. **g**. Frequencies of marker-positive γδ T cells, CD56^+^ NK cells, CD4^+^ T cells, and CD8^+^ T cells in the sole *B2M*^MUT^ MMR-d colon cancer that contained cancer cells upon ICB treatment. **h**. Frequencies of CD103^+^CD39^+^ γδ T cells, CD8^+^ T cells, and CD4^+^ T cells in the sole *B2M*^MUT^ MMR-d colon cancer that contained cancer cells upon ICB treatment.

# Reporting Summary

## Statistics

For all statistical analyses, confirm that the following items are present in the figure legend, table legend, main text, or Methods section.

| n/a | Confirmed | |
|---|---|---|
| ☐ | ☒ | The exact sample size (*n*) for each experimental group/condition, given as a discrete number and unit of measurement |
| ☐ | ☒ | A statement on whether measurements were taken from distinct samples or whether the same sample was measured repeatedly |
| ☐ | ☒ | The statistical test(s) used AND whether they are one- or two-sided<br>*Only common tests should be described solely by name; describe more complex techniques in the Methods section.* |
| ☐ | ☒ | A description of all covariates tested |
| ☐ | ☒ | A description of any assumptions or corrections, such as tests of normality and adjustment for multiple comparisons |
| ☐ | ☒ | A full description of the statistical parameters including central tendency (e.g. means) or other basic estimates (e.g. regression coefficient) AND variation (e.g. standard deviation) or associated estimates of uncertainty (e.g. confidence intervals) |
| ☐ | ☒ | For null hypothesis testing, the test statistic (e.g. *F*, *t*, *r*) with confidence intervals, effect sizes, degrees of freedom and *P* value noted<br>*Give P values as exact values whenever suitable.* |
| ☒ | ☐ | For Bayesian analysis, information on the choice of priors and Markov chain Monte Carlo settings |
| ☒ | ☐ | For hierarchical and complex designs, identification of the appropriate level for tests and full reporting of outcomes |
| ☒ | ☐ | Estimates of effect sizes (e.g. Cohen's *d*, Pearson's *r*), indicating how they were calculated |

*Our web collection on statistics for biologists contains articles on many of the points above.*

## Software and code

Policy information about availability of computer code

| Data collection | All data of The Cancer Genome Atlas (TCGA) Research Network were downloaded via the GDC data portal (https://portal.gdc.cancer.gov) on April 10th, 2019. DRUP study and Hartwig database mutation and copy number calls were generated as previously described (Priestley et al., 2019) with an optimized pipeline based on open source tools, which is freely available on GitHub (https://github.com/hartwigmedical/ pipeline5). This pipeline also outputs genome-wide number of indels in microsatellites as determined by MSIseq version 1.0.0 (Huang et al, 2015), as an integrated functionality. These data (mutations, copy numbers, indels in microsatellites) and raw RNA reads (as FASTA files) were downloaded from Hartwig Medical Foundation via Google Cloud software. For These RNA reads were aligned to the human reference genome (GRCh38 primary assembly as distributed by Gencode) with STAR software (Dobin et al., 2013), version 2.7.7a, using default settings in two-pass mode and Gencode's v29 annotation file. Flow cytometry data were collected in BD FACSDiva software (versions 8.0.2 and 9.0). Single-cell RNA libraries were sequenced on a HiSeq X Ten using paired-end 2x150 bp sequencing (Illumina). Immune cell killing assays were performed in the IncuCyte S3 system and analyzed with IncuCyte Analysis Software (version 2020B). Imaging mass cytometry data were collected using CyTOF software (version 7.0), and exported with MCD Viewer (version 1.0.5). For NICHE study (ClinicalTrials.gov: NCT03026140) RNA sequencing data, we used raw RNA reads as FASTA files, which were generated as described in the original publication (Chalabi et al., 2020). For NICHE study somatic mutation data, we used DNA sequencing of pre-treatment tumor biopsies and matched germline DNA, as described in the original publication (Chalabi et al., 2020). |
|---|---|
| Data analysis | Python 3, Jupyter Notebook 6.0.1, STAR 2.7.7a, Numpy 1.17.2, Pandas 0.25.1, Scipy 1.3.1, Statsmodels 0.10.1, Matplotlib 3.2.1, Seaborn 0.9.0, R 3.6.1, EdgeR 3.28.1, Limma (including Voom) 3.42.2, Lme4 1.1.26, MSIseq version 1.0.0, FlowJo 10.6.1, Cell Ranger 3.1.0, R version 4.1.0, Seurat R package 3.1.5, IncuCyte software 2020B, ilastik 1.3.3, CellProfiler 2.2.0, ImaCytE 1.1.4, Cytosplore 2.3.0, GraphPad Prism 9.0.0 and 9.0.1. |

For manuscripts utilizing custom algorithms or software that are central to the research but not yet described in published literature, software must be made available to editors and reviewers. We strongly encourage code deposition in a community repository (e.g. GitHub). See the Nature Portfolio guidelines for submitting code & software for further information.

## Data

Policy information about availability of data

All manuscripts must include a data availability statement. This statement should provide the following information, where applicable:
- Accession codes, unique identifiers, or web links for publicly available datasets
- A description of any restrictions on data availability
- For clinical datasets or third party data, please ensure that the statement adheres to our policy

RNA expression data (raw counts) of the colon adenocarcinoma (COAD), stomach adenocarcinoma (STAD) and Uterus Corpus Endometrium Carcinoma (UCEC) cohorts of The Cancer Genome Atlas (TCGA) Research Network is publicly available via the National Cancer Institute GDC Data Portal (https://portal.gdc.cancer.gov; cohorts COAD [https://portal.gdc.cancer.gov/projects/TCGA-COAD], STAD [https://portal.gdc.cancer.gov/projects/TCGA-STAD] and UCEC [https://portal.gdc.cancer.gov/projects/TCGA-UCEC]). Of these cohorts, mutation, copy number, purity and ploidy data were downloaded from GDC on November 11th, 2021, as the controlled access ABSOLUTE-annotated MAF file (mutations), SNP6 whitelisted copy number segments file (copy numbers), and ABSOLUTE purity/ploidy file of the TCGA PanCanAtlas project (Taylor et al, 2018). Mismatch repair-deficiency status was obtained from Thorsson et al., 2019 (TCGA Subtype = GI.HM-indel or UCEC.MSI).

Of DRUP study subjects included in this preliminary analysis across all (complete and incomplete) cohorts of the study, we included all clinical data, genomics data on B2M status and RNA-expression data of marker gene sets to this manuscript in Supplemental Table 1. The raw sequencing data of the DRUP and Hartwig cohorts can be accessed through Hartwig Medical Foundation upon approval of a research access request (https://www.hartwigmedicalfoundation.nl/en/data/data-acces-request). As determined in the original publication, NICHE study RNA- and DNA-sequencing data is deposited into the European Genome–Phenome Archive under accession no. EGAS00001004160 and are available on reasonable request to the NICHE study team for academic use and within the limitations of the provided informed consent. The single-cell RNA-sequencing data is deposited into the GEO database under accession no. GSE216534 and is publicly available. The GRCh38 primary assembly of the human reference genome was downloaded from Gencode (https://ftp.ebi.ac.uk/pub/databases/gencode/Gencode_human/release_42/GRCh38.primary_assembly.genome.fa.gz) with Gencode's matching v29 annotation file (https://ftp.ebi.ac.uk/pub/databases/gencode/Gencode_human/release_29/gencode.v29.annotation.gtf.gz) for gene expression analysis.

# Field-specific reporting

Please select the one below that is the best fit for your research. If you are not sure, read the appropriate sections before making your selection.

☒ Life sciences　　　☐ Behavioural & social sciences　　　☐ Ecological, evolutionary & environmental sciences

For a reference copy of the document with all sections, see nature.com/documents/nr-reporting-summary-flat.pdf

# Life sciences study design

All studies must disclose on these points even when the disclosure is negative.

| | |
|---|---|
| Sample size | In all analyses, sample sizes were determined by availability of patients and data. Hence, no statistical methods were used to predetermine sample sizes, but we included as many patients as possible according to availability for the different analyses.<br><br>Sample sizes were as follows:<br>TCGA colon adenocarcinoma (COAD; n=50 β2-microglobulin wild type (B2M-WT), n=7 β2-microglobulin mutant (B2M-MUT), stomach adenocarcinoma (STAD; n=48 B2M-WT, n=12 B2M-MUT), and endometrium carcinoma (UCEC; n=118 B2M-WT, n=4 B2M-MUT). LUMC imaging mass cytometry (IMC) cohort: n=17 treatment-naïve patients with mismatch repair-deficient (MMR-d) colon cancer. DRUP study: n=50 B2M-WT, n=21 B2M-MUT. Single-cell RNA-sequencing: n=5 MMR-d colon cancer patients. γδ T cell subsets culture: n=5 MMR-d colon cancer patients. NICHE IMC cohort: n=10 patients with MMR-d colon cancer (n=5 B2M-WT, n=5 B2M-MUT) from which samples were obtained before and after immune checkpoint blockade (ICB).<br><br>These sample sizes were based on previous experiences (Chalabi et al., Nat Med., 2020), and are in line with standards in the field (Middha et al., JCO Precis Oncol., 2019; Wu et al., Sci Transl Med., 2019; Wu et al., Nat Cancer., 2022). |
| Data exclusions | For the Hartwig dataset, we excluded 89 tumors from rare primary tumor locations, defined as locations with less that <20 patients in our selection. When individual patients had data available of biopsies obtained at different timepoints, we only included data of the first biopsy. In bulk transcriptomic analyses, the gene set "NK CD56dim cells" of Danaher et al. (comprising IL21R, KIR2DL3, KIR3DL1, and KIR3DL2) was excluded, as three out of four genes within this set were killer-cell immunoglobulin-like receptors (KIRs) and hence this set showed high collinearity/redundancy to the full KIR gene set. As XLC1 and XLC2 are highly expressed by tumor-infiltrating γδ T cells, these genes were removed from the NK cell marker gene set and replaced by KLRF1, which encodes the well-established NK cell marker NKp80. The resulting gene set consisted of NCR1 and KLRF1, encoding the well-established NK cell markers NKp46 and NKp80, respectively. Finally, we reduced the "cytotoxic cells" marker gene set of Danaher et al. to those genes in the set encoding cytotoxic molecules (GZMA, GZMB, GZMH, PRF1, GNLY, CTSW). For single-cell RNA-sequencing data, cells that had less than 200 detected genes and genes that were expressed in less than six cells were excluded. |
| Replication | All findings reported are replicable and the source of materials and experimental procedure are documented in detail in the Methods section. The in vitro reactivity data reported were reproducible and were shown for >= 2 independent experiments where possible (depending on availability of γδ T cells). Regarding reproducibility of the IMC data, six independent 1x1 mm Regions of Interest (ROIs) were analyzed for each tissue sample with the exception of the pre-treatment NICHE biopsies, where to to three ROIs were analyzed due to the small tissue size. For scRNA-seq data, γδ T cell clusters were identified in multiple patient samples. |

| Randomization | No randomization was applicable as this study was purely observational. |
|---|---|
| Blinding | In the DRUP study, response assessment was performed in a standardized manner by clinicians blinded to B2M status.<br>The TCGA-based differential expression analysis were standardized and unbiased (performed across all human genes) and hence did not require blinding.<br>The transcriptomics analyses of Hartwig, DRUP, and NICHE followed the same methodology as the TCGA-based analysis and were independent from judgement of individual researchers and hence did not require blinding.<br>IMC staining and analysis of the LUMC and NICHE (ICB-naive and ICB-treated) cohorts were performed blinded to group allocation. |

# Reporting for specific materials, systems and methods

We require information from authors about some types of materials, experimental systems and methods used in many studies. Here, indicate whether each material, system or method listed is relevant to your study. If you are not sure if a list item applies to your research, read the appropriate section before selecting a response.

## Materials & experimental systems

| n/a | Involved in the study |
|---|---|
| ☐ | ☒ Antibodies |
| ☐ | ☒ Eukaryotic cell lines |
| ☒ | ☐ Palaeontology and archaeology |
| ☒ | ☐ Animals and other organisms |
| ☐ | ☒ Human research participants |
| ☐ | ☒ Clinical data |
| ☒ | ☐ Dual use research of concern |

## Methods

| n/a | Involved in the study |
|---|---|
| ☒ | ☐ ChIP-seq |
| ☐ | ☒ Flow cytometry |
| ☒ | ☐ MRI-based neuroimaging |

## Antibodies

| Antibodies used | Immunohistochemistry tumor tissue samples LUMC:<br>Antibody, Clone, Supplier, Catalog number, Lot number, Dilution<br>PMS2, EP51, DAKO, M364729-2, 10122891, 1:40<br>MSH6, EPR3945, Abcam, ab92471, GR262215-21, 1:400<br>β2m, EP2978Y, Abcam, Ab75853, GR3275781-2, 1:100<br>HLA class I, HCA2, Nordic-MUbio, MUB2036P, 8152b, 1:3200<br>HLA class I, HC10, Nordic-MUbio, MUB2037P, 818b, 1:3200<br><br>Immunohistochemistry tumor tissue samples NICHE study:<br>Antibody, Clone, Supplier, Catalog number, Lot number, Dilution<br>β2m, D8P1H, Cell Signaling, 12851, 4, 1:1500<br>HLA class I, HCA2, Nordic-MUbio, MUB2036P, 7302, 1:5000<br>HLA class I, HC10, Nordic-MUbio, MUB2037P, 8108B, 1:20000<br><br>Imaging mass cytometry:<br>Antibody, Metal, Clone, Supplier, Catalog number, Lot number, Incubation time (temperature), Dilution<br>β-catenin, 89Y, D10A8, CST, 8480BF, 8, Overnight (4°C), 1:100<br>CD103, 168 Er, EPR4166(2), Abcam, ab221210, GR3355784-7, 5h (RT), 1:50<br>CD11b, 144 Nd, D6X1N, CST, 49420BF, 4, 5h (RT), 1:100<br>CD11c, 176 Yb, EP1347Y, Abcam, ab216655, GR3357092-9, 5h (RT), 1:100<br>CD14, 163 Dy, D7A2T, CST, 56082BF, 2, 5h (RT), 1:100<br>CD15, 171 Yb, MC480, CST, 4744BF, 5, Overnight (4°C), 1:100<br>CD163, 173 Yb, EPR14643-36, Abcam, 93498BF, Not available, 5h (RT), 1:50<br>CD20, 142 Nd, E7B7T, CST, 48750BF, 9179056, Overnight (4°C), 1:100<br>CD204, 164 Dy, J5HTR3, Thermo Fisher Scientific, 14-9054-95, 4338161, 5h (RT), 1:50<br>CD3, 153 Eu, EP449E, Abcam, ab271850, GR3341846-3, Overnight (4°C), 1:50<br>CD31, 147 Sm, 89C2, CST, 3528BF, Not available, Overnight (4°C), 1:100<br>CD38, 169 Tm, EPR4106, Abcam, ab226034, GR3378690-1, Overnight (4°C), 1:100<br>CD39, 157 Gd, EPR20627, Abcam, ab236038, GR3274485-6, 5h (RT), 1:100<br>CD4, 145 Nd, EPR6855, Abcam, ab181724, GR3285644-10, Overnight (RT), 1:100<br>CD45 , 149 Sm, D9M8I, CST, 13917BF, 11, Overnight (4°C), 1:50<br>CD45RO, 165 Ho, UCHL1, CST, 55618BF, 2, Overnight (4°C), 1:100<br>CD56, 167 Er, E7X9M, CST, 99746BF, 2, 5h (RT), 1:100<br>CD57, 151 Eu, HNK-1 / Leu-7, Abcam, ab269781, GR3373313-3, Overnight (4°C), 1:100<br>CD68, 143 Nd, D4B9C, CST, 76437BF, 2, Overnight (4°C), 1:100<br>CD7, 174 Yb, EPR4242, Abcam, ab230834, Not available, 5h (RT), 1:100<br>CD8α, 146 Nd, D8A8Y, CST, 85336BF, Not available, 5h (RT), 1:50<br>Cleaved caspase-3, 172 Yb, 5A1E, CST, 9664BF, 24, 5h (RT), 1:100<br>D2-40, 166 Er, D2-40, BioLegend, 916606, B316467, Overnight (4°C), 1:100<br>FOXP3, 159 Tb, D608R, CST, 12653BF, 8, Overnight (4°C), 1:50<br>Granzyme B, 150 Nd, D6E9W, CST, 46890BF, 3, 5h (RT), 1:100 |
|---|---|

Histone H3, 209, D1H2, CST, 4499BF, Not available, Overnight (4°C), 1:50
HLA-DR, 141 Pr, TAL 1B5, Abcam, ab176408, GR3384096-1, 5h (RT), 1:100
ICOS, 161 Dγ, D1K2TTM, CST, 89601BF, 4, 5h (RT), 1:50
IDO, 162 Dy, D5J4ETM, CST, 86630BF, 7, Overnight (4°C), 1:100
Ki-67, 152 Sm, 8D5, CST, 9449BF, 11, Overnight (4°C), 1:100
LAG-3, 155 Gd, D2G40TM, CST, 15372BF, Not available, 5h (RT), 1:50
p16ink4a, 175 Lu, D3W8G, CST, 92803BF, 2, Overnight (4°C), 1:100
Pan-keratin, 198 Pt, C11 and AE1/AE3, CST / BioLegend, 4545BF / 914204, 12/B302316, Overnight (4°C), 1:50
PD-1, 160 Gd, D4W2J, CST, 86163BF, 7, 5h (RT), 1:50
PD-L1, 156 Gd, E1L3NR, CST, 13684BF, 17, Overnight (4°C), 1:50
T-bet, 170 Er, 4B10, BioLegend, 644825, B298378, 5h (RT), 1:50
TCRδ, 148 Nd, H41, Santa Cruz, sc-100289, D3021, Overnight (RT), 1:50
TGFβ, 115 In, TB21, Thermo Fisher Scientific, MA5-16949, 151471, 5h (RT), 1:100
TIM-3, 154 Sm, D5D5RTM, CST, 45208BF, 9, 5h (RT), 1:100
Vimentin, 194 Pt, D21H3, CST, 5741BF, 9, Overnight (4°C), 1:50
VISTA, 158 Gd, D1L2GTM, CST, 64953BF, 7, 5h (RT), 1:100
Donkey anti-rabbit IgG, ab6701, Abcam, GR3215731-15, 1 ug/mL
Goat anti-mouse IgG, ab6708, Abcam, GR3300461-15, 1 ug/mL

FACS sorting for scRNA-sequencing:
Antibody, Fluorochrome, Clone, Supplier, Catalog number, Lot number, Dilution
CD3, PE, SK7, BD Biosciences, 345765, 9238465, 1:50
CD45, PerCP-Cy5.5, 2D1, eBioscience, 45-9459-42, 4334219, 1:160
CD7, APC, 124-1D1, eBioscience, 17-0079-42, E12980-101, 1:200
EPCAM, FITC, HEA-125, Miltenyi, 130-098-113, 5161121293, 1:60
TCRγδ, BV421, 11F2, BD Biosciences, 744870, 9340519, 1:80
Live/dead, nIR, n.a., Life Technologies, L10119, 1808830, 1:1000

CITE-sequencing and cell hashing for scRNA-sequencing:
Antibody, Barcode, Clone, Supplier, Catalog number, Lot number, Concentration
TotalSeq-C CD298/β2M (1), GTCAACTCTTTAGCG, LNH-94/2M2, BioLegend, 394661, B264730, 0.5 μg
TotalSeq-C CD298/β2M (6), GGTTGCCAGATGTCA, LNH-94/2M2, BioLegend, 394671, B264724, 0.5 μg
TotalSeq-C CD298/β2M (7), TGTCTTTCCTGCCAG, LNH-94/2M2, BioLegend, 394673, B264730, 0.5 μg
TotalSeq-C CD298/β2M (8), CTCCTCTGCAATTAC, LNH-94/2M2, BioLegend, 394675, B264722, 0.5 μg
TotalSeq-C CD298/β2M (9), CAGTAGTCACGGTCA, LNH-94/2M2, BioLegend, 394677, B264720, 0.5 μg
TotalSeq-C CD45RA, TCAATCCTTCCGCTT, HI100, BioLegend, 304163, B301193, 1 μg
TotalSeq-C CD45RO, CTCCGAATCATGTTG, UCHL1, BioLegend, 304259, B294712, 1 μg

FACS sorting for cell culturing:
Antibody, Fluorochrome, Clone, Supplier, Catalog number, Lot number, Dilution
CD103, FITC, Ber-ACT8, BD Biosciences, 550259, 2332847, 1:10
CD3, Am Cyan, SK7, BD Biosciences, 339186, 9073920, 9161745, 1:20
CD38, PE-Cy7, HIT2, eBioscience, 25-0389-42, 4319912, 1:200
CD39, APC, A1, BioLegend, 328210, B249211, 1:60
CD45RA, PE-Dazzle594, HI100, Sony, 2120730, 126470, 1:20
CD45RO, PerCP-Cy5.5, UCHL1, Sony, 2121110, 138351, 1:20
PD-1, PE, MIH4, eBioscience, 12-9969-42, 1952441, 1:30
TCR α/β, PE-Cy7, IP26, BioLegend, 306720, B303059, 1:40
TCRγδ, BV421, 11F2, BD Biosciences, 744870, 9340519, 1:80
TCR Vδ1, FITC, TS8.2, Invitrogen, TCR2730, UH286015, 1:50
TCR Vδ2, PerCP-Cy5.5, B6, BioLegend, 331424, B279957, 1:200
Live/dead, nIR, n.a., Life Technologies, L10119, 1808830, 1:1000

Immunophenotyping by flow cytometry:
Antibody, Fluorochrome, Clone, Supplier, Catalog number, Lot number, Dilution
CD16, PE, B73.1, BD Biosciences, 332779, 9045985, 1:60
CD103, FITC, Ber-ACT8, BD Biosciences, 550259, 2332847, 1:10
CD122/IL-2Rβ, BV421, TU27, BioLegend, 339010, B313155, 1:20
CD132/IL-2Rγ, APC, TUGh4, BioLegend, 338608, B293032, 1:80
CD161, BV605, DX12, BD Biosciences, 563863, 7030586, 1:20
CD25/IL-2Rα, PE-Cy7, M-A251, BD Biosciences, 557741, 9301660, 1:25
CD215/IL-15Rα, PE, JM7A4, BioLegend, 330208, B265801, 1:80
CD226/DNAM-1, BV510, DX11, BD Biosciences, 742494, 9203072, 1:150
CD3, Am Cyan, SK7, BD Biosciences, 339186, 9161745, 1:20
CD32, APC, FLI8.26, BD Biosciences, 559769, 184743, 1:20
CD38, PE-Cy7, HIT2, eBioscience, 25-0389-42, 4319912, 1:200
CD39, APC, A1, BioLegend, 328210, B249211, 1:60
CD45RA, FITC, L48, BD Biosciences, 335039, 8227525, 1:30
CD45RA, PE-Dazzle594, HI100, Sony, 2120730, 126470, 1:20
CD45RO, PerCP-Cy5.5, UCHL1, Sony, 2121110, 138351, 1:20
CD56, APC-R700, NCAM16.2, BD Biosciences, 565139, 5251693, 1:150
CD64, FITC, 10.1, BD Biosciences, 555527, 58058, 1:20
CD69, PerCP-Cy5.5, FN50, BioLegend, 310925, B266970, 1:200
CD8α, BV605, SK1, BD Biosciences, 564115, 7092, 1:100
CD94, BV605, HP-3D9, BD Biosciences, 743950, 7138571, 1:200
GITR, PE, 108-17, BioLegend, 371204, B244963, 1:50

Granzyme B, PE, GB11, eBiosciences, 12-8899-41, 1928380, 1:50
KIR2DL1, PE, HP3-E4, BD Biosciences, 556063, 86798, 1:20
KIR2DL1/S1, PE, EB6, Beckman Coulter, A09778, 12, 1:50
KIR2DL2/L3/S2, PE, GL183, Beckman Coulter, IM2278U, 200051, 1:50
KIR2DL4, PE, 181703, R&D Systems, FAB2238P, AAHO0209081, 1:10
KIR2DS4, PE, FES172, Beckman Coulter, IM3337, 200037, 1:40
KIR3DL1, PE, DX9, BD Biosciences, 555967, 121769, 1:80
KIR3DL1/S1, PE, Z27, Beckman Coulter, IM3292, 200044, 1:20
KIR3DL2, PE, #539304, R&D Systems, FAB2878P, ADBO0217051, 1:10
LAG3, PE-Cy7, 11C3C65, BioLegend, 369309, B289009, 1:100
NKG2A, APC, z199, Beckman Coulter, A60797, 200046, 1:30
NKG2C, PE, 134591, Beckman Coulter, FAB138P, LCN0818011, 1:20
NKG2D, PE-Cy7, 1D11, BD Biosciences, 562365, 9045733, 1:300
NKp44, APC, P44-8, BioLegend, 325109, B160899, 1:20
NKp46, PE, 9E2, BioLegend, 331907, B150121, 1:20
PD-1, PE, MIH4, eBioscience, 12-9969-42, 1952441, 1:30
Perforin, PE-Cy7, dG9, BioLegend, 308125, B215704, 1:20
TCRγδ, BV421, 11F2, BD Biosciences, 744870, 9340519, 1:80
TCRγδ, BV650, 11F2, BD Biosciences, 745359, 7222894, 1:40
TCR Vδ1, FITC, TS8.2, Invitrogen, TCR2730, UH286015, 1:50
TCR Vδ2, PerCP-Cy5.5, B6, BioLegend, 331424, B279957, 1:200
TIGIT, APC, 1D11, BD Biosciences, 562365, 9045733, 1:300
Live/dead, nIR, n.a., Life Technologies, L10119, 1808830, 1:1000

Characterization of cancer cell lines:
Antibody, Fluorochrome, Clone, Supplier, Catalog number, Lot number, Dilution
β2m, PE, 2M2, BioLegend, 316305, B326396, 1:100
CD112, PE, R2.525, BD Biosciences, 551057, 11886, 1:10
CD155, PE, 300907, R&D Systems, FAB25301P, AANQ0108091, 1:10
CD277/BTN3A1, PE, BT3.1, Miltenyi, 130-117-693, 5201109412, 1:50
HLA-A/B/C, FITC, W6/32, eBioscience, 11-9983-41, 4291421, 1:100
HLA-A/B/C, AF647, W6/32, BioLegend, 311414, 2132555, 1:160
HLA-E, BV421, 3D12, BioLegend, 342611, B296865, 1:20
HLA-G, APC, 87G, BioLegend, 335909, B297169, 1:20
MICA/B, PE, 6D4, BioLegend, 320906, B279674, 1:300
ULBP1, PE, 170818, R&D Systems, FAB1380P, AAJW0419061, 1:10
ULBP2/5/6, PE, 165903, R&D Systems, FAB1298P, LWE0718071, 1:20
ULBP3, PE, 166510, R&D Systems, FAB1517P, ABPX0719061, 1:20
ULBP4, PE, 709116, R&D Systems, FAB6285P, ADXP0420021, 1:20
Isotype control mouse IgG1, PE, MOPC-21, BD Biosciences, 556650, 65215
Isotype control mouse IgG2a, PE, G155-178, BD Biosciences, 556653, 8121794
Isotype control mouse IgG2a, FITC, BD Biosciences, 349051, 6194980
Isotype control mouse IgG2b, PE, 133303, R&D Systems, IC0041P, LHG1919091
Live/dead, nIR, n.a., Life Technologies, L10119, 1808830, 1:1000

Characterization of organoids:
Antibody, Fluorochrome, Clone, Supplier, Catalog number, Lot number, Dilution
β2m, FITC, 2M2, BioLegend, 316304, B261327, 1:100
HLA-A,B,C, PE, W6/32, ThermoFisher, MA1-19027, 7096730, 1:20
PD-L1, APC, MIH1, eBioscience, 17-5983-42, 4307992, 1:200
Isotype mouse IgGκ, APC, P36281, eBioscience, 17471442, 1943772, 1:200
Isotype mouse IgGκ, FITC, MOPC-21, BD Bioscience, 555748, 1054855, 1:100
Isotype mouse IgGκ, PE, MOPC-31C, BD Bioscience, 550617, 0065215, 1:20
Live/dead, nIR, n.a., Life Technologies, L10119, 2184311, 1:2000

Reactivity assay with cancer cell lines:
Antibody, Fluorochrome, Clone, Supplier, Catalog number, Lot number, Dilution
CD137, APC, 4B4-1, BD Biosciences, 550890, 0188145, 1:100
CD226/DNAM-1, BV510, DX11, BD Biosciences, 742494, 9203072, 1:150
CD3, AF700, UCHT1, BD Biosciences, 557943, 3263589, 1:400
CD39, BV421, A1, BioLegend, 328214, B308983, 1:80
CD40L, PE, TRAP1, BD Biosciences, 555700, 0267576, 1:10
NKG2D, PE-Cy7, 1D11, BD Biosciences, 562365, 9045733, 1:300
OX40, FITC, ACT35, BioLegend, 350006, B314952, 1:20
PD-1, PE, MIH4, eBioscience, 12-9969-42, 1952441, 1:30
TCRγδ, BV650, 11F2, BD Biosciences, 745359, 7222894, 1:40
Live/dead, nIR, n.a., Life Technologies, L10119, 1808830, 1:1000

Reactivity assay with organoids:
Antibody, Fluorochrome, Clone, Supplier, Catalog number, Lot number, Dilution
CD107a, FITC, H4A3, BioLegend, 328605, B27833, 1:50
CD28, - , CD28.2, eBioscience, 16028981, 2310226, 1:200 (coating)
CD3, PerCP-Cy5.5, SK7, eBioscience, 332771, 0203984, 1:20
CD4, FITC, RPA-T4, BD Bioscience, 555346, 9073869, 1:20
CD8, BV421, RPA-T8, BD Bioscience, 562429, 7082750, 1:200
IFNγ, APC, B27, BD Bioscience, 554702, 0332147, 1:40

TCRγδ, PE, 11F2, BD Bioscience , 333141, 0070146, 1:20
Live/dead, nIR, n.a., Life Technologies, L10119, 2184311, 1:2000

Blocking antibodies:
Antibody, Fluorochrome, Clone, Supplier, Catalog number, Lot number, Working concentration
DNAM-1, Purified, DX11, BD Biosciences, 559786, 0016942, 3 µg/mL
MICA/MICB, Purified, 6D4, BioLegend, 320919, B321272, 12 µg/mL
TCRγδ, Purified, 5A6.E9, Invitrogen, TCR1061, VA288448, 3 µg/mL
ULBP1, Purified, 170818, R&D Systems, MAB1380-100, JHI0317071, 1 µg/mL
ULBP2/5/6, Purified, 165903, R&D Systems, MAB1298-100, JQE0420031, 3 µg/mL
ULBP3, Purified, 166510, R&D Systems, MAB1517-100, JFY0219122, 6 µg/mL

| Validation | All antibodies were validated and titrated to determine their optimal concentration, as explained in more detail below. |

The mismatch repair (MMR), HLA class I, and β2m status were determined by immunohistochemical (IHC) staining. MMR-deficiency was defined as the lack of expression of at least one of the MMR-proteins by tumor cells in the presence of an internal positive control. Immunohistochemical detection of HLA class I expression on tumors was performed with the monoclonal HLA class I antibodies HCA2 and HC10, and classified as HLA class I positive when membranous staining of both HCA2 and HC10 was observed , weak, when membranous staining was observed but in lower levels compared to the immune-infiltration, or loss, when one or both HCA2 and HC10 do not show membranous staining, as validated previously (Ijsselsteijn et al, Revisiting immune escape in colorectal cancer in the era of immunotherapy, Br J Cancer, 2019). β2m expression was defined as positive, negative or aberrant expression in tumor cells in the presence of a positive internal control. All tissue sections were scored by a pathologist.

For imaging mass cytometry, the specificities of unconjugated antibodies was first tested by immunohistochemical (IHC) staining. Thereafter, the antibodies were conjugated to their respective metal and validated once more by IHC staining, of which the results were compared to the pre-conjugation IHC results. Antibodies were then tested by imaging mass cytometry on FFPE tonsil sections or colorectal cancer sections as positive control. The optimal staining protocol and antibody specificities have been previously validated and described by Ijsselsteijn et al. A 40-Marker Panel for High Dimensional Characterization of Cancer Immune Microenvironments by Imaging Mass Cytometry, Frontiers in Immunology, 2019.

The specificities of flow cytometry antibodies were validated by the suppliers through flow cytometry staining versus isotype controls. Detailed information and references of the antibodies can be found on https://www.bdbiosciences.com, https://www.biolegend.com, https://www.thermofisher.com, https://www.beckmancoulter.com, https://www.rndsystems.com, https://www.sonybiotechnology.com, or https://www.miltenyibiotec.com by searching using the catalog number of the antibodies listed above on the supplier's website. We further validated the antibody staining patterns on PBMC control samples with or without stimulation with PMA/ionomycin. In the flow cytometry experiments, isotype and/or FMO control samples were included.

Antibodies used for the characterization and reactivity assay with organoids:
The following antibodies have been previously used and described in Dijkstra et al. Cell 2018 and Cattaneo et al. Nature Protocols 2020 (Dijkstra et al. Generation of Tumor-Reactive T Cells by Co-culture of Peripheral Blood Lymphocytes and Tumor Organoids, Cell, Volume 174, Issue 6, 2018, doi.org/10.1016/j.cell.2018.07.009; Cattaneo et al. Tumor organoid–T-cell coculture systems, Nat Protoc 15, 15–39 (2020), doi.org/10.1038/s41596-019-0232-9):
Antibody, Fluorochrome, Clone, Supplier, Catalog number, Lot number, Dilution
HLA-A,B,C, PE, W6/32, ThermoFisher, MA1-19027, 7096730, 1:20
PD-L1, APC, MIH1, eBioscience, 17-5983-42, 4307992, 1:200
Isotype mouse IgGκ, APC, P36281, eBioscience, 17471442, 1943772, 1:200
Isotype mouse IgGκ, FITC, MOPC-21, BD Bioscience, 555748, 1054855, 1:100
Isotype mouse IgGκ, PE, MOPC-31C, BD Bioscience, 550617, 0065215, 1:20
CD107a, FITC, H4A3, BioLegend, 328605, B27833, 1:50
CD28, -, CD28.2, eBioscience, 16028981, 2310226, 1:200 (coating)
CD3, PerCP-Cy5.5, SK7, eBioscience, 332771, 0203984, 1:20
CD4, FITC, RPA-T4, BD Bioscience, 555346, 9073869, 1:20
CD8, BV421, RPA-T8, BD Bioscience, 562429, 7082750, 1:200
IFNγ, APC, B27, BD Bioscience, 554702, 0332147, 1:40
Live/dead, nIR, n.a., Life Technologies, L10119, 2184311, 1:2000

The following antibody has been referenced by Zhu et al. Nat Commun. 2019 (Zhu et al. Precisely controlling endogenous protein dosage in hPSCs and derivatives to model FOXG1 syndrome. Nat Commun. 2019 Feb 25;10(1):928. doi: 10.1038/s41467-019-08841-7):
β2m, FITC, 2M2, BioLegend, 316304, B261327, 1:100

# Eukaryotic cell lines

Policy information about cell lines

| Cell line source(s) | All cancer cell lines were derived from human: HCT-15 from colon adenocarcinoma, LoVo from colon adenocarcinoma (derived from metastatic site: left supraclavicular lymph node), HT-29 from colon adenocarcinoma, SW403 from colon adenocarcinoma, SK-CO-1 from colon adenocarcinoma (derived from metastatic side: ascites), Daudi from Burkitt lymphoma, and K-562 from chronic myeloid leukemia. All cancer cell lines derived from the ATCC. |
| Authentication | All cancer cell lines were authenticated by STR profiling. |
| Mycoplasma contamination | All cancer cell lines were tested for mycoplasma. None of the cell lines used in this study tested positive for mycoplasma. |

| Commonly misidentified lines<br>(See ICLAC register) | No commonly misidentified lines were used in this study. |
|---|---|

# Human research participants

Policy information about studies involving human research participants

| Population characteristics | DRUP study cohort:<br>The DRUP is a national, prospective, non-randomized, multi-drug and multi-tumor study, designed and conducted on behalf of the Center for Personalized Cancer Treatment (CPCT). Patients who were eligible for the study had an advanced or metastatic solid tumor, multiple myeloma or B cell non-Hodgkin lymphoma, and had exhausted standard-treatment options. A tumor genetic or protein-expression test (CPCT or regular diagnostics) had to have revealed a potentially actionable variant, for which FDA- and/or EMA-approved targeted therapy was available—but not for the tumor type in question. Across all completed and ongoing DRUP study cohorts, we performed a preliminary analysis on all patients with MMR-d tumors treated with PD-1 blockade with available WGS, RNA-seq and clinical outcome data. Detection of MMR-d by standard of care diagnostics and conformation by WGS was required for inclusion in this analysis. At inclusion in the DRUP, patients were required to be ≥18 years of age, with acceptable organ function and performance status (Eastern Cooperative Oncology Group (ECOG) score ≤ 2), and to have objectively evaluable disease of which a fresh baseline tumor biopsy could safely be obtained. Detailed baseline characteristics of this cohort are provided in Supplemental Table 1. In total, 32 out of 71 (45%) patients were female. The median (IQR) age was 68 (57-74) years.<br><br>Hartwig cohort:<br>We analyzed 2,256 metastatic tumors included in the freely available Hartwig database (Priestley et al, 2019), which (i) were MMR-p (WGS-based MSIseq6 score ≤4), (ii) had available WGS data passing standard quality controls (as defined before by Priestley et al, 2019, including a sequencing-based tumor purity of at least 20%), (iii) and had available RNA-seq data. Extensive details on the consortium and the full patient cohort has been published previously (Priestley et al, 2019). In brief, the CPCT-02 consortium was established to collect tumor biopsies of patients with advanced stage solid malignancies, in order to analyze the cancer genome by WGS and to discover predictors for systemic treatment outcome. Patients eligible for inclusion were at least 18 years and had locally advanced or metastasized solid tumors. Condition for enrollment was the possibility to safely obtain a histological biopsy from a metastasis or primary tumor prior to the start of a new line of systemic treatment. Patient-level clinical characteristics of this cohort can be found in Supplemental Table 3.<br><br>Leiden IMC cohort:<br>Primary colon cancer tissues were from 17 patients with colon cancer who underwent surgical resection of their tumor at the Leiden University Medical Center (LUMC, the Netherlands). Clinical characteristics of these patients are outlined in Supplemental Table 4. In total, 10 out of 17 (59%) patients were female. The median (range) age was 68 (36-82) years.<br><br>NICHE study cohort:<br>In addition, primary colon cancer tissues from 10 patients with colon cancer included in the NICHE study (NCT03026140) carried out at the Netherlands Cancer Institute (NKI, the Netherlands) were used for this study. A detailed description of the NICHE samples has recently been published by us in a sister journal, Nature Medicine (Chalabi, M. et al. Neoadjuvant immunotherapy leads to pathological responses in MMR-proficient and MMR-deficient early-stage colon cancers. Nat Med 26, 566-576, doi:10.1038/s41591-020-0805-8 (2020)). We have referred to this study in this manuscript.<br><br>ITO study cohort:<br>In the ITO (Immunogenicity of Tumor Organoids) study, tumor organoids were derived from two patients with MMR-d colorectal cancer included in the study at the Netherlands Cancer Institute (NKI, The Netherlands). Mismatch repair deficiency was confirmed by immunohistochemical staining for the mismatch repair proteins MSH2, MSH6, MLH1 and PMS2 in routine assessment by a pathologist. Tumor organoids generated from this study have been previously described by Dijkstra et al. (Dijkstra et al. Generation of Tumor-Reactive T Cells by Co-culture of Peripheral Blood Lymphocytes and Tumor Organoids, Cell, Volume 174, Issue 6, 2018, doi.org/10.1016/j.cell.2018.07.009). We have referred to this study in this manuscript.<br><br>Patient-derived tumor organoids:<br>PDTO-1 was derived from resection material of a primary colorectal cancer tumor. PDTO-2 was derived from a biopsy of a peritoneal metastasis of the colorectal cancer. Procedures performed with patient specimens were approved by the Medical Ethical Committee of the Netherlands Cancer Institute – Antoni van Leeuwenhoek hospital (study NL48824.031.14) and written informed consent was obtained from all patients. All PDTOs were authenticated by SNP or STR profiling. All PDTOs were tested for mycoplasma. None of the organoids used in this study tested positive for mycoplasma. |
|---|---|
| Recruitment | DRUP study cohort:<br>Patients with MMR-d cancers of various anatomical origins who exhausted all regular treatment options were recruited in 22 Dutch hospitals participating in the DRUP trial. Patients who were eligible for the study had an advanced or metastatic solid tumor, multiple myeloma or B cell non-Hodgkin lymphoma, and had exhausted standard-treatment options. A tumor genetic or protein-expression test (CPCT or regular diagnostics) had to have revealed a potentially "actionable variant", for which FDA- and/or EMA-approved targeted therapy was available—but not for the tumor type in question. In case of the subgroup of DRUP-included patients analyzed in our study, the "actionable variant" was microsatellite instability. In addition, patients were required to be ≥18 years of age, with acceptable organ function and performance status (Eastern Cooperative Oncology Group (ECOG) score ≤ 2), and to have objectively evaluable disease of which a fresh baseline tumor biopsy could safely be obtained. A detailed description of patient recruitment in the DRUP study can be found in the previous publication by us in this journal (van der Velden, D. L. et al. The Drug Rediscovery protocol facilitates the expanded use of existing anticancer drugs. Nature 574, 127-131, doi:10.1038/s41586-019-1600-x (2019).<br><br>Hartwig cohort:<br>Patients were included in 42 academic, teaching, and general hospitals across the Netherlands, under the protocol of the |

Center for Personalized Cancer Treatment (CPCT) consortium (CPCT-02 Biopsy Protocol, ClinicalTrial.gov no. NCT01855477). A detailed description of patient recruitment in the Hartwig dataset can be found in the previous publication by us in this journal (Priestley, D. L. et al. Pan-cancer whole-genome analyses of metastatic solid tumours. Nature 575, pages 210–216, doi:10.1038/s41586-019-1689-y (2019)).

Leiden IMC cohort:
The LUMC IMC cohort constitutes a consecutive cohort of treatment-naive patients with colon cancer who underwent surgical resection of their tumor at the Leiden University Medical Center (LUMC, the Netherlands).

NICHE study cohort:
Recruitment of patients included in the NICHE study can be found in the previous publication by us in a sister journal, Nature Medicine (Chalabi, M. et al. Neoadjuvant immunotherapy leads to pathological responses in MMR-proficient and MMR-deficient early-stage colon cancers. Nat Med 26, 566-576, doi:10.1038/s41591-020-0805-8 (2020)). We have referred to this study in this manuscript.

ITO study cohort:
Patients were accrued at the Netherlands Cancer Institute. Eligible patients were those with colorectal, gastric or non-small cell lung cancer that were pre-planned to undergo a study-related or standard of care biopsy procedure/tumor resection, during which an extra tissue specimen was obtained for tumor organoid generation.

| Ethics oversight | The DRUP study and the generation of the Hartwig database were initiated and conducted on behalf of the Center for Personalized Cancer Treatment (CPCT; clinicaltrials.gov: NCT02925234, NCT01855477). These studies were approved by the Medical Ethical Committee of the Netherlands Cancer Institute in Amsterdam and the University Medical Center Utrecht, respectively, and were conducted in accordance with good clinical practice guidelines and the Declaration of Helsinki's ethical principles for medical research. Written informed consent was obtained from all study subjects. This study was approved by the Medical Ethical Committee of the Leiden University Medical Center (protocol P15.282), and patients provided written informed consent. Details on the protocol of the NICHE study have been described in the previous publication by us in a sister journal, Nature Medicine (Chalabi, M. et al. Neoadjuvant immunotherapy leads to pathological responses in MMR-proficient and MMR-deficient early-stage colon cancers. Nat Med 26, 566-576, doi:10.1038/s41591-020-0805-8 (2020)). We have referred to this study in this manuscript. The ITO study (Dutch Central Committee on Research Involving Human Subjects number: NL48824.031.14) was approved by the Medical Ethical Committee of the Netherlands Cancer Institute – Antoni van Leeuwenhoek hospital and written informed consent was obtained from all patients. |
|---|---|

Note that full information on the approval of the study protocol must also be provided in the manuscript.

# Clinical data

Policy information about clinical studies

All manuscripts should comply with the ICMJE guidelines for publication of clinical research and a completed CONSORT checklist must be included with all submissions.

| Clinical trial registration | DRUP: NCT02925234; CPCT study: NCT01855477; NICHE trial: NCT03026140 |
|---|---|
| Study protocol | The study protocols will be made available upon request. |
| Data collection | Patients in the DRUP study were accrued at 22 hospitals throughout the Netherlands between 2016 and 2021. Recruitment and data collection involved dozens of medical specialists and trained research nurses, both at the site of inclusion as at the sites of coordination (Netherlands Cancer Institute, Amsterdam Medical Center, Leiden University Medical Center), which minimizes self-selection biases. Radiological response evaluations were performed according to RECIST 1.1 criteria at the site of accrual. A detailed description of the data collection of the DRUP study can be found in the previous publication by us in this journal (van der Velden, D. L. et al. The Drug Rediscovery protocol facilitates the expanded use of existing anticancer drugs. Nature 574, 127-131, doi:10.1038/s41586-019-1600-x (2019).

Patients in the Hartwig dataset were included in 42 academic, teaching, and general hospitals across the Netherlands between February 2016 and December 2019. A detailed description of data collection in the Hartwig dataset can be found in the previous publication by us in this journal (Priestley, D. L. et al. Pan-cancer whole-genome analyses of metastatic solid tumours. Nature 575, pages 210–216, doi:10.1038/s41586-019-1689-y (2019)).

A detailed description of the data collection of the NICHE study can be found in the previous publication by us in a sister journal, Nature Medicine (Chalabi, M. et al. Neoadjuvant immunotherapy leads to pathological responses in MMR-proficient and MMR-deficient early-stage colon cancers. Nat Med 26, 566-576, doi:10.1038/s41591-020-0805-8 (2020)). We have referred to this study in this manuscript. |
|---|---|
| Outcomes | For DRUP-based analyses, we used clinical benefit (defined as at least 4 months of disease control) as the primary outcome, as predefined in the study protocol. Additionally, we reported the best overall response, which was defined as a secondary outcome measure in the study protocol. Following RECIST1.1 criteria, these outcomes were assessed by the local treatment team at the site of accrual. Consistent with the study protocol, we considered these outcomes evaluable in patients who received at least two cycles of intravenous study medication, and for whom response was radiologically or clinically evaluable (at the treating physician's discretion). A detailed description of the outcomes of the DRUP study can be found in the previous publication by us in this journal (van der Velden, D. L. et al. The Drug Rediscovery protocol facilitates the expanded use of existing anticancer drugs. Nature 574, 127-131, doi:10.1038/s41586-019-1600-x (2019).

A detailed description of primary objectives, safety and feasibility, and secondary and translational endpoints of the NICHE study can be found in the previous publication by us in a sister journal, Nature Medicine (Chalabi, M. et al. Neoadjuvant immunotherapy leads to pathological responses in MMR-proficient and MMR-deficient early-stage colon cancers. Nat Med 26, 566-576, doi:10.1038/ |

# Flow Cytometry

## Plots

Confirm that:

☒ The axis labels state the marker and fluorochrome used (e.g. CD4-FITC).

☒ The axis scales are clearly visible. Include numbers along axes only for bottom left plot of group (a 'group' is an analysis of identical markers).

☒ All plots are contour plots with outliers or pseudocolor plots.

☒ A numerical value for number of cells or percentage (with statistics) is provided.

## Methodology

| | |
|---|---|
| Sample preparation | γδ T cells were isolated from MMR-deficient colon cancer tissues and expanded in vitro for functional assays. Reactivity of γδ T cells was tested by co-culturing with cancer cell lines and organoids, after which the γδ T cells were collected for analysis. |
| Instrument | For experiments at the LUMC, flow cytometry data were acquired with a FACS Canto II 3L (BD Biosciences), FACS LSR Fortessa 4L (BD Biosciences), and FACS LSR Fortessa X-20 4L (BD Biosciences). For experiments at the Netherlands Cancer Institute, flow cytometry data were acquired with a BD LSRFortessa™ Cell Analyzer SORP (BD Biosciences). |
| Software | For experiments at the LUMC, FACSDiva software version 9.0 (BD Biosciences) and FlowJo software version 10.6.1 (Tree Star Inc). For experiments at the Netherlands Cancer Institute, FACSDiVa 8.0.2 (BD Biosciences) was used. |
| Cell population abundance | γδ T cells accounted for 0.5-8.1% of the single, live lymphocyte population in the tumor samples at the time of FACS-sorting. The purity of γδ T cells was evaluated after culturing by flow cytometry and was >98%. |
| Gating strategy | In Figure 3a and Extended Data Fig. 5a:<br>1) FSC-A versus SSC-A gated on lymphocytes<br>2) FSC-A versus FSC-H gated on single cells<br>3) Live/dead-nIR versus SSC-A gated on live cells<br>4a) CD3-Am Cyan versus TCRγδ-BV421 gated on TCRγδ+ CD3+ cells<br>4b) CD3-Am Cyan versus TCRγδ-BV650 gated on TCRγδ+ CD3+ cells for detection of cytokine receptors (CD122, CD25, CD215, CD132)<br>5) Indicated markers for immunophenotyping versus TCRγδ-BV421 (or TCRγδ-BV650) for detection of marker-positive cells (see Supplemental Table 7)<br><br>In Figure 3b and Extended Data Fig. 5b:<br>1) FSC-A versus SSC-A gated on lymphocytes<br>2) FSC-A versus FSC-H gated on single cells<br>3) Live/dead-nIR versus SSC-A gated on live cells<br>4) CD112-PE versus SSC-A gated on CD112+ target cells, CD155-PE versus SSC-A gated on CD155+ target cells, CD277-PE versus SSC-A gated on CD277+ target cells, HLA-A/B/C-FITC versus SSC-A gated on HLA-A/B/C+ target cells, HLA-E-BV421 versus SSC-A gated on HLA-E+ target cells, HLA-G-APC versus SSC-A gated on HLA-G+ target cells, MICA/B-PE versus SSC-A gated on MICA/B+ target cells, ULBP1-PE versus SSC-A gated on ULBP1+ target cells, ULBP2/5/6-PE versus SSC-A gated on ULBP2/5/6+ target cells, ULBP3-PE versus SSC-A gated on ULBP3+ target cells, ULBP4-PE versus SSC-A gated on ULBP4+ target cells.<br><br>In Figure 3c, Extended Data Fig. 5c,d,f, Extended Data Fig. 8a,b,d:<br>1) FSC-A versus SSC-A gated on lymphocytes<br>2) FSC-A versus FSC-H gated on single cells<br>3) Live/dead-nIR versus SSC-A gated on live cells<br>4) CD3-AF700 versus TCRγδ-BV650 gated on TCRγδ+ CD3+ cells<br>5) CD137-APC versus SSC-A gated on CD137+ γδ T cells, OX40-FITC versus SSC-A gated on OX40+ γδ T cells, PD-1-PE versus SSC-A gated on PD-1+ γδ T cells, NKG2D-PE-Cy7 versus SSC-A gated on NKG2D+ γδ T cells<br><br>In Figure 3f and 3g:<br>1) FSC-A versus SSC-A gated on lymphocytes<br>2) SSC-H versus SSC-A gated on single cells<br>3) Live/dead-nIR versus CD3-PerCP-Cyanine5.5 gated on CD3+ live cells<br>4) TCRγδ-PE versus FSC-H gated on TCRγδ+ cells<br>5) CD4-FITC versus CD8-BV421 gated on CD4- CD8+/- cells<br>6) IFNγ-APC versus FSC-H gated on IFNγ+ cells<br><br>In Figure 3i:<br>1) FSC-A versus SSC-A gated on lymphocytes<br>2) SSC-H versus SSC-A gated on single cells<br>3) Live/dead-nIR versus CD3-PerCP-Cyanine5.5 gated on CD3+ live cells<br>4) CD8-BV421 versus TCRγδ-PE gated on CD8+/- TCRγδ+ cells<br>5) IFNγ-APC versus CD107a-FITC gated on IFNγ+ CD107a-, IFNγ+ CD107a+ and IFNγ- CD107+ cells |

In Extended Data Fig. 2a:
1) FSC-A versus SSC-A gated on lymphocytes
2) FSC-A versus FSC-W gated on single cells
3) Live/dead-nIR versus SSC-A gated on live cells
4) CD45-PerCp-Cy5.5 versus EpCAM-FITC gated on CD45+ EpCAM− cells
5) CD3-PE versus TCRγδ-BV421 gated on TCRγδ+ CD3+ cells

In Extended Data Fig. 4:
1) FSC-A versus SSC-A gated on lymphocytes
2) FSC-A versus FSC-W gated on single cells
3) Live/dead-nIR versus SSC-A gated on live cells
4) CD3-Am Cyan versus TCRγδ-BV421 gated on TCRγδ+ CD3+ cells
5) PD-1-PE versus TCRγδ-BV421 gated on PD-1− and PD-1+ γδ T cells, where possible

In Extended Data Fig. 6a:
1) FSC-A versus SSC-A gated on cells
2) FSC-A versus FSC-H gated on single cells
3) Live/dead-nIR versus FSC-A gated on live cells
4) B2M-PE as count (histogram)

In Extended Data Fig. 7b:
1) FSC-A versus SSC-A gated on cells
2) SSC-A versus SSC-H gated on single cells
3) Live/dead-nIR versus FSC-H gated on live cells
4) B2M-FITC/MHC-I-PE/PD-L1-APC as count (histogram)

All gating strategies are provided in the Extended Data Figures.

☒ Tick this box to confirm that a figure exemplifying the gating strategy is provided in the Supplementary Information.

