## [Peer Review File · Nature]

Manuscript Title: $\gamma\delta$ T cells are effectors of immunotherapy in cancers with HLA class I defects

Reviewer Comments & Author Rebuttals

Reviewer Reports on the Initial Version:

Referees' comments:

Referee #2 (Remarks to the Author):

The authors of this study have performed a very extensive analyses on $\gamma\delta$ T cells and their correlation with mismatch repair-deficient (MMR-d) colon cancers treated with immune checkpoint blockade. Although previous studies have already reported that MMR-d cancers with antigen presentation defects do respond to anti-PD-1 and have implicated immune cells such as CD4+ cells as contributors to response in these cases, this study highlights $\gamma\delta$ T cells, an effector cell that requires more study in regards to their relation to immune checkpoint blockade.

The strengths of this study are 1) a very thorough and comprehensive analyses on a not well-studied T cell population in the context of MMR-d tumors and 2) the analysis of samples before and after immune checkpoint blockade.

The manuscript in its current form is comprehensive in regards to its analyses of the $\gamma\delta$ T cells. However, because of its focus on $\gamma\delta$ T cells, several of its assays lack the inclusion of other immune cell types, and other important controls, limiting the generalizability of the findings. Inclusion of these controls could further highlight the importance of $\gamma\delta$ T cells to MMR-d cancers and in particular, its potential role as a therapeutic.

Major comments:

1) No difference is made through all the manuscript between mono or biallelic inactivation of B2M. As shown by others, B2M monoallelic loss doesn't lead to HLA class I expression loss. Is it a major control, as could restrict the applicability of the general findings of the study. All the B2M KO models mimic a biallelic B2M loss while in most cases B2M loss is monoallelic in MMRd tumors. Please display difference between mono and biallelic losses, for figure 1 and figure 4, and for the status of the cell lines in figure 3.

2) Further, MMRd tumors being hypermutated and highly diverse, one cannot exclude that B2M mutations are very subclonal, which could be suspected with low allelic frequency of B2M mutations. This consideration is major, and HLA class I / B2M stainings should be performed for the pretherapeutic B2M mutant tumors of the NICHE trial to confirm that they really have a downregulation of these proteins. Is it indeed surprising that B2M expression is upregulated in B2M mutated tumors in responders in fig 4b, results that could be explained because B2M mutant clones are eliminated or because they were not lacking class I in the first place. On figure 4 it seems also mandatory to highlight classical and non classical HLA type I gene expression as a control.

3) It has been shown by others that a major population leading to ICB response in B2M deficient MMRd tumors are CD4+ T cells. One cannot exclude in that setting a involvement of HLA class II and

cross presentation and/or participation of CD4+ cytotoxic T cells. It seems important in figure 1 and 4 images stained for $\gamma\delta$ T cells that this population doesn't colocalized with CD4+ T cells. Indeed in figures 1e and more specifically 4h, the granzyme B staining seems to be adjacent to $\gamma\delta$ T cells rather than in the same location, raising concerns that they are the effector population.

4) In Fig. 1f, the authors show an increased number of CD103+CD39+ $\gamma\delta$ T cells in B2m- cancers. What are the distributions of other cells types (i.e. CD4+ cells, NK cells)? What other cell types are increased in B2m- cancers?

In the present figure 3, while the organoid KO experiments are convincing for the role of $\gamma\delta$ T cells depending on B2M defect, the cancer cell line experiments seems not informative as lacking important controls. The authors note that "cell reactivity was most pronounced against HLA class I-negative cell lines" in Fig 3c. Nevertheless, they compare here two MMRd B2M defective / HLA deficient cell lines to a MMRp B2M+ HLA defective cell line. To have adequate controls, other cell lines need to be added, including a MMRd B2M proficient / HLA proficient cell line. The overall results of figure 3 with the HT29 MMRp cell line are interesting as it seems to show that $\gamma\delta$ T cells are also active in a MMRp context if HLA class I is absent. Panel e indeed shows that HT29 is more subject to apoptosis when exposed to $\gamma\delta$ T cells than Lovo for CRC94, suggesting an unspecific killing. To confirm that B2M loss lead to increase response whatever the MMR status, a MMRp HLA class I proficient cell line should also be added to the controls.

5) In Fig 3, the overall findings of the importance of PD-1+ $\gamma\delta$ T cells seemed to be skewed by the $\gamma\delta$ T cells from one particular sample (CRC167). Another sample, CRC96, with increased reactivity for cancer cells, seems to show comparable activity between B2m wt and mut cancer cell lines.

6) In Fig 3, the authors state that $\gamma\delta$ T cells have increased reactivity towards HLA class I-negative cancer cell lines and organoids. However these assays do not include important controls such as patient NK cells or CD4+ cells. Is the increased activity observed specific to $\gamma\delta$ T cells, PD-1+ cells, or PD-1+ $\gamma\delta$ T cells? Can the authors include data showing that other canonical effector immune cells do not exhibit this pattern of behavior?

7) In Fig 4f, the authors show that B2m mut tumors after checkpoint blockade have increased $\gamma\delta$ T cells compared to other immune cells. What are the cell counts prior to treatment for the corresponding cases?

8) In Fig 4g-h, the authors show in one case that tissue-resident $\gamma\delta$ T cells after checkpoint blockade express GZMB, Ki-67, CD103, and PD-1. They indicate that it is because of the pathological response after treatment, there are few residual cancer cells left. Do the authors see any in difference in the quality of $\gamma\delta$ T cells in B2m mut cancers compared to B2m wt MMR-d tumors? Is the difference primarily due to just quantity?

9) Do the authors have any response data related to overall increased in $\gamma\delta$ T cells? Are increased $\gamma\delta$ T cells associated with better objective response or increased survival of patients with B2m mut MMR-d cancers or MMR-d cancers in general?

Minor comments:

1) In Fig 4e, the row indicating B2m status is confusing. Indicating B2m mut is B2m (-) and B2m wt is B2m (+) is more intuitive.

Referee #3 (Remarks to the Author):

de Vries and colleagues report on gd T cells in colon cancer patients with mismatch repair-deficient tumors lacking B2M expression and increased numbers of these cells after treatment with checkpoint blockade.

1. While the detection of increased gd TcR transcripts in MMR-d cancers lacking B2M in the TCGA database is interesting, this conclusion in endometrium carcinoma is based on 3 patients, 6 patients with colon adenocarcinoma, and 13 patients with stomach adenocarcinoma. Curiously, there is an INCREASE of HLA-A, HLA-B, HLA-C and TAP1 transcripts. However, as the transcripts are measured in the entire tumor tissue, it is unclear whether the upregulation of the HLA genes are in tumor cells or other cells of the microenvironment. Is there any precedent for increased transcription of HLA class I genes when B2M is disrupted? How is this explained?

2. The authors also note increased KIR transcripts in tumors with increased gd TcR transcripts. This is expected because if there are more gd T cells, it is well known that gd T cells frequently express KIR. There is no evidence in the study that the KIR are anything other than surrogate markers for the presence of gd T cells.

3. It appears that all of the functional studies are done with gd T cells isolated from the 5 MMR-d colon cancer patients were cultured for 3-4 weeks in high concentrations of IL-2 and IL-15 and then tested against long-term colon cancer cell lines. There is nothing surprising about the ability of these cultured gd T cells to kill these cell lines and the ability to block killing of those lines that express NKG2D ligands with anti-NKG2D blocking reagents. The same result would be obtained if gd T cells were sorted and expanded in vitro from healthy donors. This has been demonstrated more than 20 years ago.

4. Why was it necessary to use CRISPR to disrupt B2M from the MMR-d patient derived tumor organoid lines? A more relevant experiment would be to establish organoids from patients who had endogenous B2M defects as it is possible that these tumors have undergone additional in vivo selection for tumors that are resistant to gd T cells or NK cells by silencing genes encoding ligands for activating innate receptors.

5. Apparently the killing assays using the organoids were also performed with long-term cytokine-cultured gd T cells, questioning whether ex vivo gd T cells have this lytic activity. Further, why were these killing assays done on plates coated with anti-CD28 monoclonal antibody?

6. In Figure 2 only a very minor frequency of gd T cells have transcripts for NKG2D, questioning whether these gd T cells actually express the NKG2D protein on the cell surface (hence potentially functional) or this is just due to NKG2D transcripts below the resolution of the RNA-Seq detection.

7. To make the conclusion that checkpoint blockade treatment affects the gd T cells in the

B2Mmutant MMR-d patients it would be necessary to compare tumor biopsy samples before checkpoint treatment and then in the same patient after therapy. Simply comparing gd T cells in B2Mwt and B2Mmutant patients post-treatment doesn't demonstrate that the checkpoint treatment is responsible for the differences in these different small patient samples.

Author Rebuttals to Initial Comments:

Referee #2 (Remarks to the Author):

The authors of this study have performed a very extensive analyses on $\gamma\delta$ T cells and their correlation with mismatch repair-deficient (MMR-d) colon cancers treated with immune checkpoint blockade. Although previous studies have already reported that MMR-d cancers with antigen presentation defects do respond to anti-PD-1 and have implicated immune cells such as CD4+ cells as contributors to response in these cases, this study highlights $\gamma\delta$ T cells, an effector cell that requires more study in regards to their relation to immune checkpoint blockade.

The strengths of this study are 1) a very thorough and comprehensive analyses on a not well-studied T cell population in the context of MMR-d tumors and 2) the analysis of samples before and after immune checkpoint blockade.

The manuscript in its current form is comprehensive in regards to its analyses of the $\gamma\delta$ T cells. However, because of its focus on $\gamma\delta$ T cells, several of its assays lack the inclusion of other immune cell types, and other important controls, limiting the generalizability of the findings. Inclusion of these controls could further highlight the importance of $\gamma\delta$ T cells to MMR-d cancers and in particular, its potential role as a therapeutic.

Major comments:

1) *No difference is made through all the manuscript between mono or biallelic inactivation of B2M. As shown by others, B2M monoallelic loss doesn't lead to HLA class I expression loss. Is it a major control, as could restrict the applicability of the general findings of the study. All the B2M KO models mimic a biallelic B2M loss while in most cases B2M loss is monoallelic in MMRd tumors. Please display difference between mono and biallelic losses, for figure 1 and figure 4, and for the status of the cell lines in figure 3.*

The reviewer makes an important point as complete loss of $\beta 2m$ expression is likely dependent on biallelic hits and we have now performed additional studies to illustrate this. First, published data indicates that, often, genetic analyses fail to identify both hits (for instance due to epigenetic silencing mechanisms) and that the discovery of disrupting mutations at the *B2M* gene, independently of the allelic status, can be associated with loss of protein expression. Middha *et al.* ([PMC6469719](https://pubmed.ncbi.nlm.nih.gov/36469719/)) recently published an extensive analysis on the correlation between *B2M* mutation status and $\beta 2m$ and HLA-I protein expression in CRC. We obtained detailed data of this study and performed a subgroup analysis restricted to the MMR-d cancers in this cohort. We found that monoallelic *B2M* mutations were also significantly associated with a high frequency of $\beta 2m$ protein expression loss (see **Figure R1**).

MMR-d cancers of Middha et al, 2019 (PMC6469719)

Figure R1. Stacked bar plot showing the percentage of patients (x-axis) with tumors with full loss of $\beta 2m$ protein expression (red), or with retained $\beta 2m$ protein expression (green), as measured through IHC, vs B2M status (y-axis). The number of patients is denoted at the y-axis. Fisher's exact test-based two-sided P-value is shown.

To provide complete insight into this important question, in the revised manuscript we now comprehensively report on *B2M* allelic status and $\beta 2m$ protein expression in several data sets from clinical studies:

- *MSI patients treated in the context of the Drug Rediscovery Protocol (DRUP; van de Velde et al. Nature 2019)*. We added data of 71 patients treated in DRUP with MMR-d tumors treated with ICB and report deep whole-genome sequencing-based allelic status of *B2M*, including phasing when possible. In line with the results of Middha et al., the majority of *B2M* alterations were bi-allelic (reported in **Figure 1** and **lines 92-95**).
- *Analysis of TCGA cohort*. We determined allelic status and report differential expression results for mono-allelic *B2M* alterations vs *B2M* wildtype, as well as (potentially) bi-allelic *B2M* alterations vs wildtype. Importantly, both analyses provided clear evidence of enrichment of $\gamma\delta$ T cells in *B2M* mutated tumors (reported in **Extended Data Fig. 1**).
- *Patients from the Leiden University Medical Center (LUMC) IMC cohort*. We have performed *B2M* mutation analysis of the 17 treatment-naïve MMR-d tumors included in **Figure 2d-f**, and added the *B2M* mutation status in **Supplemental Table 3**. Among the five samples of the $\beta 2m$ -negative group, we observed mutations in *B2M* in four (80%) samples (reported in **Supplemental Table 3**). Allelic status could not be determined from Sanger sequencing.
- *NICHE cohort (Chalabi et al. Nature Medicine 2020)*. We performed IHC to detect $\beta 2m$ protein expression in all five (treatment-naïve) *B2M*-mutant tumors in this set. In line with the results of Middha et al., 4 out of 5 (80%) tumors were negative for $\beta 2m$ protein expression (reported in **lines 218-220**).

In summary, after analyzing >340 MSI patients, we demonstrated that the majority of *B2M* defects result from biallelic alteration and that, in general, there is a good association between *B2M* mutations and $\beta 2m$ protein expression.

2) Further, MMRd tumors being hypermutated and highly diverse, one cannot exclude that *B2M* mutations are very subclonal, which could be suspected with low allelic frequency of *B2M* mutations. This consideration is major, and HLA class I / *B2M* stainings should be performed for the pretherapeutic *B2M* mutant tumors of the NICHE trial to confirm that they really have a

downregulation of these proteins. Is it indeed surprising that B2M expression is upregulated in B2M mutated tumors in responders in fig 4b, results that could be explained because B2M mutant clones are eliminated or because they were not lacking class I in the first place. On figure 4 it seems also mandatory to highlight classical and non classical HLA type I gene expression as a control.

We thank the reviewer for raising this important point. We addressed this in two ways:

1. We first assessed the frequency of subclonal *B2M* alterations in MMR-d cancers using deep whole genome (tumor-normal) sequencing data of the 71 patients in the DRUP cohort (see under comment 1 for details of DRUP). We leveraged the precise copy number profiles and purity/ploidy estimates of our WGS data to perform accurate calculations of the probability of subclonality for each mutation, as previously published by Priestley *et al.* (PMC6872491). Interestingly, not a single one of the 41 non-synonymous *B2M* mutations were subclonal (based on a subclonality probability >0.5: the situation in which the mutation is more likely subclonal than clonal). These results are well in line with the findings of Middha *et al.* (PMC6469719), who used deep panel sequencing and also found that the majority of *B2M* mutations is clonal (for all MMR-d cancers in the study by Middha *et al.*: 44 predicted clonal, 7 predicted subclonal, 16 no data available; based on upper bound of cancer cell fraction >0.8). Together, these data show that the vast majority of *B2M* mutations in MMR-d cancers are clonal events.
2. Second, as requested by the reviewer, we complemented genomics-based analyses of *B2M* status with IHC-based assessment of cancer cell-specific β 2m protein expression, which confirmed a strong correlation between *B2M* mutations and β 2m-negativity at the protein level in all tumor cells (see discussion above at comment 1).

Regarding the observation that *B2M* is upregulated in *B2M*-mutated tumors upon ICB: we agree that, based on our initial data, both explanations suggested by the reviewer could underlie this result. Given that our new data (discussed above) confirms that the majority of *B2M* alterations are (i) bi-allelic, (ii) clonal, and (iii) resulting in loss of β 2m expression, this result is likely driven by the replacement of *B2M*-negative tumor cells by *B2M*-high immune cells as a consequence of a good response to immunotherapy. However, although these are interesting observations, we aimed to make the NICHE trial data-based analyses more concise, and decided not to include this result in the revised manuscript.

3) It has been shown by others that a major population leading to ICB response in *B2M* deficient MMRd tumors are CD4+ T cells. One cannot exclude in that setting an involvement of HLA class II and cross presentation and/or participation of CD4+ cytotoxic T cells. It seems important in figure 1 and 4 images stained for $\gamma\delta$ T cells that this population doesn't colocalized with CD4+ T cells. Indeed, in figures 1e and more specifically 4h, the granzyme B staining seems to be adjacent to $\gamma\delta$ T cells rather than in the same location, raising concerns that they are the effector population.

We fully agree with the reviewer that this is an essential point. However, we would like to emphasize that we do not exclude a role for other (MHC class I-independent) mechanisms of cancer-cell recognition, like the ones mediated by CD4⁺ T cells. To better understand the role of other immune cells, and compare their relative contribution to $\gamma\delta$ T cells, we added a substantial body of work to the revised manuscript.

The observation that in MSI tumors *B2M*-mutated tumors respond as well as *B2M*-wildtype tumors to ICB, implies the existence of immune cell population(s) which compensate for the lack of CD8⁺ T cell reactivity in the *B2M*-mutant setting. This is why we set out to identify immune cell population(s) which show increased reactivity to *B2M*-mutant as compared to -wildtype tumors. Throughout the revised manuscript, we now use RNA marker gene sets and IMC to quantify multiple other immune cell types, including CD4⁺ T cells, CD56⁺ NK cells, and CD8⁺ T cells, and determine whether their levels were associated with *B2M* status in ICB-naïve cancers (**Figure 1, Figure 2d-f, Extended Data Fig. 1, Extended Data Fig. 4**), as well as in response to ICB (**Figure 4, Extended Data Fig. 11**). Consistently across 4 cohorts (DRUP, TCGA, LUMC IMC cohort, NICHE cohort), $\gamma\delta$ T cells were clearly the most prominent immune subset demonstrating increased infiltration in ICB-naïve and ICB-treated *B2M*-deficient tumors. Before and after ICB, CD4⁺ T cells were not higher in *B2M*-mutant vs -wildtype tumors (LUMC IMC cohort: **Figure 2d**; NICHE cohort: **Figure 4b**). Immunophenotyping demonstrated that, compared to $\gamma\delta$ T cells, CD4⁺ T cells showed low expression of granzyme B. (LUMC IMC cohort: **Figure 2e**; NICHE cohort: **Figure 4c, Extended Data Fig. 11f**) making them less likely the primary effector cell.

Regarding the co-localization between the gamma delta signal and granzyme B in the Imaging Cytometry figures, we apologize for the unclarity of the images included in the previous version of the manuscript. In fact, the detection of the gamma delta TCR is not trivial by IMC and the signal of the antibody against this receptor is sub-optimal. In addition, it is true that other cells in the tumor microenvironment are also expressing granzyme B together with $\gamma\delta$ T cells. In this version of the manuscript, we provide images that demonstrate, in the same tumor area, co-localization of granzyme B primarily with $\gamma\delta$ T cells and, to a lower extent, with NK cells and CD4⁺ T cells (**Figure 4c**, see below). Importantly, the quantification of granzyme B on the different immune cell subsets does not rely on the subjective observation of the images but on semi-supervised automatic cell segmentation and data-driven classification of cells into phenotypes, followed by quantification of granzyme B on those.

Fig 4c. Representative images of granzyme B-positive $\gamma\delta$ T cells infiltrating the tumor epithelium in a $B2M^{MUT}$ MMR-d colon cancer upon ICB treatment by imaging mass cytometry.

4) In Fig.1f, the authors show an increased number of $CD103+CD39+$ $\gamma\delta$ T cells in $B2m-$ cancers. What are the distributions of other cells types (i.e. $CD4+$ cells, NK cells)? What other cell types are increased in $B2m-$ cancers?

Related to the point above, we have now added the distribution of $CD56+$ NK cells, $CD4+$ T cells, and $CD8+$ T cells in treatment naïve $\beta2m$ -positive vs $\beta2m$ -negative cancers in **Figure 2d**. There are no significant differences in the distribution of immune cell types among these groups (summarized in **Figure R2** below).

Figure R2. Frequencies of immune cell types in treatment-naïve $\beta2m^+$ (n=12) and $\beta2m^-$ (n=5) MMR-d colon cancers. Bars indicate median \pm IQR. Each dot represents an individual sample. P-values were calculated by Wilcoxon rank sum test.

With regard to the number of CD103⁺CD39⁺ $\gamma\delta$ T cells, we performed a new updated imaging mass cytometry analysis to identify all immune cell types and found: (a), the difference in the number of CD103⁺CD39⁺ $\gamma\delta$ T cells in β 2m-negative vs β 2m-positive ICB-naïve MMR-d tumors does not attain statistical significance (P=0.11; **Figure R3**). However, $\gamma\delta$ T cells still showed the highest percentage of CD103⁺CD39⁺ double-positive cells in comparison to CD8⁺ T cells (and CD4⁺ T cells) (**Extended Data Fig. 4b**; summarized in **Figure R3** below). In addition, upon ICB, the sole *B2M*-mutant sample with tumor cells left showed a high percentage of CD103⁺CD39⁺ $\gamma\delta$ T cells as compared to CD8⁺ and CD4⁺ T cells (**Extended Data Fig. 11g**; summarized in **Figure R3** below). We included this in the revised results section (**lines 163-165 and lines 245-246**).

In the present figure 3, while the organoid KO experiments are convincing for the role of $\gamma\delta$ T cells depending on *B2M* defect, the cancer cell line experiments seems not informative as lacking important controls. The authors note that “cell reactivity was most pronounced against HLA class I-negative cell lines” in Fig 3c. Nevertheless, they compare here two MMRd *B2M* defective / HLA deficient cell lines to a MMRp *B2M*⁺ HLA defective cell line. To have adequate

controls, other cell lines need to be added, including a MMRd B2M proficient / HLA proficient cell line. The overall results of figure 3 with the HT29 MMRp cell line are interesting as it seems to show that $\gamma\delta$ T cells are also active in a MMRp context if HLA class I is absent. Panel e indeed shows that HT29 is more subject to apoptosis when exposed to $\gamma\delta$ T cells than Lovo for CRC94, suggesting an unspecific killing. To confirm that B2M loss lead to increase response whatever the MMR status, a MMRp HLA class I proficient cell line should also be added to the controls.

We agree with the reviewer and included extra controls in the cell line experiments. To be able to specifically investigate the effect of *B2M/HLA class I*, we created *B2M*-knockin cell lines of the *B2M/HLA*-deficient HCT-15 and LoVo cells (new **Extended Data Fig. 8**, shown below), and rescued HLA class I expression in those. Subsequently, we repeated the immune cell killing experiments and demonstrated an inhibitory effect of HLA class I expression on $\gamma\delta$ T cell-mediated cancer cell killing (**Extended Data Fig. 8**). These results are in line with the organoid experiments, showing that loss of *B2M* increases $\gamma\delta$ T cell activation. Furthermore, the reviewer correctly notes that $\gamma\delta$ T cells are capable of killing HT-29 cells (MMR-p, HLA class I positive). Indeed, we do not claim that these cells specifically kill HLA class I-negative cancers but rather that they can be effectors in that context. Of note, we have previously shown that infiltration by activated $\gamma\delta$ T cells is rare in MMR-p CRCs ([PMID21325295](https://pubmed.ncbi.nlm.nih.gov/21325295/)).

5) In Fig 3, the overall findings of the importance of PD-1+ $\gamma\delta$ T cells seemed to be skewed by the $\gamma\delta$ T cells from one particular sample (CRC167). Another sample, CRC96, with increased reactivity for cancer cells, seems to show comparable activity between B2m wt and mut cancer cell lines.

We agree with the reviewer that $\gamma\delta$ T cells display various levels of reactivity depending on the donor. CRC94 and CRC167 do indeed have most anti-tumor reactivity, which might depend on their functional status following isolation from the tumor and culture and the expression levels of stimulatory and inhibitory receptors/ligands on $\gamma\delta$ T cells and target cancer cells. For instance, CRC96-derived $\gamma\delta$ T cells are the ones with the lowest anti-tumor reactivity against the cancer cell lines, however, they display high reactivity against the *B2M*-deficient organoids. The overarching observation is that anti-tumor reactivity is consistently and strongly enriched in PD-1⁺ $\gamma\delta$ T cells.

6) In Fig 3, the authors state that $\gamma\delta$ T cells have increased reactivity towards HLA class 1-negative cancer cell lines and organoids. However, these assays do not include important controls such as patient NK cells or CD4⁺ cells. Is the increased activity observed specific to $\gamma\delta$ T cells, PD-1⁺ cells, or PD-1⁺ $\gamma\delta$ T cells? Can the authors include data showing that other canonical effector immune cells do not exhibit this pattern of behavior?

We thank the reviewer for this comment and agree that this is indeed an important focus of discussion in our manuscript. As discussed above in comment 3, we first would like to state that we do not claim that other (HLA class I-unrestricted) immune subsets, like CD4⁺ T cells and NK cells, are irrelevant in the context of immunity towards HLA class I-negative MMR-d cancers. Rather, we conclude that $\gamma\delta$ T cells are prominent cytotoxic effectors in this context, representing an important compensatory mechanism for the absence of conventional CD8⁺ T cell reactivity.

To further address this important point, we redesigned our transcriptomics and IMC analyses. For transcriptomics-based analyses, we (i) made some slight improvements to the marker gene sets used (see methods), and (ii) now show results for more immune cell types, whereas for IMC-based analyses, we adapted the methodology to quantify a multitude of additional immune cell types. Throughout, we now also report results for CD56⁺ NK cells, CD4⁺ T cells, and CD8⁺ T cells. In the ICB-naïve setting, $\gamma\delta$ T cells were clearly the most prominent immune subset demonstrating increased levels of (spontaneous) infiltration in *B2M*-mutated tumors (LUMC IMC cohort: **Figure 2d**; NICHE cohort: **Figure 4b**). Upon ICB, the $\gamma\delta$ T cell response was strongly amplified and again the most distinctive feature of *B2M*-mutated vs -wildtype tumors (**Figure 4b, Extended Data Fig. 11d**).

To gain more insight on the contribution of $\gamma\delta$ T cells, CD56⁺ NK cells, and CD4⁺ T cells, we also compared the expression of functional markers on these cells. In ICB-naïve β 2m-negative MMR-d tumors, $\gamma\delta$ T cells showed the highest expression of granzyme B (cytotoxicity), CD103 (tumor-residency), and Ki-67 (proliferation) as compared to CD4⁺ T cells and CD56⁺ NK cells (**Figure 2f, Extended Data Fig. 4a**). In the sole *B2M*-mutant case with tumor cells left after ICB, the $\gamma\delta$ T cells again showed highest expression of CD103 and Ki-67 (**Extended Data Fig. 11e-f**). Moreover, we provided new images of this specific case showing the colocalization of granzyme B, primarily with $\gamma\delta$ T cells and, to a lower extent, on NK cells and CD4⁺ T cells (**Figure 4c**, see comment 3). Of note, PD-1 expression was found on $\gamma\delta$ T cells and CD4⁺ T cells, but was generally absent on CD56⁺ NK cells (**Extended Data Fig. 4a, Extended Data Fig. 11f**), which is in line with our previous study on intra-tumoral NK cells in colorectal cancer ([PMID21325295](https://pubmed.ncbi.nlm.nih.gov/21325295/)).

Because sorting of intra-tumoral NK cells is complicated by the possible blood-contamination of tumors, where NK cells are an abundant immune cell population, we attempted to isolate

tissue-resident (CD103⁺) ILC/NK cells from colorectal tumors, but culturing of this population proved difficult.

Taken together, additional transcriptomics-based and imaging mass cytometry analyses performed for this revised manuscript confirmed that intra-tumoral $\gamma\delta$ T cells play a prominent role in the context of *B2M* defects, but we acknowledge that CD4⁺ T cells and CD56⁺ NK cells may also play a role in this context and have rephrased our statements in the results (lines 158-159, lines 229-231, lines 237-239) and discussion (lines 292-299) sections.

7) In Fig 4f, the authors show that *B2m* mut tumors after checkpoint blockade have increased $\gamma\delta$ T cells compared to other immune cells. What are the cell counts prior to treatment for the corresponding cases?

To answer this reviewer's question, we have analyzed the pre-treatment biopsies of the NICHE samples by imaging mass cytometry and included these results in Figure 4 and Extended Data Fig. 11. As expected, ICB treatment strongly increased the $\gamma\delta$ T cell counts, especially in *B2M*-mutant samples (Extended Data Fig. 11c).

8) In Fig 4g-h, the authors show in one case that tissue-resident $\gamma\delta$ T cells after checkpoint blockade express GZMB, Ki-67, CD103, and PD-1. They indicate that it is because of the pathological response after treatment, there are few residual cancer cells left. Do the authors see any in difference in the quality of $\gamma\delta$ T cells in *B2m* mut cancers compared to *B2m* wt MMR-d tumors? Is the difference primarily due to just quantity?

We have addressed this question in more detail and the differences seem to be mainly quantitative, although we cannot exclude the possibility that functional differences exist but cannot be detected with the current marker panel. We compared $\gamma\delta$ T cell functional marker expression of CD103, CD39 (described to be markers for tumor-reactive CD8⁺ T cells), Ki-67, granzyme B and PD-1 between *B2M*-mutant and *B2M*-wildtype cancers post-ICB. However,

due to very low $\gamma\delta$ T cell infiltration in *B2M*-wildtype tumors (see **Figure R4a** below), only three out of the five *B2M*-wildtype samples could be included for this comparison. Minimal differences in marker expression of $\gamma\delta$ T cells in *B2M*-wildtype vs *B2M*-mutant cancers were observed (**Figure R4b**).

9) Do the authors have any response data related to overall increased in $\gamma\delta$ T cells? Are increased $\gamma\delta$ T cells associated with better objective response or increased survival of patients with *B2m* mut MMR-d cancers or MMR-d cancers in general?

We agree with this reviewer that associating our findings to clinical response to immune checkpoint blockade is important, we performed an extensive genomic, transcriptomic and clinical analysis of 71 patients with MMR-d cancers treated with PD-1 blockade across multiple cohorts in the DRUP (DOI: [10.1038/s41586-019-1600-x](https://doi.org/10.1038/s41586-019-1600-x)). This has led to several important insights:

1. We could confirm the early finding reported by Middha *et al.* ([PMc6469719](https://pubmed.ncbi.nlm.nih.gov/36469719/)) that MMR-d tumors with *B2M* alterations show high clinical benefit rates when treated with PD-1 blockade. Interestingly, we even observed a significant enrichment for clinical benefit in the subgroup displaying genomic loss of *B2M*, as 20 out of 21 (95%) *B2M* altered patients experienced clinical benefit, vs 31 out of 50 (62%) *B2M* wildtype patients (Fisher's exact P=0.0038).
2. Reassuringly, transcriptional analysis showed that high levels of $\gamma\delta$ 1/3 T cells were (again) the only discriminatory feature of the immune infiltrate of *B2M*-altered tumors in the DRUP cohort, which directly associated the abundance of $\gamma\delta$ 1/3 T cells to the high response rates seen in this subgroup. As only 1 out of 21 *B2M* altered patients did not obtain clinical benefit, it was not possible to perform a robust analysis to test if the levels of $\gamma\delta$ 1/3 T cells or other immune cells were determinants of ICB benefit in MMR-d tumors with antigen presentation defects. The non-responsive patient did show very low levels of $\gamma\delta$ 1/3 T cells, which fits the rest of our findings but these data should not be overinterpreted.

Minor comments:

1) *In Fig 4e, the row indicating B2m status is confusing. Indicating B2m mut is B2m (-) and B2m wt is B2m (+) is more intuitive.*

We have adjusted the indication of B2M(-) for B2M-mutant and B2M(+) for B2M-wildtype according to the Reviewer's suggestion.

Referee #3 (Remarks to the Author):

de Vries and colleagues report on gd T cells in colon cancer patients with mismatch repair-deficient tumors lacking B2M expression and increased numbers of these cells after treatment with checkpoint blockade.

1. While the detection of increased gd TcR transcripts in MMR-d cancers lacking B2M in the TCGA database is interesting, this conclusion in endometrium carcinoma is based on 3 patients, 6 patients with colon adenocarcinoma, and 13 patients with stomach adenocarcinoma. Curiously, there is an INCREASE of HLA-A, HLA-B, HLA-C and TAP1 transcripts. However, as the transcripts are measured in the entire tumor tissue, it is unclear whether the upregulation of the HLA genes are in tumor cells or other cells of the microenvironment. Is there any precedent for increased transcription of HLA class I genes when B2M is disrupted? How is this explained?

This reviewer is right that the number of *B2M*-mutant MMR-d tumors in TCGA is limited. In the revised manuscript, we have now added an analysis of a cohort of 71 MMR-d cancers treated with ICB within the Drug Rediscovery Protocol (DRUP). Reassuringly, we could confirm an enrichment for $\gamma\delta$ T cells in the 21 tumors with *B2M* alterations (**Figure 1**). Please also note that, in the revised manuscript, we were able to slightly increase the TCGA patient numbers by using a more recent release of the dataset (although the increase is minimal).

We agree with the reviewer that the upregulation of HLA genes in *B2M*-mutant tumors is a curious finding, which has not only drawn our attention but also that of others. Grasso et al have also described the upregulation of HLA class I genes in *B2M*-mutant tumors, which they suggest may reflect lack of selective pressure to (genetically or epigenetically) target HLA genes when *B2M* is lost ([PMC5984687](https://pubmed.ncbi.nlm.nih.gov/35984687/)).

Although we were convinced that the conclusions of Grasso *et al.* provided a likely explanation for our data, we also experimentally tested the hypothesis that *B2M*-mutant cancer cells upregulate *HLA class I* (RNA) expression via a feedback loop. We investigated whether CRISPR KO of *B2M* in the CRC MMR-d organoids directly induced expression of HLA genes. These experiments did not demonstrate a consistent increase of expression of HLA genes upon *B2M* KO (**Figure R5**).

2. The authors also note increased KIR transcripts in tumors with increased $\gamma\delta$ TcR transcripts. This is expected because if there are more $\gamma\delta$ T cells, it is well known that $\gamma\delta$ T cells frequently express KIR. There is no evidence in the study that the KIR are anything other than surrogate markers for the presence of $\gamma\delta$ T cells.

We thank the reviewer for highlighting this observation. Indeed, the set of KIR genes that is upregulated in $B2M$ -mutant tumors in the TCGA cohort (**Figure 1d**) clusters together with the $\gamma\delta$ TCR transcripts (**Extended Data Fig. 1g**) and is exactly the set of KIRs we identify using single cell sequencing to be expressed by $V\delta 1$ and $V\delta 3$ T cells (**Figure 2c**). In line with the reviewer's suggestion, we also think that the upregulation of $\gamma\delta$ TCR transcripts and KIRs are driven by increased infiltration by $V\delta 1$ and $V\delta 3$ T cells in $B2M$ -mutant tumors, as this would be the most parsimonious explanation for the data. Nevertheless, given that we only have bulk RNA sequencing data of the TCGA (and DRUP) cohort(s), it is impossible to formally prove that this interpretation of the data is correct. Therefore, we decided to report results for $\gamma\delta$ TCR transcripts and KIRs separately.

3. It appears that all of the functional studies are done with $\gamma\delta$ T cells isolated from the 5 MMR-d colon cancer patients were cultured for 3-4 weeks in high concentrations of IL-2 and IL-15 and then tested against long-term colon cancer cell lines. There is nothing surprising about the ability of these cultured $\gamma\delta$ T cells to kill these cell lines and the ability to block killing of those lines that express NKG2D ligands with anti-NKG2D blocking reagents. The same result would be obtained if $\gamma\delta$ T cells were sorted and expanded in vitro from healthy donors. This has been demonstrated more than 20 years ago.

We thank the reviewer for this comment. We acknowledge that already in 1996, Maeurer *et al.* ([PMC2192504](https://pubmed.ncbi.nlm.nih.gov/14711111/)) demonstrated the killing ability of $V\delta 1$ cells in cancers of epithelial origin (reference 25 in the revised manuscript). The novelty related to our study, specifically regarding the function of $\gamma\delta$ T cells, relates to the differential recognition and killing of

colorectal cancer cells by PD-1⁺ Vδ1 and Vδ3 subsets and their involvement in the context of therapy with checkpoint blockade. To our knowledge, this is the first study reporting such findings. Please note that very little anti-tumor activity was observed in the PD-1-negative subsets, containing Vδ2 T cells, the most well studied γδ T cell subset in relation to reactivity to cancer cells. Furthermore, our study adds multiple other novel findings regarding γδ T cell-based anti-tumor immunity, by (i) associating the abundance of Vδ1 and Vδ3 T cells in MMR-d tumors to antigen presentation defects, (ii) associating antigen presentation defects in MMR-d tumors to paradoxically improved responsiveness to ICB, (iii) providing a comprehensive study of the phenotypes and biology of PD-1⁻ and PD-1⁺ γδ T cells in MMR-d cancers, (iv) demonstrating that ICB results in profoundly increased levels of γδ T cells in MMR-d cancers, but only when these tumors harbor antigen presentation defects. Collectively, these novelties demonstrate the potential of γδ T cells in cancer immunotherapy.

Regarding the reviewer's comment on the use of IL-2 and IL-15 we used these cytokines in order to generate sufficient cell numbers to perform functional experiments. Nevertheless, we feel confident that our *in vitro* results are not a reflection of a cytokine-induced hyperreactive state of the γδ T cells, since *in vitro* reactivity against cancer cell lines and organoids was restricted to specific subsets γδ T cell subsets, whereas all subsets were exposed to the same culture conditions.

Regarding NKG2D, we agree with the Reviewer that NKG2D is important for the killing of target cell lines that express NKG2D ligands. The γδ T cell populations used in the *in vitro* assays expressed similar, high levels of NKG2D (**Figure 3a**). However, the killing activity observed of these γδ T cell subsets against the target cell lines expression NKG2D ligands were different. This suggests that other mechanisms of γδ T cell activation might play a role.

4. Why was it necessary to use CRISPR to disrupt B2M from the MMR-d patient derived tumor organoid lines? A more relevant experiment would be to establish organoids from patients who had endogenous B2M defects as it is possible that these tumors have undergone additional *in vivo* selection for tumors that are resistant to gd T cells or NK cells by silencing genes encoding ligands for activating innate receptors.

We thank the reviewer for this question. The main goal of the tumor organoid experiments was to demonstrate a direct causal effect between the disruption of *B2M* in tumor cells and increased reactivity of γδ T cells. We considered that the ideal experimental setup to assess this was in an isogenic model because of the multiplicity of factors that can affect γδ T cell activity. Differential ligand expression between samples (e.g., MICA/B, ULBPs, DNAM-1 ligands) could potentially have an enormous impact on γδ T cell activation and would make it difficult to convincingly demonstrate the causal effect between *B2M* loss and increased activation of γδ T cells. We agree with the reviewer that it would have been interesting to do so by re-expressing intact *B2M* in *B2M*-deficient tumor organoids. However, such organoid lines were not available. Hence, we chose to perform the reverse experiment in which we knocked out *B2M* in a *B2M*-proficient organoid model. Furthermore, we now included data where we re-express *B2M* on *B2M*-deficient cell lines and demonstrate an inhibitory effect of HLA class I expression on γδ T cell activity.

5. Apparently the killing assays using the organoids were also performed with long-term

cytokine-cultured gd T cells, questioning whether ex vivo gd T cells have this lytic activity. Further, why were these killing assays done on plates coated with anti-CD28 monoclonal antibody?

We agree with the reviewer that the CD28-coated plates were not necessary in the reactivity assays with the tumor organoids. We have previously developed and reported on a T cell-organoid co-culture model in which we used CD28-coated plates (PMC6558289), and for consistency reasons we adhered to the same protocol here. We previously demonstrated that $\gamma\delta$ T cells in colorectal tumor tissues did not show expression of CD28 (PMC7063399). Using flow cytometry, we analyzed expanded $\gamma\delta$ T cells of CRC94 and confirmed that CD28 expression was not induced upon expansion (1% positivity; **Figure R6**).

Figure R6. Flow cytometry plot showing the CD28 expression on the expanded $\gamma\delta$ T cells of CRC94.

Furthermore, we investigated reactivity of healthy donor $\gamma\delta$ T cells with and without anti-CD28 antibodies and observed similar levels of IFN- γ expression (**Figure R7**).

Figure R7. Histogram showing IFN- γ expression of $\gamma\delta$ T cells from a healthy donor upon stimulation with a $B2M^{WT}$ and $B2M^{KO}$ CRC MMR-d organoid with and without anti-CD28 coated antibodies. Whiskers indicate SEM.

Taken together, these observations show that the presence of anti-CD28 coated antibodies has no effect on the reactivity assays performed.

6. In Figure 2 only a very minor frequency of gd T cells have transcripts for NKG2D, questioning whether these gd T cells actually express the NKG2D protein on the cell surface (hence potentially functional) or this is just due to NKG2D transcripts below the resolution of the RNA-Seq detection.

NKG2D is indeed generally not well covered in single-cell RNA-sequencing data, which explains the low number of *NKG2D* transcripts. However, NKG2D protein expression was detectable on the surface of >84% of the $\gamma\delta$ T cells (**Figure 3a**). Unfortunately, there is currently no good antibody to detect NKG2D for IMC, which made it impossible to add NKG2D to the IMC panel.

7. To make the conclusion that checkpoint blockade treatment affects the gd T cells in the B2Mmutant MMR-d patients it would be necessary to compare tumor biopsy samples before checkpoint treatment and then in the same patient after therapy. Simply comparing gd T cells in B2Mwt and B2Mmutant patients post-treatment doesn't demonstrate that the checkpoint treatment is responsible for the differences in these different small patient samples.

We agree with the reviewer that extending the IMC-based analyses to cover both timepoints is of clear value. In addition to the analysis of all available pre- and post-ICB samples of MMR-d cancers NICHE trial using RNA-seq, we therefore added pre- and post-ICB results for RNA-seq analysis (5 *B2M*-mutant cases and 13 *B2M*-wildtype cases) and IMC analysis (5 *B2M*-mutant cases and 5 *B2M*-wildtype, HLA class I positive cases) in the revised manuscript. Using both techniques, we estimated the levels of a broad set of immune cells and assessed how these levels were affected by ICB treatment (**Figure 4, Extended Data Fig. 11**).

Prior to ICB, $\gamma\delta$ T cells were the only immune cell type showing significantly different levels between *B2M*-mutant vs -wildtype tumors in the IMC analysis (**Figure 4, Extended Data Fig. 11**; summarized in **Figure R8** below). Upon ICB treatment, both transcriptomics- and IMC-based analyses showed a pronounced increase of $\gamma\delta$ T cell levels in *B2M*-mutant samples (**Figure R8**). Together, these data support a model in which natural immunity of $\gamma\delta$ T cells against *B2M*-mutant MMR-d tumors is present before treatment but strongly amplified upon ICB treatment.

In the revised manuscript, we now report the pre- and post-treatment levels of $\gamma\delta$ T cells, CD56⁺ NK cells, CD4⁺ T cells, and CD8⁺ T cells. $\gamma\delta$ T cells were the only subset that was consistently increased in *B2M*-mutant tumors both in pre-ICB and post-ICB samples, as detected by IMC.

Reviewer Reports on the First Revision:

Referees' comments:

Referee #2 (Remarks to the Author):

Referee #2 (Remarks to the Author):

We thank the authors for the inclusion of extensive genomic, transcriptomic, and clinical analyses of several data sets. The B2M expression analysis of several cohorts and inclusion of B2M status on the figures significantly clarifies the conclusions of the study. We also appreciate that the authors have included data on other immune cell subsets (i.e. CD4, NK cells) to better support and allow for increased generalizability of their findings. While the authors don't replicate the findings reported by others on B2M decreased expression and increased CD4+ T cells infiltration in MMRd tumors, they confirm previous findings by other teams that B2M expression/mutations don't preclude immune checkpoint blockade benefit in MMRd tumors. Overall, the authors complete the understanding of immune cells populations implicated in the response to ICB in HLA deficient tumors/B2M mutants by highlighting the role of $\gamma\delta$ T cells.

Through thorough analyses, the authors of this study show that $\gamma\delta$ T cells are likely an important effector cell that requires more study in regards to their relation to immune checkpoint blockade in mismatch repair-deficient (MMR-d) colon cancers with B2M defect, a common feature of these cancers. Although there are limitations in that the authors did not directly test the functionality of other immune cells populations, their analyses support their findings and the revised text does not overstate their role or suggest that other immune cells are not important. Additional immune cells analyses on the pre and post ICB samples would be of particular interest to better understand the role of $\gamma\delta$ T cells.

Major comments:

1) The authors have adequately addressed the concerns about the disconnect between B2M expression vs mutational status in regard to their analyzed samples. As the authors have correctly indicated that mutation does not necessarily indicate loss of expression, their additional comprehensive analyses do show that B2M monoallelic losses are indirectly enriched for B2M loss of expression over wildtype. Although their NICHE cohort sample is small, the IHC analysis of 5 samples directly supports their samples have bonafide B2M loss of expression.

2) As mentioned in comment 1, we thank the authors' comprehensive genomics and IHC analysis of NICHE trial samples and feel that they have adequately addressed our concerns about B2M expression in the context of their study.

3) In the revised manuscript, the authors have now clearly included CD4 T cells and other non-CD8+ immune cells into their analysis and figures. At least in their samples, CD4 T cells do not seem to play a significant role in response to ICB-naïve MMRd cancers. The inclusion of "In this context, other HLA class I-independent immune subsets, like NK cells and (neoantigen-specific) CD4+ T cells may also contribute. The latter were shown to play an important role in response to ICB (as reported in

murine B2M-deficient MMR-d cancer models⁴⁰), and may also support $\gamma\delta$ T cell-driven responses." in lines 294-297 helps to put their findings in the context of already published ICB B2M-mutant MMRd cancer literature.

4) We thank the authors for the additional analyses on other immune cell types as well as further clarification of the CD103+CD39+ population. We agree with the author's conclusions above. However, given that only CD103+ $\gamma\delta$ T cells and not double positive CD103+CD39+ $\gamma\delta$ T cells were significant, we feel that the language "ICB treatment of MMR-d colon cancer profoundly increases the intra-tumoral presence of activated, cytotoxic, and proliferating $\gamma\delta$ T cells" in lines 247-248 should be tempered to exclude "activated, cytotoxic, and proliferating" or rephrased without "profoundly". Moreover, patients in that cohort received dual PD-1 and CTLA4 blockade, and therefore it is difficult to address the question of the enrichment in $\gamma\delta$ T cells when patients receive Anti PD-1 monotherapy and generalize the findings to ICB in general. On top of that, the NICHE trial included only patients with early-stage disease, and recent reports showed that early/locally advanced MMRd tumors are exquisitely sensitive to PD-1 monotherapy, much more than in the advanced setting. Therefore, it seems difficult to generalize the findings to any setting, and these data suggest at best two different type of immune response in B2M WT and B2M mutants MMRd tumors as already shown by others unless data from the advanced setting is provided. I also don't understand why there are 5 patients with samples analyzed in panel c post ICB while only one patient is reported in panel d?

It would have been of interest to report other populations of immune cells in that panel to better understand the specific role of $\gamma\delta$ T cells in the response to ICB in B2M mutant MMRd cancers. One could indeed imagine in that setting that the response is so deep and sustained that all type of immune cells could be recruited. While the authors clearly show that B2M mutants MMRd tumors are enriched in intratumoral $\gamma\delta$ T cells, Immune exclusion of TILs is a common phenomenon in MMRd tumors, and these peritumoral immune cells can be allowed to traffic inside the tumor during ICB treatment and participate in the immune response. No difference in the manuscript is done between the intra and peritumoral compartments which is debatable while their overall results suggest interestingly that the intratumoral traffic of $\gamma\delta$ T cells is increased in MMRd B2M mutants tumors compared to other population, and may likely been involved in the initial and quick response to ICB. Showing different dynamics of T cells in B2M WT vs mutants before and after ICB would be extremely supportive and would help to conclude on the role of $\gamma\delta$ T cells.

5) It remains unclear as to why the killing of HT-29 cells is better than Lovo cells and is not commented on in the revised manuscript. Although these additional experiments are certainly suggestive that B2M loss increases sensitivity to $\gamma\delta$ T cells in the context of those cancer cell lines, the sensitivity of HT-29 cells is not commented on or addressed and the reader is left to wonder why HT-29 was selected as a positive comparison. To complete the number of controls needed, HT-29 should have B2M knocked out and show increased sensitivity to $\gamma\delta$ T cells.

6) We understand the limitations and thank the authors for the additional IMC analyses. Given that all the additional data unfortunately does not exclude the role of CD4+ T cells and NK cells, the rephrased statements address this concern.

7) Although the sample size is small, we thank the authors for the additional analyses.

8) We thank the authors for the additional comprehensive analysis on MMRd cancers across several cohorts. As noted above, this concern has already been addressed in earlier comments.

Minor comments:

1) In Fig 4e, the row indicating B2m status is confusing. Indicating B2m mut is B2m (-) and B2m wt is B2m (+) is more intuitive.

We have adjusted the indication of B2M(-) for B2M-mutant and B2M(+) for B2M-wildtype according to the Reviewer's suggestion.

On Fig 4a, there are two designations for B2M which is confusing. It is labeled mutant in the legend but altered in the x-axis. Using mutant is more intuitive.

Referee #3 (Remarks to the Author):

The revised manuscript by de Vries and colleagues is significantly improved and addresses many, but not all, of the concerns. Important new data provide evidence that gd T cells are increased due to immune checkpoint therapy by comparing samples from the same patient before and after treatment in MMR-d cancers deficient in B2M.

There are still concerns about over-interpretation of the findings related to the in vitro functional assays using gd T cells that have been cultured for weeks in IL-2.

In vitro culture of Vd1 and Vd3 gd T cells from healthy individuals would almost certainly also kill these tumor cells lines and be blocked with anti-NKG2D, so the relevance to the current study are questionable. Even conventional CD8+ ab T cells when extensively cultured in IL-2 gain the ability to kill via NKG2D, so these experiments add virtually nothing to the study and could be deleted from the manuscript.

The authors ignore that fact that KIR are highly polymorphic and are expressed on gd T cells and NK cells in a stochastic fashion, even when HLA class I ligands in the individual are not recognized by the expressed KIR. Whether the precise KIR expressed by these gd T cell can recognize the HLA class I genes possessed by the individual patients is never addressed. The authors imply that the lack of killing of the B2M wildtype cells is mediated by KIR; however, this is never demonstrated. gd T cells can also express other inhibitory receptors for HLA class I- for example NKG2A and LILRB1. The authors should acknowledge that the KIR serve as a marker for these gd T cells, but their functional relevance is never directly addressed and is largely a distraction.

Using CRISPR to delete B2M in organoids from patients that were wildtype for B2M really doesn't address the issue of whether organoids produced from patients with B2M deficiency would be susceptible to gd T cell killing as these tumor may have been "edited" in vivo to lack other ligands for

innate receptors that would render them resistant to gd T cells or NK cells. Therefore, again the experiments presented showing that long-term cultured gd T cells can kill these organoids doesn't add a lot to the study.

Referee #4 (Remarks to the Author):

In this elegant study, De Vries and colleagues report very solid and convincing data on the importance of unconventional gd T cells in the surveillance and immunotherapy (in the context of immune checkpoint blockade, ICB) of cancers deficient in the antigen presentation process required for conventional T cells. The authors demonstrate that the antigen presentation defects associate with increased gd T cell infiltration in tumors; and – critically – that these lymphocytes are important immune effectors of ICB in cases of β 2m/antigen presentation deficiency, where CD8+ T cells are unable to respond. Moreover, the authors pinpoint PD-1+ gd T cells (comprising Vd1+ and Vd3+ cells) as the main cytotoxic subset responsible for anti-tumour reactivity. The study is novel, timely and presents a thorough and comprehensive analysis of gd T cells in a setting where their contribution has been neglected. As such, it is of very high relevance to the Immunology and Oncology (or Immuno-Oncology) communities, and will be of interest to the wide readership of Nature.

This being said, there are some important aspects to be clarified and/or further discussed in the manuscript, even at this late stage of review/ revision, before it is suitable for publication:

1) Is the presence of gd T cells with potential to respond to ICB restricted to MMR-d? Or are β 2m mutations/HLA class-I inactivation sufficient? i.e, could gd T cells respond to ICB in MMR-p tumors that are β 2m mutants/HLA-deficient? Is the mutational burden associated with the presence of gd T cells?

2) [lines 85-91] In the DRUP cohort, the authors report that from the 21 patients with B2Mmut 20 experienced a clinical benefit, of which 12 experienced a partial and 3 a complete response. It is unclear what was the outcome of the remaining 5 patients that experienced clinical benefit. Moreover, it would be interesting to understand if there is any association between the amount of gd T cells and the extension of response to ICB. Do the 3 patients that showed complete responses correspond to the 3 patients that in Fig.1g show >0.5 RNA expression of Vd1Vd3 loci in the pan-cancer analysis?

3) How much do the 3-4 week-expanded PD-1+ gd T cells resemble the naturally occurring PD-1+ gd TILs in terms of granzyme, perforin, granulysin, IFN γ , NKG2D, DNAM-1 expression? Doesn't the expansion induce a stronger anti-tumour phenotype that does not resemble TIL phenotype? Do expanded PD-1+ gd T cells respond to ICB in vitro? Is their cytotoxicity enhanced upon anti-PD-1 blockade?

4) [line187] The authors state that they observed the "highest killing of HLA class I-negative HCT-15 cells" which is accurate, however, the authors abstained to comment on the higher killing observed against HT-29 vs LoVo which goes against the expected pattern, if indeed gd T cell-mediated killing is to be explained by HLA class I expression/ β 2m mutational status.

5) [lines 217-220] The authors state that 4 out of 5 B2Mmut cancers show complete pathological clinical response, and that 4 out of 5 mutated cases show loss of β 2m expression. Are the 4 cases with loss of β 2m expression the same 4 that show complete pathological clinical response?

6) Do gd T cells maintain the expression of KIRs upon ICB in B2Mmut tumors?

7) What drives gd T cell accumulation in $\beta 2m$ -deficient tumors? Could it be that the lack of a CD8+ T cell response creates the conditions for gd T cell expansion? It is also unclear what is the mechanism that the authors suggest to be involved in the sensing by PD-1+ gd T cells of the lack of HLA class I expression. Are KIRs involved? These topics should be covered in the discussion.

Minor issue – references:

Instead of refs. 27 and 28, which are dispensable and can be easily replaced by the review article #20 (and even better, this ref 20 from 2015 should be replaced by the updated version in Nat Rev Cancer, Silva-Santos et al. 2019), I would strongly encourage to cite, either in the introduction or in the discussion, these two recent papers from the Hayday group that show the prognostic value of V δ + gd T cells in breast and lung cancer patients:

- Wu Y, Kyle-Cezar F, Woolf RT, Naceur-Lombardelli C, Owen J, Biswas D, Lorenc A, Vantourout P, Gazinska P, Grigoriadis A, Tutt A, Hayday A. An innate-like V δ 1+ $\gamma\delta$ T cell compartment in the human breast is associated with remission in triple-negative breast cancer. *Sci Transl Med*. 2019 Oct 9;11(513):eaax9364. doi: 10.1126/scitranslmed.aax9364

- Wu Y, Karasaki T, Veeriah S, Czyzewska-Khan J, Morton C, Joseph M, Hessey S, Reading J, Georgiou A, Al-Bakir M, TRACERx Consortium, McGranahan N, Jamal-Hanjani M, Hackshaw A, Biswas D, Usaite I, Angelova M, Boeing S, Hayday AC, Swanton C. A local human V δ 1 T-cell population is associated with survival in nonsmall-cell lung cancer. *Nat Cancer*. 2022 Jun;3(6):696-709. doi: 10.1038/s43018-022-00376-z

Author Rebuttals to First Revision:

Referee #2 (Remarks to the Author):

We thank the authors for the inclusion of extensive genomic, transcriptomic, and clinical analyses of several data sets. The B2M expression analysis of several cohorts and inclusion of B2M status on the figures significantly clarifies the conclusions of the study. We also appreciate that the authors have included data on other immune cell subsets (i.e. CD4, NK cells) to better support and allow for increased generalizability of their findings. While the authors don't replicate the findings reported by others on B2M decreased expression and increased CD4+ T cells infiltration in MMRd tumors, they confirm previous findings by other teams that B2M expression/mutations don't preclude immune checkpoint blockade benefit in MMRd tumors. Overall, the authors complete the understanding of immune cells populations implicated in the response to ICB in HLA deficient tumors/B2M mutants by highlighting the role of $\gamma\delta$ T cells. Through thorough analyses, the authors of this study show that $\gamma\delta$ T cells are likely an important effector cell that requires more study in regards to their relation to immune checkpoint blockade in mismatch repair-deficient (MMR-d) colon cancers with B2M defect, a common feature of these cancers. Although there are limitations in that the authors did not directly test the functionality of other immune cells populations, their analyses support their findings and the revised text does not overstate their role or suggest that other immune cells are not important. Additional immune cells analyses on the pre and post ICB samples would be of particular interest to better understand the role of $\gamma\delta$ T cells.

We appreciate the supportive and insightful comments from Referee #2 and would like to sincerely thank for this contribution towards the improvement of our manuscript. According to this reviewer virtually all previous comments were adequately addressed and we now only address the remaining outstanding comments.

Major comments:

4) *We thank the authors for the additional analyses on other immune cell types as well as further clarification of the CD103+CD39+ population. We agree with the author's conclusions above. However, given that only CD103+ $\gamma\delta$ T cells and not double positive CD103+CD39+ $\gamma\delta$ T cells were significant, we feel that the language "ICB treatment of MMR-d colon cancer profoundly increases the intra-tumoral presence of activated, cytotoxic, and proliferating $\gamma\delta$ T cells" in lines 247-248 should be tempered to exclude "activated, cytotoxic, and proliferating" or rephrased without "profoundly".*

We thank the Referee for these important comments. In line with the Referee's suggestion, we have rephrased the sentence without "profoundly" (**lines 257-259**).

Moreover, patients in that cohort received dual PD-1 and CTLA4 blockade, and therefore it is difficult to address the question of the enrichment in $\gamma\delta$ T cells when patients receive Anti PD-1 monotherapy and generalize the findings to ICB in general. On top of that, the NICHE trial included only patients with early-stage disease, and recent reports showed that early/locally advanced MMRd tumors are exquisitely sensitive to PD-1 monotherapy, much more than in the advanced setting. Therefore, it seems difficult to generalize the findings to any setting, and these data suggest at best two different type of immune response in B2M WT and B2M mutants MMRd tumors as already shown by others unless data from the advanced setting is provided. I also don't understand why there are 5 patients with samples analyzed in panel c post ICB while only one patient is reported in panel d?

With regard to the question of the Referee why only one patient is reported in **panel d (Figure R3, Response to Reviewers, version R1)**, we aimed to show the localization and phenotype of $\gamma\delta$ T cells in the only B2M-mutant case that still contained cancer cells following ICB treatment. This enabled us to

specifically demonstrate that $\gamma\delta$ T cells expressing granzyme B (**Figure 4c**) as well as other markers of activation (**Extended Figure 11e**) displayed intraepithelial localization and were therefore in direct contact with cancer cells.

It would have been of interest to report other populations of immune cells in that panel to better understand the specific role of $\gamma\delta$ T cells in the response to ICB in B2M mutant MMRd cancers. One could indeed imagine in that setting that the response is so deep and sustained that all type of immune cells could be recruited. While the authors clearly show that B2M mutants MMRd tumors are enriched in intratumoral $\gamma\delta$ T cells, Immune exclusion of TILs is a common phenomenon in MMRd tumors, and these peritumoral immune cells can be allowed to traffic inside the tumor during ICB treatment and participate in the immune response. No difference in the manuscript is done between the intra and peritumoral compartments which is debatable while their overall results suggest interestingly that the intratumoral traffic of $\gamma\delta$ T cells is increased in MMRd B2M mutants tumors compared to other population, and may likely been involved in the initial and quick response to ICB.

We agree with the Referee that the analysis of intratumoral vs. peritumoral localization of lymphocytes would be important to study. For this purpose, we examined the frequencies of intraepithelial $\gamma\delta$ T cells, CD56⁺ NK cells, CD4⁺ T cells, and CD8⁺ T cells in ICB-naïve *B2M^{WT}* and *B2M^{MUT}* MMR-d colon cancers from the NICHE cohort. The complete responses that occurred in most of the patients make it impossible to define intratumoral or peritumoral localization of lymphocytes following responses to ICB. In the pre-ICB setting, we found that a large proportion of the $\gamma\delta$ T cells showed an intraepithelial localization in *B2M^{MUT}* MMR-d colon cancers as compared to the *B2M^{WT}* samples (Wilcoxon rank sum-based two-sided P=0.0079). We thank the Referee for this insightful comment, and have now included this data in the new **Extended Data Figure 12d** (see below) and in lines **245-247** of the Results section.

Showing different dynamics of T cells in B2M WT vs mutants before and after ICB would be extremely supportive and would help to conclude on the role of $\gamma\delta$ T cells.

Concerning the dynamics of other populations of immune cells to better understand the specific role of $\gamma\delta$ T cells in response to ICB, these are shown in **Figure 4b** (dynamics of NK cells, CD4⁺ T cells, and CD8⁺ T cells before and after ICB). Furthermore, the frequency of other immune cell populations including Tregs, B cells, dendritic cells, macrophages, monocytes, and granulocytes can be found in **Extended Data Figure 12e**.

Lastly, and in agreement with the Referee's suggestion, we restrain from generalizing our findings regarding the role of $\gamma\delta$ T cells in ICB. We have specifically contextualized our findings regarding (colorectal) cancers that have lost $\beta 2m$ expression. In the discussion section, we also specify the implications of our findings specifically in the context of MMR-d (colorectal) cancers and other malignancies with frequent HLA class I defects (**lines 301-306**).

5) It remains unclear as to why the killing of HT-29 cells is better than Lovo cells and is not commented on in the revised manuscript. Although these additional experiments are certainly suggestive that B2M loss increases sensitivity to $\gamma\delta$ T cells in the context of those cancer cell lines, the sensitivity of HT-29 cells is not commented on or addressed and the reader is left to wonder why HT-29 was selected as a positive comparison. To complete the number of controls needed, HT-29 should have B2M knocked out and show increased sensitivity to $\gamma\delta$ T cells.

As mentioned by this Referee, the $\gamma\delta$ T cells of one patient (CRC94) demonstrate high killing capacity of HT-29 cells, which are indeed HLA class I-positive (**Figure 3e**). Importantly, we do not wish to claim that HLA class I loss is a necessary condition for $\gamma\delta$ T cell activity. We rather state that $\gamma\delta$ T cells remain functional in cancers that have lost HLA class I expression and that their activity is further enhanced in this setting. Also, our previous work (PMID: [31270164](https://pubmed.ncbi.nlm.nih.gov/31270164/)) shows that MMR-deficient tumors (all carrying HLA class I-defects) have high frequencies of PD-1⁺ $\gamma\delta$ T cells, whereas these cells were nearly absent in MMR-proficient tumors (91% of those were determined HLA class I-positive). How V δ 1 $\gamma\delta$ T cells specifically recognize their target cells remains largely unknown. HT-29 cells express ULBP1 and butyrophilin 3A1 (**Figure 3b**), which may be ligands for NKG2D and $\gamma\delta$ TCR, respectively, on $\gamma\delta$ T cells. Taken together, $\gamma\delta$ T cells may also kill HLA class I-positive tumors, however, they are generally not present in those in CRC. We have adapted the manuscript throughout to avoid any misleading statements that would suggest HLA class I loss is a necessary condition for $\gamma\delta$ T cell activity. In line with this, we now report on the killing of both HLA class I-negative and -positive cell lines in the results section (**lines 188-189**).

Unfortunately, because of the additional experiments in the previous review round, not enough $\gamma\delta$ T cells were left from the patients reported in this study to provide a comparison for HT-29 and a B2M-knockout HT-29 model (we prioritized the remaining $\gamma\delta$ T cells for the requested experiments to introduce B2M in the $\beta 2m$ -defective cell lines). However, we instead tested this specific question in our organoid models, where B2M-knockout variants were generated from B2M-wildtype organoids. In this experimental setting, increased sensitivity to $\gamma\delta$ T cells was observed when the organoid models were B2M-knocked out (**Figure 3g**).

Minor comments:

1) In Fig 4e, the row indicating B2m status is confusing. Indicating B2m mut is B2m (-) and B2m wt is B2m (+) is more intuitive.

We have adjusted the indication of B2M(-) for B2M-mutant and B2M(+) for B2M-wildtype according to the Reviewer's suggestion.

On Fig 4a, there are two designations for B2M which is confusing. It is labeled mutant in the legend but altered in the x-axis. Using mutant is more intuitive.

We thank the Referee for the suggestion and have now adjusted the B2M-altered into B2M-mutant.

Referee #3 (Remarks to the Author):

The revised manuscript by de Vries and colleagues is significantly improved and addresses many, but not all, of the concerns. Important new data provide evidence that gd T cells are increased due to immune checkpoint therapy by comparing samples from the same patient before and after treatment in MMR-d cancers deficient in B2M.

There are still concerns about over-interpretation of the findings related to the in vitro functional assays using gd T cells that have been cultured for weeks in IL-2.

We thank the Referee for these important comments and insights and tried to address those below. Of note, we still consider the *in vitro* assays to be significant to support the message of the paper while we recognize the limitations that are inherent to such an experimental setting. As such we have adapted the revised manuscript to avoid overinterpretation of findings.

In vitro culture of Vd1 and Vd3 gd T cells from healthy individuals would almost certainly also kill these tumor cells lines and be blocked with anti-NKG2D, so the relevance to the current study are questionable. Even conventional CD8+ ab T cells when extensively cultured in IL-2 gain the ability to kill via NKG2D, so these experiments add virtually nothing to the study and could be deleted from the manuscript.

The relevance of the experiments performed with isolated $\gamma\delta$ T cells from tumors is not so much related to the demonstration that V δ 1 and V δ 3 can kill but rather the demonstration that PD-1⁺ (V δ 1 or V δ 3⁺) $\gamma\delta$ T cell subsets, infiltrating colorectal cancer, are capable of tumor cell-killing while PD-1⁻ subsets were largely ineffective, although some still contained V δ 1 and V δ 3 $\gamma\delta$ T cells. This also provides a link to the successful treatment of these patients with ICB. Furthermore, when comparing the phenotypes of PD-1⁻ and PD-1⁺ $\gamma\delta$ T cells that have been cultured in identical conditions, we observed clear differences in the expression of natural cytotoxicity receptors (NCRs) and KIRs, among others, which were generally increased on PD-1⁺ $\gamma\delta$ T cells as compared to PD-1⁻ subsets (**Figure 3a, Extended Data Figure 6a**). We have now clarified this in the results section (**line 180**).

To further clarify the main aim and message of the *in vitro* results, we have rephrased **lines 184-189**, and **lines 195-197**. Of note, we also observe differential anti-tumor activity of V δ 1 and V δ 3 cells, depending on the source of $\gamma\delta$ T cells and the target tumor cells. These observations do not support the generalization that these cells are “universal killers” (as suggested by the Referee) and provide a more differentiated insight.

The authors ignore that fact that KIR are highly polymorphic and are expressed on gd T cells and NK cells in a stochastic fashion, even when HLA class I ligands in the individual are not recognized by the expressed KIR. Whether the precise KIR expressed by these gd T cell can recognize the HLA class I genes possessed by the individual patients is never addressed. The authors imply that the lack of killing of the B2M wildtype cells is mediated by KIR; however, this is never demonstrated. gd T cells can also express other inhibitory receptors for HLA class I- for example NKG2A and LILRB1. The authors should acknowledge that the KIR serve as a marker for these gd T cells, but their functional relevance is never directly addressed and is largely a distraction.

The Referee is correct that no hard evidence is provided for the involvement of KIRs in the activity of $\gamma\delta$ T cells against HLA class I-negative cancer cells. In line with the Referee's suggestion, we have rephrased the involvement of KIRs as a potential mechanism of recognition of HLA class I phenotypes in the results section (**lines 122-123**). Furthermore, we now discuss other potential mechanisms of recognition of HLA class I phenotypes in the discussion section (**lines 293-298**), including the inhibitory receptors NKG2A and LILRB-1. We also investigated the expression of NKG2A directly *ex vivo* on $\gamma\delta$ T cells by scRNAseq, and

found that NKG2A is mainly expressed on V δ 2 subsets (**Figure 3c**). Furthermore, we examined the expression of LILRB1 on $\gamma\delta$ T cells from four MMR-deficient CRC tumors (**Figure R1**, see below), three of which were included in the *in vitro* assays of this study. Absent to low LILRB1 expression was found on the $\gamma\delta$ T cells.

Using CRISPR to delete B2M in organoids from patients that were wildtype for B2M really doesn't address the issue of whether organoids produced from patients with B2M deficiency would be susceptible to $\gamma\delta$ T cell killing as these tumor may have been "edited" in vivo to lack other ligands for innate receptors that would render them resistant to $\gamma\delta$ T cells or NK cells. Therefore, again the experiments presented showing that long-term cultured $\gamma\delta$ T cells can kill these organoids doesn't add a lot to the study.

We do not claim that $\gamma\delta$ T cells are specifically active against HLA class I-negative cancer cells but rather that their activity can be modulated by HLA class I expression on target cells. This was the main objective of this specific experiment and is currently the only experiment included in the manuscript where KO of *B2M* was performed on otherwise β 2m/HLA class I proficient cells. As such, we consider it important to maintain it as part of the manuscript. We agree, however, that performing the same experiment on β 2m-deficient organoids to assess whether they remain sensitivity to $\gamma\delta$ T cell killing, despite their previous exposure to $\gamma\delta$ T cells *in vivo*, would be an elegant complementary approach. Hence, we conducted an additional experiment where we demonstrate V δ 1 $\gamma\delta$ T cell reactivity against an organoid, PDO-3 *B2M*^{MUT}, that was derived from a β 2m-deficient colorectal cancer (**Figure R2**, see below). This organoid was grown from a patient included in the DRUP (hmfSampleId HMF000872B) and harbored two deleterious *B2M* mutations: a c.37_38delCT frameshift mutation (p.Leu15fs) and a c.68-2A>G splice acceptor variant. FACS analysis confirmed lack of HLA class I protein expression at the tumor cell surface of these organoids (**Figure R2a**). Of note, the reactivity against this *B2M*^{MUT} organoid line was superior to the reactivity detected against a β 2m-proficient MMR-d organoid and its *B2M*-KO variant (**Figure R2**).

Referee #4 (Remarks to the Author):

In this elegant study, De Vries and colleagues report very solid and convincing data on the importance of unconventional gd T cells in the surveillance and immunotherapy (in the context of immune checkpoint blockade, ICB) of cancers deficient in the antigen presentation process required for conventional T cells. The authors demonstrate that the antigen presentation defects associate with increased gd T cell infiltration in tumors; and – critically – that these lymphocytes are important immune effectors of ICB in cases of β 2m/antigen presentation deficiency, where CD8+ T cells are unable to respond. Moreover, the authors pinpoint PD-1+ gd T cells (comprising Vd1+ and Vd3+ cells) as the main cytotoxic subset responsible for anti-tumour reactivity. The study is novel, timely and presents a thorough and comprehensive analysis of gd T cells in a setting where their contribution has been neglected. As such, it is of very high relevance to the Immunology and Oncology (or Immuno-Oncology) communities, and will be of interest to the wide readership of Nature.

This being said, there are some important aspects to be clarified and/or further discussed in the manuscript, even at this late stage of review/ revision, before it is suitable for publication:

1) *Is the presence of gd T cells with potential to respond to ICB restricted to MMR-d? Or are β 2m mutations/HLA class-I inactivation sufficient? i.e, could gd T cells respond to ICB in MMR-p tumors that are β 2m mutants/HLA-deficient? Is the mutational burden associated with the presence of gd T cells?*

We thank the Referee for this important question. In the revised results section, we now show results of an analysis of 2,256 MMR-p metastatic cancers from the Hartwig database (PMID: 31645765). We leveraged our previous work (PMID: 31645765) showing that a WGS-based MSIsq (PMID: 26306458) score >4 was highly sensitive and specific for MMR-d, in order to identify MMR-p cases in the Hartwig database. Among these cancers, *B2M* mutations were rare: only 36 out of 2,256 (1.6%) MMR-p cancers harbored a *B2M* alteration. The *B2M* mutation pattern differed between MMR-p and MMR-d cancers, as MMR-p cancers never demonstrated cases of multiple *B2M* mutations within a single tumor, whereas such cases occurred frequently in MMR-d cancers (**Figure 1c**, new **Extended Data Figure 1h**, see below). Nevertheless, all *B2M* alterations in MMR-p cancers were clonal and most were bi-allelic, through bi-allelic deletion or mutation plus LOH, suggesting that (a large fraction of) these alterations are functionally relevant (**Extended Data Figure 1h**). Interestingly, also in MMR-p cancers we found that *B2M* alterations were strongly associated with increased expression of *TRDV1/TRDV3* loci (linear regression-based two-sided $P=2.2 \times 10^{-17}$, adjusted for tumor type; new **Extended Data Figure 1i**, see below). We consider these results to be an important addition to our story, as they suggest that our findings may extend to MMR-p cancers, which further broadens their relevance.

Extended Data Fig. 1

h

i

Extended Data Figure 1h. The allelic alteration status of $B2M$ in the Hartwig cohort of MMR-p cancers. **i.** The RNA expression of $V\delta 1+V\delta 3$ loci in MMR-p $B2M^{WT}$ (gray), and MMR-p $B2M^{MUT}$ (red) cancers in the Hartwig cohort, stratified per primary tumor location. Of cancers originating from the skin, 200 out of 208 (96%), including all $B2M^{MUT}$ cases, were melanomas. Boxes, whiskers, and dots indicate quartiles, 1.5 interquartile ranges, and individual data points, respectively. The linear regression-based, two-sided, primary tumor location-adjusted P-value for association of $B2M$ status with $V\delta 1+V\delta 3$ loci expression is shown.

2) [lines 85-91] In the DRUP cohort, the authors report that from the 21 patients with $B2M^{mut}$ 20 experienced a clinical benefit, of which 12 experienced a partial and 3 a complete response. It is unclear what was the outcome of the remaining 5 patients that experienced clinical benefit. Moreover, it would be interesting to understand if there is any association between the amount of gd T cells and the extension of response to ICB. Do the 3 patients that showed complete responses correspond to the 3 patients that in Fig. 1g show >0.5 RNA expression of $V\delta 1/V\delta 3$ loci in the pan-cancer analysis?

We thank the Referee for pointing out this unclarity. We have now updated the results section and specify that the remaining 5 $B2M^{MUT}$ patients with clinical benefit experienced a durable stable disease as best overall response. Although the sole $B2M^{MUT}$ patient without clinical benefit of ICB showed very low infiltration of $V\delta 1/V\delta 3$ T cells, expression of $V\delta 1/V\delta 3$ loci was not significantly different between subgroups as defined by their best overall response. As such, the 3 patients with a complete response did not stand out in terms of their expression of $V\delta 1/V\delta 3$ loci when compared to other $B2M^{MUT}$ patients.

3) How much do the 3-4 week-expanded PD-1+ gd T cells resemble the naturally occurring PD-1+ gd TILs in terms of granzyme, perforin, granulysin, IFN γ , NKG2D, DNAM-1 expression? Doesn't the expansion induce a stronger anti-tumour phenotype that does not resemble TIL phenotype? Do expanded PD-1+ gd T cells respond to ICB in vitro? Is their cytotoxicity enhanced upon anti-PD-1 blockade?

We examined the phenotype of PD-1+ ($V\delta 1/V\delta 3^+$) $\gamma\delta$ T cells directly *ex vivo* as compared to 3-4 weeks after expansion for the expression of granzyme B, perforin, NKG2D, DNAM-1, and IFN- γ (Figure R3, see below). Of note, these data were generated by different methodologies (FACS, scCyTOF, scRNA-seq) for other purposes and must therefore be interpreted with caution.

Directly *ex vivo*, the PD-1+ ($V\delta 1/V\delta 3^+$) $\gamma\delta$ T cells generally showed high expression levels of granzyme B/perforin, which is preserved upon expansion of the cells. PD-1+ ($V\delta 1/V\delta 3^+$) $\gamma\delta$ T cells expressed high levels of IFN γ directly *ex vivo*, which decreased upon 3-4 weeks of expansion (measured for CRC134 and

CRC96). DNAM-1 was expressed in 0-10% of the PD-1⁺ (V δ 1/V δ 3⁺) $\gamma\delta$ T cells directly *ex vivo*, which was comparable to the levels observed 3-4 weeks after expansion. For the interpretation of the results of NKG2D, it is important to note that NKG2D transcripts are generally not well represented in scRNA-seq data. For two patients, CRC94 and CRC167, of which the $\gamma\delta$ T cells showed the highest killing of target cancer cells, we measured the protein level of NKG2D directly *ex vivo*. For CRC94 the $\gamma\delta$ T cells showed high levels of NKG2D directly *ex vivo* (87.0%), which was preserved upon culturing of the cells (99.7%). For CRC167, 21% of the $\gamma\delta$ T cells showed NKG2D expression directly *ex vivo*, whereas after expansion 99.4% of the $\gamma\delta$ T cells were NKG2D-positive. Here, NKG2D-positive $\gamma\delta$ T cells might have been positive selected during the expansion of the cells. Taken together, the 3-4 week-expanded PD-1⁺ (V δ 1/V δ 3⁺) $\gamma\delta$ T cells largely resemble the phenotypes of intratumoral PD-1⁺ (V δ 1/V δ 3⁺) $\gamma\delta$ T cells in terms of cytotoxicity markers and DNAM-1 expression, while the results for NKG2D are variable.

The scRNA-seq data on the $\gamma\delta$ T cells directly *ex vivo* shown in **Figure R3** is included in **Figure 2c** and **Extended Data Figure 2b** of the revised manuscript, and the FACS data of the $\gamma\delta$ T cells after expansion is shown in **Figure 3a** and **Extended Data Figure 6a** of the manuscript.

	Directly ex vivo			After expansion
	FACS	scCyTOF	scRNA-seq	FACS
$\gamma\delta$ T cells CRC 94				
Granzyme B	NA	40,00	NA	93,10
Perforin	NA		NA	83,70
NKG2D	87,00	NA	NA	99,70
IFN- γ	NA	NA	NA	NA
PD-1⁺ $\gamma\delta$ T cells CRC 167				
Granzyme B	NA	93,50	89,67	62,60
Perforin	NA		72,30	47,10
NKG2D	21,00	NA	5,16	99,40
DNAM-1	NA	NA	9,86	3,74
IFN- γ	NA	NA	73,71	NA
PD-1⁺ $\gamma\delta$ T cells CRC 134				
Granzyme B	NA	NA	100,00	74,70
Perforin	NA	NA	33,33	91,10
NKG2D	NA	NA	16,67	99,40
DNAM-1	NA	NA	0,00	0,20
IFN- γ	NA	NA	50,00	10,10
PD-1⁺ $\gamma\delta$ T cells CRC 96				
Granzyme B	NA	NA	90,48	87,30
Perforin	NA	NA	90,48	25,10
NKG2D	NA	NA	9,52	99,40
DNAM-1	NA	NA	4,76	0,20
IFN- γ	NA	NA	61,90	8,69
PD-1⁺ $\gamma\delta$ T cells CRC 154				
Granzyme B	NA	72,30	70,97	99,50
Perforin	NA		67,74	48,40
NKG2D	NA	NA	6,45	98,40
DNAM-1	NA	NA	0,00	0,02
IFN- γ	NA	NA	54,84	NA

Figure R3. Phenotype of PD-1⁺ (V δ 1/V δ 3⁺) $\gamma\delta$ T cells directly *ex vivo* versus after expansion. Table showing the percentage of marker-positive PD-1⁺ (V δ 1/V δ 3⁺) $\gamma\delta$ T cells from MMR-d colon cancers measured directly *ex vivo* versus 3-4 weeks after expansion. The data were obtained with different techniques (FACS, scCyTOF, scRNA-seq).

We agree with the Referee that it would be interesting to examine whether the cytotoxicity of PD-1⁺ (Vδ1/Vδ3⁺) γδ T cells is enhanced upon PD-1 blockade. We tested PD-1⁺ (Vδ1/Vδ3⁺) γδ T cell reactivity against target cancer cell lines in the presence of anti-PD-1 blocking antibodies (20 μg/mL nivolumab) *in vitro*. In these experiments, we did not find an enhancement of cytotoxicity of γδ T cells. It is likely that to address this question one would require an *in vivo* model where γδ T cells can become chronically activated and that also allows their interaction with additional immune cell subsets.

4) [line187] *The authors state that they observed the “highest killing of HLA class I-negative HCT-15 cells” which is accurate, however, the authors abstained to comment on the higher killing observed against HT-29 vs LoVo which goes against the expected pattern, if indeed gd T cell-mediated killing is to be explained by HLA class I expression/b2m mutational status.*

We fully agree with this comment, the γδ T cells of one patient (CRC94) demonstrated high killing capacity of HT-29 cells, which are indeed HLA class I-positive (**Figure 3e**). To avoid any misleading statements in the manuscript, we have clarified that the γδ T cells showed *in vitro* killing of both HLA class I-negative and -positive cell lines in the results section (**lines 188-189**). Importantly, we do not claim that HLA class I loss is a necessary condition for γδ T cell activity, but state that γδ T cells remain functional in cancers that have lost HLA class I expression and that their activity is further enhanced in this setting. Also, our previous work (PMID: 31270164) showed that MMR-deficient tumors (all carrying HLA class I-defects) showed high frequencies of PD-1⁺ γδ T cells, whereas these cells were nearly absent in MMR-proficient tumors (91% of those were determined HLA class I-positive). Hence, γδ T cells may also kill HLA class I-positive tumors, however, they are generally not present in those in the context of CRC.

5) [lines 217-220] *The authors state that 4 out of 5 B2Mmut cancers show complete pathological clinical response, and that 4 out of 5 mutated cases show loss of β2m expression. Are the 4 cases with loss of β2m expression the same 4 that show complete pathological clinical response?*

We have revised the β2m immunohistochemistry data and determined that, in fact, all 5 cases displaying deleterious mutations in *B2M* also presented loss of protein expression. We apologize for this oversight in the previous version of the manuscript. The immunohistochemical detection results have now been evaluated by two pathologists and images of the 5 cases are presented below and in supplementary data (new **Extended Data Figure 11**, see below).

Extended Data Fig. 11

Extended Data Figure 11. Loss of $\beta 2m$ protein expression on tumor cells in $B2M$ -mutant MMR-d colon cancers. Immunohistochemical analysis of $\beta 2m$ protein expression in FFPE tissue from all five $B2M^{MUT}$ MMR-d colon cancers of the NICHE cohort. A $B2M^{WT}$ case (GD02) staining positive for $\beta 2m$ is included as control. Details on the staining procedure can be found in Methods.

6) Do gd T cells maintain the expression of KIRs upon ICB in $B2M$ mut tumors?

Although our extensive analyses of pre- and post-treatment samples of the NICHE trial show clear overexpression of KIRs upon ICB, which in turn cluster tightly together with *TRDV1* transcripts, these data are only correlative and do not formally prove that the KIRs originate from $\gamma\delta$ T cells. Unfortunately, we were not able to confirm this point within a reasonable timeframe. In the revised manuscript, we have rephrased our conclusions regarding KIRs (**lines 122-123 and lines 293-298**) and describe KIRs only as a potential mechanism in which $\gamma\delta$ T cells may recognize HLA class I-negative cancers. We hope this is more precise and sufficiently addresses the Referee's concerns.

7) What drives gd T cell accumulation in $\beta 2m$ -deficient tumors? Could it be that the lack of a $CD8^+$ T cell response creates the conditions for gd T cell expansion? It is also unclear what is the mechanism that the authors suggest to be involved in the sensing by $PD-1^+$ gd T cells of the lack of HLA class I expression. Are KIRs involved? These topics should be covered in the discussion.

We agree with the Referee that these are important future research questions. In the discussion section, we now discuss i) the outstanding question how $\gamma\delta$ T cells accumulate in $\beta 2m$ -deficient tumors, ii) whether the lack of $CD8^+$ T cell activity might favor the activity of other immune effector cells, iii) potential

mechanisms of recognition of HLA class I phenotypes including KIRs, NKG2A, and LILRB1. Of these markers, PD-1⁺ (Vδ1/Vδ3⁺) γδ T cells showed most pronounced expression of KIRs.

Minor issue – references:

Instead of refs. 27 and 28, which are dispensable and can be easily replaced by the review article #20 (and even better, this ref 20 from 2015 should be replaced by the updated version in Nat Rev Cancer, Silva-Santos et al. 2019), I would strongly encourage to cite, either in the introduction or in the discussion, these two recent papers from the Hayday group that show the prognostic value of Vδ + γδ T cells in breast and lung cancer patients:

- Wu Y, Kyle-Cezar F, Woolf RT, Naceur-Lombardelli C, Owen J, Biswas D, Lorenc A, Vantourout P, Gazinska P, Grigoriadis A, Tutt A, Hayday A. An innate-like Vδ1⁺ γδ T cell compartment in the human breast is associated with remission in triple-negative breast cancer. *Sci Transl Med.* 2019 Oct 9;11(513):eaax9364. doi: 10.1126/scitranslmed.aax9364

- Wu Y, Karasaki T, Veeriah S, Czyzewska-Khan J, Morton C, Joseph M, Hessey S, Reading J, Georgiou A, Al-Bakir M, TRACERx Consortium, McGranahan N, Jamal-Hanjani M, Hackshaw A, Biswas D, Usaita I, Angelova M, Boeing S, Hayday AC, Swanton C. A local human Vδ1 T-cell population is associated with survival in nonsmall-cell lung cancer. *Nat Cancer.* 2022 Jun;3(6):696-709. doi: 10.1038/s43018-022-00376-z

In line with the Referee's suggestions, we have now replaced refs. 20, 27, and 28 with the updated review article (Nat Rev Cancer, Silva-Santos et al. 2019) and added the two recent papers from the Hayday group in **line 273** of the Discussion section.

Reviewer Reports on the Second Revision:

Referees' comments:

Referee #4 (Remarks to the Author):

I would like to acknowledge the authors' detailed and convincing response to my previous concerns.
No further issues or comments.